# SceneNAT: Masked Generative Modeling for Language-Guided Indoor Scene Synthesis

## Abstract

We present SceneNAT, a single-stage masked non-autoregressive Transformer that synthesizes complete 3D indoor scenes from natural language instructions through only a few parallel decoding passes, offering improved performance and efficiency compared to prior state-of-the-art approaches. SceneNAT is trained via masked modeling over fully discretized representations of both semantic and spatial attributes. By applying a masking strategy at both the attribute level and the instance level, the model can better capture intra-object and inter-object structure. To boost relational reasoning, SceneNAT employs a dedicated triplet predictor for modeling the scene's layout and object relationships by mapping a set of learnable relation queries to a sparse set of symbolic triplets (subject, predicate, object). Extensive experiments on the 3D-FRONT dataset demonstrate that SceneNAT achieves superior performance compared to state-of-the-art autoregressive and diffusion baselines in both semantic compliance and spatial arrangement accuracy, while operating with substantially lower computational cost.

## 1 Introduction

Generating realistic 3D indoor scenes that faithfully adhere to text instructions is essential for applications like virtual staging (Ji et al., 2023; Tukur et al., 2023; 2024), AR/VR content creation (Dionisio et al., 2013; Coelho et al., 2022; Li et al., 2020), and large-scale synthetic data generation (Fernandez-Chaves et al., 2022; Deitke et al., 2022; Huang et al., 2023a; Fu et al., 2024; Yang et al., 2024a; Khanna et al., 2024). For these applications to be practically useful—especially in interactive design or simulation settings—users must be able to flexibly and reliably steer the output to match their intent. This requires fine-grained control over both object content and spatial layout through natural language. Such instructions would encode complex object attributes, spatial relationships, and implicit commonsense constraints, making it particularly difficult to ground language accurately and generate structured layouts in a controllable manner.

As a result, many prior works have proposed and advanced approaches based on either autoregressive or diffusion models. Representative autoregressive approaches (Wang et al., 2021; Paschalidou et al., 2021; Leimer et al., 2022; Liu et al., 2023a; Para et al., 2023; Zhao et al., 2024) formulate scene synthesis as set generation conditioned on room type and layout, enabling diverse and structured layout generation. However, their sequential decoding suffers from inefficient generation and error accumulation, and it also prevents bidirectional reasoning due to reliance on pre-defined decoding orders across scene elements (e.g., category → attributes → layout). More recently, diffusion models (Ho et al., 2020; Nichol & Dhariwal, 2021) have been widely adopted in this domain (Zhai et al., 2023; Wei et al., 2023; Tang et al., 2024; Lin & Mu, 2024; Maillard et al., 2024; Hu et al., 2024; Zhai et al., 2025; Bai et al., 2025), harnessing their strengths in capturing fine-grained semantics and spatial detail. However, their reliance on hundreds of sampling steps leads to high computational cost and latency, limiting their applicability in real-time settings.

In parallel with these architectural advancements, another important line of research has studied controllable 3D scene generation, aiming to better align outputs with detailed user input. Prior works (Paschalidou et al., 2021; Wang et al., 2021; Hu et al., 2024; Zhai et al., 2023; 2025; Feng et al., 2023; Tang et al., 2024; Lin & Mu, 2024; Sun et al., 2024; Bai et al., 2025) have sought to enhance controllability using various conditioning inputs, including floorplans, scene graphs, and natural language instructions.

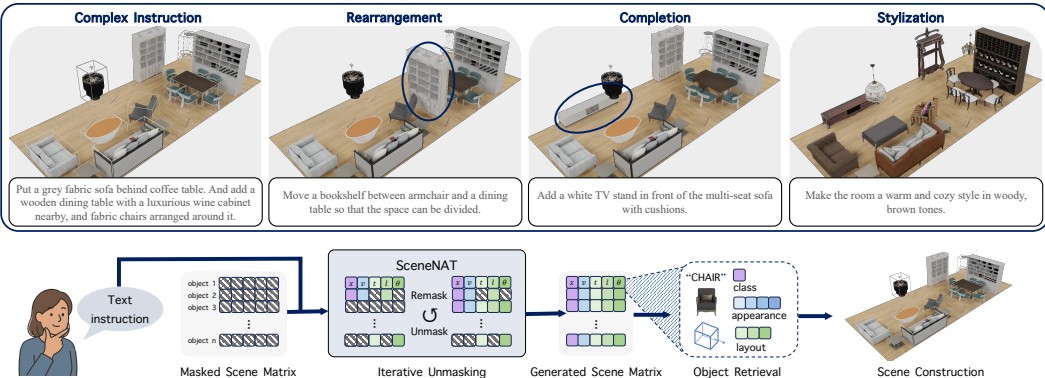

Figure 1: An illustration of the proposed method for text-driven 3D indoor scene generation. SceneNAT supports a wide range of applications, including complex instruction following, object rearrangement, recommendation, and stylization. Given a text instruction, our model iteratively predicts the attributes (class, appearance, layout) for a set of initially masked objects in a non-autoregressive manner. The resulting set of object attributes guides the retrieval and placement of 3D assets to construct the scene.

Although notable progress has been made, reflecting detailed user intent is further complicated when instructions contain *multiple* spatial relations, nested object dependencies, or detailed numerical constraints (Ye et al., 2024; Tam et al., 2025). Consequently, generated layouts frequently deviate from desired configuration, especially as instruction complexity increases.

In this paper, we introduce a novel Masked Indoor Scene Modeling (MISM) approach for instruction-conditioned 3D layout generation using a non-autoregressive transformer designed for parallel decoding. This modeling strategy progressively reconstructs complete 3D scenes from fully masked tokens through iterative refinement, enabling efficient inference and bidirectional context modeling. To address the challenges of complex instructions, we also introduce a dedicated triplet predictor module. This module uses a transformer decoder to predict a sparse set of symbolic constraints that explicitly describe the relations expressed in the user instruction. The constraints are formatted as (subject, predicate, object) triplets. This design provides robust and interpretable guidance, leading to more structured and accurate scene layouts.

Our contributions are as follows:

- We present SceneNAT, the first framework to leverage a non-autoregressive transformer with carefully designed masked modeling for language-guided 3D indoor scene synthesis. Our model achieves superior results while significantly reducing computational costs compared to diffusion-based approaches.
- We propose a novel approach that disentangles relational reasoning from scene generation using a dedicated triplet predictor. By explicitly modeling inter-object dependencies as a direct set prediction problem with bipartite matching, our approach overcomes a fundamental limitation of prior methods, enabling our model to achieve superior controllability and fidelity with complex language instructions.
- Extensive experiments on high-quality 3D scene benchmark demonstrate that SceneNAT achieves state-of-the-art performance, outperforming representative baselines in spatial accuracy, semantic alignment, and overall scene quality.

## 2 RELATED WORKS

**3D Scene Synthesis**  The synthesis of realistic 3D layouts has been a long-standing challenge. Early approaches utilized convolutional networks (Wang et al., 2018; 2019; Ritchie et al., 2019) and graph encoders (Li et al., 2019; Zhou et al., 2019), while later autoregressive Transformers (Wang et al., 2021; Paschalidou et al., 2021) sequentially generated scene elements. More recently, diffusion models (Huang et al., 2023b; Maillard et al., 2024; Lin & Mu, 2024; Zhai et al., 2023; Bai et al., 2025) have been introduced, which excel at generating diverse and physically plausible layouts but are computationally expensive due to their iterative sampling process. In parallel, recent works

have explored leveraging large language models (LLMs) (Feng et al., 2023; Yang et al., 2024b; Sun et al., 2024) to directly map text into structured 3D scene representations.

**Controllable 3D Scene Generation**  A critical aspect of scene generation is controllability, enabling users to specify objects, their arrangement, and style. Various control signals have been explored, including floor plans (Paschalidou et al., 2021; Wang et al., 2021), scene graphs (Zhou et al., 2019; Zhai et al., 2023), etc. Among these, natural language offers the most intuitive way to specify detailed configurations. However, its free-form nature makes accurate interpretation highly challenging, particularly for fine-grained object relations and spatial constraints. Most text-conditioned frameworks (Wang et al., 2021; Lin & Mu, 2024) rely on global text representations, which often fail to capture these details. LLM-based planners (Yang et al., 2024c; Çelen et al., 2024) can interpret high-level language, but their sequential querying incurs high latency and requires additional spatial optimization.

**Non-Autoregressive Transformers (NATs)**  Originally developed for machine translation (Ghazvininejad et al., 2019; Gu & Kong, 2020), NATs have since been adopted across various generation tasks such as image (Chang et al., 2022; Lezama et al., 2022; Chang et al., 2023; Li et al., 2023; Ni et al., 2024; Bai et al., 2024), video editing (Ma et al., 2024), and motion synthesis (Guo et al., 2024), where fast and scalable inference is essential. As an alternative to sequential and diffusion-based models, they enable fast and scalable inference by decoding all elements in parallel, while still supporting bidirectional context modeling with fewer steps. Despite their potential, NATs remain largely unexplored for 3D scene layout generation, where aligning object semantics with spatial structure is inherently difficult, and further complicated by the need to satisfy complex language instructions.

## 3 METHOD

### 3.1 PROBLEM DEFINITION AND SCENE REPRESENTATION

We formulate 3D scene synthesis from natural language as a masked modeling task over structured object-level attributes. Each scene $r_i$ is represented as a set of $n_i$ objects, where a scene consists of a maximum of $N$ objects. Each object is described by its category $x \in \mathcal{C}$, vector-quantized appearance tokens $v \in \mathbb{Z}^4$, and discretized layout parameters: 3D position $t \in \mathbb{Z}^3$, scale $l \in \mathbb{Z}^3$, and yaw rotation $\theta \in \mathbb{Z}$. We obtain the discrete appearance token $v$ from features extracted by a pretrained OpenShape encoder (Liu et al., 2023b) using the vector-quantized tokenizer from InstructScene (Lin & Mu, 2024). The layout variables are also uniformly discretized into fixed bins. Collectively, these attributes define a structured scene representation $r_i = \{x_j, v_j, t_j, l_j, \theta_j\}_{j=1}^{n_i}$, including both semantic and geometric information. Given a text instruction $I$, the objective is to model the conditional distribution $P(R|I)$ and generate a plausible scene consistent with the described configuration. Following common practice, the instruction is encoded using a pretrained CLIP text encoder (Radford et al., 2021).

### 3.2 SCENENAT FRAMEWORK

Our goal is to establish a MISM framework that enables controllable 3D layout generation conditioned on text instruction. To this end, we formulate the task as masked modeling. Given a partially masked scene representation $R_{\sim M}$, the model learns to infer the masked tokens $R_M$ by modeling the conditional distribution $P(R \mid R_{\sim M}, I)$, where $M$ denotes the set of masked positions. An overview of the pipeline is provided in Figure 2, and we detail its components in the following subsections.

**Object Attribute Embedding**  Each object is represented by discrete tokens for $x$, $v$, $t$, $l$, and $\theta$ (Sec. 3.1). Each attribute type has its own embedding table. These embedding features are aggregated across attributes to form a representative feature vector for each object. The embedding tables are reused in the final attribute prediction heads through weight tying (Press & Wolf, 2017; Inan et al., 2017), a standard practice that reduces parameters and ensures consistency between input and output tokens predictions.

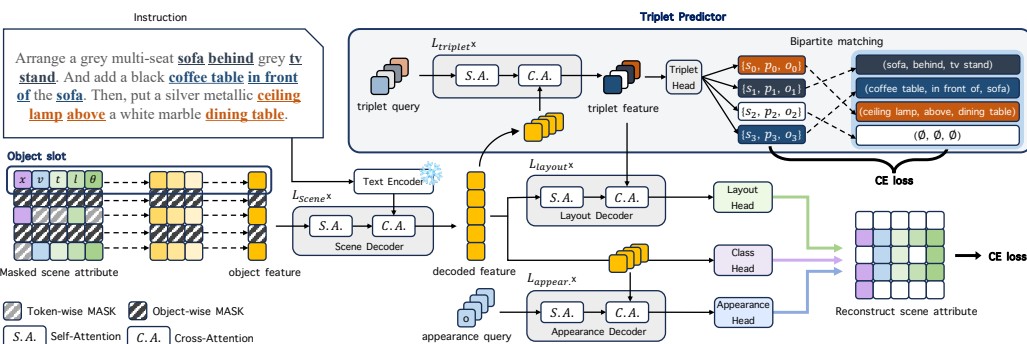

Figure 2: An overview of the SceneNAT framework. To achieve both visual quality and controllability, our model employs two specialized components. A scene decoder uses masked modeling to generate a globally coherent scene structure. Concurrently, a triplet predictor transforms a set of learnable triplet queries into specific relational constraints parsed from the text instruction. The intermediate triplet features are fused into the layout decoder via cross-attention layers to generate the final indoor scene attributes.

**Decoder Architecture** The decoder stage of SceneNAT operates on the object embeddings to produce the final scene layout. The scene decoder reconstructs masked object slots by leveraging contextual information from visible slots and guidance from the text instruction to capture the global structure. However, attribute decoding alone is often insufficient for grounding complex instructions that involve inter-object relations. To support such relation-aware modeling, our architecture separates relational reasoning from the primary attribute reconstruction task. For explicit relational guidance, we use ground-truth subject-predicate-object triplets that are parsed from the training instructions. In these triplets, the predicates from a predefined set $\mathcal{P}$ specify how objects are arranged. A dedicated triplet predictor is then trained to produce relational embeddings by attending to text and object features through learnable queries. These embeddings are supervised using the pre-extracted triplets, inducing the model to capture the specified symbolic constraints. We provide further details on the triplet predictor in Sec. 3.3.

**Attribute Prediction Heads** The decoded features are passed to attribute-specific heads to generate the complete scene. The class head determines the object's category, while the appearance head recovers fine-grained style. The layout head predicts translation, scale, and rotation by integrating both the contextual features from the scene decoder and the relational embeddings from the triplet predictor, ensuring the final object placement complies with the symbolic constraints expressed in the instruction. Together, these predictions complete the scene matrix, yielding a structured 3D layout that coherently reflects both semantics and geometry.

## 3.3 Relational Reasoning via Direct Set Prediction

To explicitly model the inter-object relationships dictated by the language instruction, we frame relational reasoning as a direct set prediction problem, inspired by DETR (Carion et al., 2020). Our triplet prediction module, illustrated in Figure 2, directly outputs a variable-sized set of symbolic triplets representing the scene's relational structure. This module uses a transformer decoder to transform a fixed set of $N_q$ learnable *triplet queries* into relation-aware features $F_{rel}$. Three independent feed-forward networks (FFNs) then map each output feature to logits for a subject, predicate, and object. Each head's output includes an additional class to represent a "no-object" or "no-relation" ($\emptyset$) class, allowing the model to handle scenes with fewer than $N_q$ relations.

Since the model predicts a set of triplets where the order is arbitrary, we employ a permutation-invariant set-based loss that finds an optimal bipartite matching between the ground-truth and predicted triplets. Let the ground-truth set be $Y = \{y_j\}_{j=1}^{N_t}$, where each $y_j = (s_j, p_j, o_j)$. Let the predicted set be $\hat{Y} = \{\hat{y}_k\}_{k=1}^{N_q}$, where each prediction $\hat{y}_k = (\hat{s}_k, \hat{p}_k, \hat{o}_k)$ is a bundle containing the logits for a subject, predicate, and object. Following DETR, we use the Hungarian algorithm to find

the optimal matching $\hat{\sigma}$ that minimizes the overall matching cost:

$$\hat{\sigma} = \arg\min_{\sigma \in \mathfrak{S}_{N_q}} \sum_{j=1}^{N_t} \mathcal{C}_{\text{match}}(y_j, \hat{y}_{\sigma(j)}) \tag{1}$$

The pair-wise matching cost $\mathcal{C}_{\text{match}}$ is defined as a linear combination of the negative log-probabilities for the ground-truth subject, predicate, and object classes. For instance, the predicate cost is $\mathcal{C}_p(p_j, \hat{p}_k) = -\mathbb{P}_{\hat{p}_k}(p_j)$, where $\mathbb{P}_{\hat{p}_k}$ is the probability distribution from the softmax of the predicate logits $\hat{p}_k$. Once the optimal matching $\hat{\sigma}$ is established, we define the target for each of the $N_q$ prediction queries to compute the triplet loss. The target for each prediction $\hat{y}_k$, denoted $y'_k$, is the matched ground-truth triplet $y_j$ if a match exists (i.e., $k = \hat{\sigma}(j)$), and the null triplet $(\emptyset, \emptyset, \emptyset)$ otherwise. The final triplet loss is a weighted sum of cross-entropy losses over all $N_q$ predictions against their corresponding targets, $y'_k = (s'_k, p'_k, o'_k)$. The loss is formulated as:

$$\mathcal{L}_{\text{triplet}} = \sum_{k=1}^{N_q} [\lambda_s \mathcal{L}_{\text{CE}}(s'_k, \hat{s}_k) + \lambda_p \mathcal{L}_{\text{CE}}(p'_k, \hat{p}_k) + \lambda_o \mathcal{L}_{\text{CE}}(o'_k, \hat{o}_k)] \tag{2}$$

To balance the training, we apply a lower weight to the loss for the $\emptyset$ class. This entire mechanism allows for end-to-end training without needing to impose a fixed order on the predicted triplets.

## 3.4 TRAINING AND INFERENCE

We optimize SceneNAT with a token-level reconstruction loss over masked scene attributes. For each masked token $i \in M$, the model is trained to maximize the likelihood of the ground-truth token $r_i$ given the unmasked context $R_{\sim M}$ and the instruction $I$. This reconstruction objective is a form of cross-entropy loss, applied separately to each attribute type. Formally, the reconstruction loss is a weighted sum of cross-entropy losses for each attribute, defined as:

$$\mathcal{L}_{\text{recon}} = \lambda_x \mathcal{L}_{\text{CE}}(x, \hat{x}) + \lambda_v \mathcal{L}_{\text{CE}}(v, \hat{v}) + \lambda_t \mathcal{L}_{\text{CE}}(t, \hat{t}) + \lambda_l \mathcal{L}_{\text{CE}}(l, \hat{l}) + \lambda_\theta \mathcal{L}_{\text{CE}}(\theta, \hat{\theta}) \tag{3}$$

where $x, v, t, l, \theta$ are the ground-truth attributes and $\hat{x}, \hat{v}, \hat{t}, \hat{l}, \hat{\theta}$ are the model's predictions. The $\lambda$ terms are hyperparameters used for attribute-specific loss weighting. This reconstruction loss is defined to be equivalent to minimizing the negative log-likelihood of the ground-truth tokens, and it can be expressed as:

$$\mathcal{L}_{\text{recon}} = \sum_{i \in M} -\log P(r_i \mid R_{\sim M}, I) \tag{4}$$

where $r_i$ is the ground-truth token at position $i$. The overall training objective combines this reconstruction loss with the triplet loss from Section 3.3 and is given by $\mathcal{L}_{\text{total}} = \mathcal{L}_{\text{recon}} + \lambda_{\text{triplet}} \mathcal{L}_{\text{triplet}}$, where $\lambda_{\text{triplet}}$ is a hyperparameter to balance two terms.

**Masking Strategy** Following prior works (Chang et al., 2022; 2023; Guo et al., 2024), we use a cosine-based dynamic masking schedule, where the masking ratio is sampled from $\gamma(\tau) = \cos\left(\frac{\pi\tau}{2}\right)$ with $\tau \sim \mathcal{U}(0, 1)$. To better capture the multi-modal nature of 3D scenes and disentangle local and global context, we stochastically split the total masking ratio between object-level and token-level masking. Specifically, we sample a proportion $p_{\text{obj}} \sim \mathcal{U}(0, 1)$ and assign $p_{\text{obj}} \cdot \gamma(\tau)$ of the masking to the object level, with the remainder assigned to the token level. This stochastic masking schedule encourages the model to handle various levels of corruption during training, from fully observed to nearly fully masked inputs, enabling joint learning of representation and generation (Li et al., 2023). Additionally, we apply a BERT-style (Devlin et al., 2019) replace-and-remask policy, simulating noise-injected refinement during inference.

**Sampling Strategy** For inference, our model starts from a fully masked scene representation and performs parallel decoding conditioned on the instruction $I$, progressively refining its predictions to reconstruct a complete scene layout. We adopt an iterative masked token prediction strategy inspired by MaskGIT (Chang et al., 2022), where uncertain tokens are progressively refined through parallel decoding. At each iteration, the model remasks and re-predicts tokens with low confidence, measured by the maximum softmax probability. This process is repeated for a fixed number of steps, with the remasking ratio at each step determined by a cosine schedule, consistent with the masking strategy used during training. This iterative refinement gradually improves the quality and consistency of the generated scene.

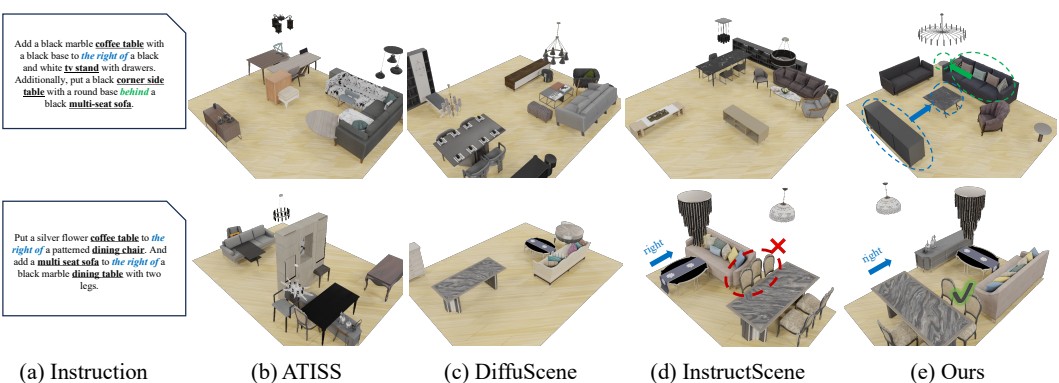

(a) Instruction     (b) ATISS     (c) DiffuScene     (d) InstructScene     (e) Ours

Figure 3: Comparison of qualitative results on language-guided scene generation. SceneNAT demonstrates superior performance in generating realistic 3D scenes that faithfully adhere to complex instructions. Additional qualitative results can be found in Appendix H.

## 4 EXPERIMENT

### 4.1 EXPERIMENTAL SETUP

**Dataset** We conduct all experiments on the extended 3D-FRONT dataset introduced in InstructScene (Chenguolin, 2024), which augments 3D-FRONT (Fu et al., 2021a) and 3D-FUTURE (Fu et al., 2021b) with instruction annotations. To improve instruction quality and coverage, we enhance the original templated instructions with more diverse sentence structures and increase the number of relations per instruction. We support up to four constraints to accommodate more complex guidance, whereas InstructScene (Lin & Mu, 2024) typically uses only one or two per instruction. We set this limit on the number of relations based on the maximum token length of the CLIP (Radford et al., 2021) text encoder. Instructions with more than four relations typically exceed this length, leading to a loss of information through truncation. Further details of the instruction refinement process are provided in the Appendix D.

**Baselines** We compare against three baseline models, including state-of-the-art methods that cover diverse architectural paradigms in 3D indoor scene generation. ATISS (Paschalidou et al., 2021) is an autoregressive transformer model that sequentially generates object attributes conditioned on previously generated outputs. DiffuScene (Tang et al., 2024) represents scene components as a fixed-size 2D matrix and generates the matrix through a continuous diffusion process. InstructScene (Lin & Mu, 2024) adopts a semantic scene graph with object relation edges as an intermediate representation and employs a *two-stage* cascaded diffusion model. To ensure fair comparison, we reimplemented all baselines following InstructScene and trained them from scratch using our instruction set. Implementation details for both baselines and our method are provided in the Appendix A.

### 4.2 METRICS

We evaluate generated scenes based on instruction adherence, visual quality, and physical plausibility. Instruction adherence is measured using iRecall (Lin & Mu, 2024), which reports the recall of spatial relation triplets specified in the instruction that are realized in the generated scene. Following prior works (Paschalidou et al., 2021; Liu et al., 2023a; Tang et al., 2024; Lin & Mu, 2024), we assess the visual plausibility of rendered scenes using three metrics. FID (Heusel et al., 2017) measures feature distribution differences, FID$^{\text{CLIP}}$ (Kynkäänniemi et al., 2022) assesses perceptual alignment using CLIP embeddings, and KID (Gruber & Buettner, 2023) provides an unbiased kernel-based similarity estimate. For all metrics, lower values indicate a closer alignment to the real scene distribution. Finally, to assess the physical validity of the generated layouts, we report the Total Intersection Volume ($\mathcal{V}_{\cap}^{\text{sum}}$)). This metric measures the global collision magnitude by aggregating the intersection volumes of oriented bounding boxes for all object pairs in the scene. Additional collision metrics and a more comprehensive evaluation are provided in Appendix A.7.

Table 1: Quantitative comparison on text-conditioned scene synthesis on three room types. We report the mean and standard deviation over 10 independent trials. For all metrics, lower is better except for iRecall. The best results are highlighted in bold, and the second-best are underlined.

| | | iRecall(%) (↑) | FID (↓) | FID$^{\text{CLIP}}$ (↓) | KID$_{\times 1e3}$ (↓) | $\mathcal{V}_{\cap}^{\text{sum}}$ (↓) |
|---|---|---|---|---|---|---|
| Bedroom | ATISS | 31.30 (1.76) | 128.50 (1.28) | 7.50 (0.14) | 3.59 (0.71) | 241.25 (20.09) |
| | DiffuScene | 45.98 (2.30) | 119.37 (1.53) | 6.71 (0.14) | 1.04 (0.82) | 93.26 (23.54) |
| | InstructScene | 66.72 (2.00) | 115.76 (1.71) | 6.50 (0.19) | -0.33 (0.49) | 79.62 (14.40) |
| | **Ours** | **70.45** (1.92) | **109.55** (1.36) | **6.19** (0.12) | **-1.18** (0.16) | **69.58** (12.00) |
| Living room | ATISS | 20.46 (1.41) | 134.71 (2.75) | 8.46 (0.22) | 52.26 (3.93) | 472.68 (16.08) |
| | DiffuScene | 27.39 (1.57) | 115.09 (1.62) | 5.64 (0.13) | 14.03 (2.11) | 249.61 (17.82) |
| | InstructScene | 47.97 (2.71) | 111.58 (1.07) | **5.31** (0.06) | 9.30 (0.95) | 204.91 (26.90) |
| | **Ours** | **50.01** (2.25) | **110.28** (1.18) | 5.49 (0.09) | **6.18** (1.11) | **151.24** (11.14) |
| Dining room | ATISS | 30.52 (3.01) | 157.60 (2.07) | 10.65 (0.31) | 61.31 (4.38) | 367.71 (23.71) |
| | DiffuScene | 36.68 (2.14) | 132.97 (0.70) | 7.93 (0.23) | 16.61 (1.53) | 240.46 (28.64) |
| | InstructScene | 46.54 (3.22) | 132.91 (1.37) | 7.64 (0.20) | 14.81 (1.87) | 187.95 (15.02) |
| | **Ours** | **56.29** (2.47) | **129.65** (1.68) | **7.51** (0.17) | 12.26 (0.99) | **169.31** (13.22) |

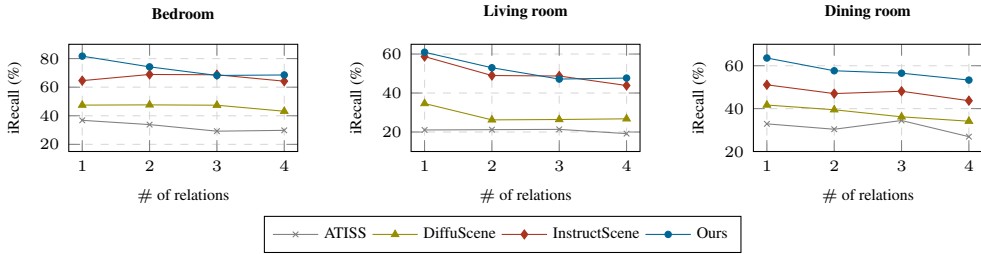

Figure 4: Comparison of iRecall with respect to the number of relational constraints in the instructions.

## 4.3 3D INDOOR SCENE SYNTHESIS FROM NATURAL LANGUAGE

Table 1 summarizes quantitative results across three room categories. We report the mean and standard deviation over 10 independent trials for all metrics. ATISS, which generates objects sequentially without explicit relation modeling, struggles to capture the multi-object dependencies specified in natural language, resulting in substantially lower iRecall. While DiffuScene benefits from diffusion-based global modeling, its holistic matrix formulation does not explicitly separate instructed from uninstructed relations. Although it produces visually plausible layouts, its controllability is limited, as evidenced by its low iRecall. InstructScene significantly advanced upon prior methods by leveraging a graph prior, achieving substantial gains in both controllability and scene quality.

SceneNAT consistently outperforms all baselines, achieving the highest iRecall by a clear margin and superior visual quality across nearly all metrics, as shown in Table 1. Moreover, SceneNAT achieves the lowest $\mathcal{V}_{\cap}^{\text{sum}}$ across all room types, notably outperforming DiffuScene which explicitly utilizes an IoU loss during training. This indicates that the global context modeling of SceneNAT effectively learns physical constraints directly from the data distribution without requiring explicit collision supervision. Figure 4 plots iRecall against the number of relational constraints, showing that our model consistently shows the best performance across all levels of instruction complexity. While the baselines ATISS and DiffuScene perform poorly in almost all cases, the comparison to In-

Table 2: Comparison of iRecall (%) on complex instructions with relational constraints ($N_{rel} = 5, 6$) exceeding the training setting ($N_{rel} = 4$). Results are reported for relational constraint counts of $N_{rel} = 4/5/6$, respectively.

| | **Ours** | InstructScene | DiffuScene | ATISS |
|---|---|---|---|---|
| Bedroom | **70.45 / 68.08 / 69.16** | 66.72 / 64.69 / 62.43 | 45.98 / 43.13 / 48.22 | 31.30 / 31.71 / 31.78 |
| Living room | **50.01 / 47.56 / 50.15** | 47.97 / 43.21 / 41.69 | 27.39 / 27.53 / 28.72 | 20.46 / 20.73 / 20.12 |
| Dining room | **56.29 / 53.08 / 53.64** | 46.54 / 46.17 / 44.94 | 36.68 / 37.01 / 33.70 | 30.52 / 27.66 / 31.33 |

Table 3: Comparison of computational costs. The reported TFLOPs are calculated by multiplying the TFLOPs per step by the number of sampling timesteps required for each model.

| Model | Inference time (s) | Params (M) | TFLOPs | FID | iRecall (%) |
|---|---|---|---|---|---|
| ATISS | 0.14 | 33.5 | 0.2 | 128.50 | 31.30 |
| DiffuScene | 33.28 | 63.4 | 63.5 | 119.37 | 45.98 |
| InstructScene | 6.73 | 87.7 | 44.3 | 115.76 | 66.72 |
| SceneNAT-S ($L_{\text{scene}} = 4$) | 1.02 | 53.1 | 7.9 | 110.91 | 67.06 |
| SceneNAT-B ($L_{\text{scene}} = 8$) | 1.35 | 69.9 | 10.3 | **109.55** | **70.45** |

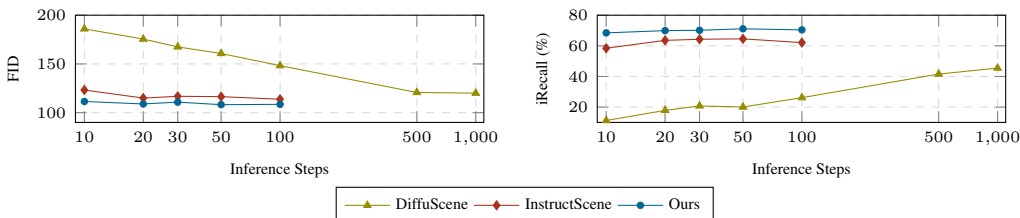

Figure 5: Comparison of FID and iRecall with respect to the number of inference steps.

structScene is particularly noteworthy. Despite InstructScene's use of a computationally heavy two-stage graph diffusion model, SceneNAT achieves superior results by incorporating a comparatively lightweight triplet predictor, highlighting the effectiveness of our approach for handling complex spatial relations.

We attribute this strong performance to our two core architectural contributions. Our masked modeling strategy learns the underlying distribution of realistic scenes to ensure both high visual quality and physical plausibility, and our triplet predictor design improves controllability by explicitly enforcing instructed relations. The qualitative comparisons are presented in Figure 3.

**Robustness to Unseen Relational Complexity**  To evaluate the capability of our model in handling highly complex scenarios, we conduct additional experiments using instructions containing 5 and 6 relational constraints which exceed the maximum of 4 constraints encountered during training. To strictly evaluate spatial reasoning without being hindered by the context limit of the text encoder, we preprocess these lengthy instructions by pruning non-essential descriptive words while preserving the core relational information.

As shown in Table 2, SceneNAT demonstrates remarkable robustness and maintains high iRecall across all room types even on these unseen complexity levels. Notably, while InstructScene exhibits clear performance degradation as complexity increases, our model consistently outperforms the baseline. This suggests that training the scene decoder alongside the auxiliary triplet predictor promotes a robust understanding of relational layouts and allows the model to generalize structural constraints beyond the strict capacity of the learnable queries.

4.4 COMPUTATIONAL EFFICIENCY

Table 3 summarizes the computational efficiency of SceneNAT against key baselines on the bedroom dataset. For clarity, SceneNAT-B is our full model, and SceneNAT-S is its smaller variant that uses only half of the scene decoder layers. For a fair comparison, we measure the inference time and TFLOPs at a batch size of 128, based on the average of 50 iterations following a warmup iteration. While ATISS is fast, its generative quality and instruction adherence are significantly weaker. In contrast, the superiority of SceneNAT is clearly evidenced in comparison with the powerful diffusion-based models. As illustrated in Figure 5, we achieve superior scene quality and instruction adherence in just 30 steps, whereas diffusion models such as DiffuScene and InstructScene require 1,000 and 110 (100+10) steps, respectively. This efficiency leads to a dramatic reduction in inference time, making our model approximately 5x faster than InstructScene and over 24.7x faster than DiffuScene. Furthermore, this efficiency is achieved with over 20% fewer parameters than InstructScene, requiring 4.3x lower TFLOPs than InstructScene and 6x lower than DiffuScene. Notably, SceneNAT-S also surpasses the performance of InstructScene with 6.6x faster inference

Table 4: Ablation studies on key components of the SceneNAT architecture.

| Configuration | FID | iRecall (%) | Configuration | FID | iRecall (%) |
|---|---|---|---|---|---|
| Full relation supervision | 110.76 | 65.16 | w/o object masking | 110.06 | 66.11 |
| w/o replace & remasking | 191.69 | - | w/o token masking | 111.20 | 68.26 |
| w/o triplet predictor | 110.91 | 62.77 | **Ours (Full)** | **109.55** | **70.45** |

speed and 5.6x lower TFLOPs. These findings highlight the effectiveness of our non-autoregressive transformer architecture for instruction-guided scene synthesis. We provide further analysis in Appendix B.

### 4.5 ZERO-SHOT DOWNSTREAM TASKS

Following InstructScene, we evaluate our model on zero-shot applications, including stylization, re-arrangement, completion, and unconditional generation. We also introduce a new task, layout-to-object, to specifically assess the bidirectional context modeling capabilities of our model. This task involves recommending the most suitable object $(x, v)$ for a given 3D bounding box $(t, l, \theta)$. For stylization, we report the $\Delta$ metric from InstructScene (Lin & Mu, 2024), which measures the difference in an object's feature alignment with a styled versus a generic class description, where a higher value indicates more successful stylization. The re-arrangement task assesses the model's spatial reasoning by requiring it to predict the correct layout for a set of objects whose positions have been perturbed, while their categories and sizes remain fixed. For the completion task, the model is given a partial scene and must generate the missing objects to form a coherent and complete layout. A quantitative comparison with the baselines is presented in Table 6.

On these zero-shot tasks, our model achieves superior or competitive performance with InstructScene, while significantly outperforming ATISS and DiffuScene across most metrics. Notably, our parallel decoding framework excels at bidirectional context modeling, overcoming the limitations of previous methods. The unidirectional flow inherent to the autoregressive architecture of ATISS and the two-stage design of InstructScene makes them incapable of performing the layout-to-object task. This leaves DiffuScene as the only baseline with this capability. We therefore compare our model against DiffuScene and demonstrate substantially superior performance, as shown in Table 6.

### 4.6 ABLATION STUDY

To validate the effectiveness of our model's key components, we conduct a series of ablation studies. We systematically remove or alter parts of our architecture and training strategy to observe the impact on performance. The results are summarized in Table 4.

**Triplet Predictor** The triplet predictor is fundamental to our model's ability to reason about relational structures. Removing the triplet predictor causes a significant decrease in iRecall. This highlights its critical role in translating high-level semantic constraints into a coherent and accurate spatial arrangement of objects. Further validation of our design choice is detailed in Appendix A.5.

**Full Relation Supervision** We also evaluate full relation supervision, a method similar to the entire scene graph approach used in InstructScene. Our findings suggest that it's more beneficial to supervise the triplet predictor primarily on the relations explicitly provided in the input, rather than

Table 5: Ablation study on spatial discretization granularity.

| | # of Bins | 32 | 64 | 128 | 256 |
|---|---|---|---|---|---|
| Bedroom | iRecall (%) | 70.17 | **70.45** | 69.93 | 68.02 |
| | FID | 110.28 | 109.55 | 109.69 | **108.56** |
| Living room | iRecall (%) | 49.19 | 50.01 | 50.80 | **52.60** |
| | FID | 109.30 | 110.28 | **108.98** | 110.41 |
| Dining room | iRecall (%) | 53.36 | 56.29 | 56.02 | **59.52** |
| | FID | 130.70 | 129.65 | **127.72** | 129.92 |

Table 6: Zero-shot generalization across downstream scene synthesis tasks. Higher is better for all metrics except FID, where lower is better. The best results are highlighted in bold, and the second-best are underlined.

| | | Stylization | Re-arrangement | | Completion | | Uncond. | Layout-to-Object |
|---|---|---|---|---|---|---|---|---|
| | | $\Delta_{\times 1e-3}$ | iRecall (%) | FID | iRecall (%) | FID | FID | FID |
| Bed | ATISS | 2.97 | 43.77 | 113.19 | 22.25 | 120.54 | **124.81** | – |
| | DiffuScene | 3.53 | 45.48 | 118.54 | 55.50 | 96.30 | 131.72 | 81.41 |
| | InstructScene | **7.03** | 75.55 | 100.95 | 64.06 | 94.36 | 125.47 | – |
| | **Ours** | 5.58 | **77.26** | **100.07** | **67.97** | **92.58** | 128.17 | **62.68** |
| Living | ATISS | 0.70 | 26.73 | 115.65 | 36.73 | 127.37 | 139.40 | – |
| | DiffuScene | 0.38 | 31.63 | 129.55 | 34.49 | **95.91** | 128.60 | 92.86 |
| | InstructScene | 0.71 | **58.16** | 110.56 | **44.49** | 96.36 | **117.59** | – |
| | **Ours** | **1.42** | 54.89 | **108.03** | 40.61 | 99.85 | 120.67 | **86.07** |
| Dining | ATISS | -0.31 | 34.89 | 141.59 | 47.33 | 147.80 | 166.23 | – |
| | DiffuScene | -0.15 | 41.56 | 142.52 | 45.78 | 120.88 | 147.73 | 107.35 |
| | InstructScene | **5.02** | 56.22 | 133.71 | **53.56** | **118.10** | **138.32** | – |
| | **Ours** | **5.02** | **62.00** | **124.34** | 46.00 | 121.91 | 139.95 | **98.64** |

on an entire scene graph. Our approach allows the scene decoder to more effectively manage the natural layout and implicit relations, leading to higher iRecall and more globally consistent scenes.

**Masking Strategy** The masking strategy is another cornerstone of our approach. The replace and remasking technique is vital for the stable and effective training of the non-autoregressive transformer, as configurations without it show a severe degradation in generation quality. Our approach further decomposes this strategy into token-level and object-level masking. While both strategies contribute to a better FID, object-level masking is more critical for improving iRecall. The full model utilizing both strategies achieves the best performance across both FID and iRecall. This result confirms that both levels of masking are helpful for synthesizing accurate and coherent scenes.

**Discretization Granularity** To analyze the impact of spatial discretization on generation quality, we conduct an ablation study by varying the number of bins for layout attributes $(t, l, \theta)$ from 32 to 256. As shown in Table 5, the results demonstrate that our model maintains robustness across different granularities. Regarding coarser discretization (32 bins), we observe a marginal reduction in iRecall while the overall visual quality remains consistent. Conversely, increasing the bin density to 128 or 256 provides marginal gains in both iRecall and FID in specific cases. Based on these findings, we set 64 bins as the default configuration since this setting offers an reasonable balance between generation quality and search space complexity.

## 5 CONCLUSION

In this work, we introduced SceneNAT, a masked non-autoregressive transformer for controllable 3D indoor scene generation. Our framework's core contribution is its masked indoor scene modeling (MISM) strategy, which is enhanced by a dedicated triplet predictor to effectively handle complex relational instructions. This design allows SceneNAT to consistently outperform leading diffusion-based methods, achieving superior controllability and visual quality at a substantially lower computational cost. We believe our work establishes masked generative modeling as a promising paradigm for scalable and interpretable 3D scene synthesis. For future work, our framework could be extended to support multi-modal inputs like images and sketches, and to scale to more complex scenes with a greater number of relations and large-scale environments.

ETHICS STATEMENT

We acknowledge that our framework, by utilizing pretrained models (e.g., CLIP (Radford et al., 2021), OpenShape (Liu et al., 2023b)), may inherit societal biases from broad and unfiltered web data. However, these potential risks are inherently limited. This is primarily due to our focus on the low-risk domain of indoor scenes, the use of public and non-sensitive training data, and a model architecture that prioritizes structural plausibility over unconstrained generation. In line with our commitment to research integrity, we will release our work under a restrictive license to prevent misuse and have no conflicts of interest to declare.

REPRODUCIBILITY STATEMENT

We provide comprehensive details in the Appendix to ensure the reproducibility of our work. We offer a full breakdown of our model's architecture, scene representation, masking strategy, and the complete training procedure, including all hyperparameters in Appendix A. Additionally, we describe our methodology for generating an enhanced instruction dataset with more complex relations in Appendix D and detail our re-implementation of baselines to ensure a fair comparison in Appendix A.3. The source code and the dataset will be released upon publication to facilitate verification and encourage future research.

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

## A  MORE DETAILS

### A.1  IMPLEMENTATION DETAILS

SceneNAT is based on the standard transformer architecture, utilizing the SiLU activation function (Ramachandran et al., 2017) in its feed-forward networks. For the main experiments, we set the common hyperparameters to an attention dimension of 512, 8 attention heads, and a dropout rate of 0.1. The decoding components include 2 triplet decoder layers, 2 appearance decoder layers, and 1 layout decoder layer. The two model variants are distinguished by the scene decoder depth: SceneNAT-B utilizes 8 scene decoder layers, compared to 4 layers for SceneNAT-S. We set the number of learnable triplet, $N_q$, to 4.

**Dataset**  We use the dataset filtered by InstructScene (Lin & Mu, 2024), including 4,041 bedrooms, 813 living rooms, and 900 dining rooms. For bedrooms, the number of objects per scene ranges from 3 to 12, across 21 distinct object categories. In contrast, for living and dining rooms, the object count per scene varies from 3 to 21, with a larger set of 24 object categories. More detailed information can be found in InstructScene paper.

**Scene Representation**  Object shape features are extracted using a pre-trained VQ-VAE (Van Den Oord et al., 2017) from the 3D-FRONT (Fu et al., 2021a) dataset, following (Lin & Mu, 2024). To prepare the data for our transformer-based model, we discretize continuous spatial attributes: object positions and sizes are quantized into 64 bins, while orientations are discretized into intervals of 10 degrees.

**VQ-VAE Training**  For VQ-VAE used in the scene representation above, frozen OpenShape encoder (PointBERT-ViT-G14) (Liu et al., 2023b) is adopted to extract 1280-dimensional semantic features from 3D point clouds. These features are processed by a Transformer-based encoder-decoder architecture with a vector-quantization bottleneck. A codebook of size 64 with 512-dimensional vectors encodes each object into a sequence of 4 ordered discrete tokens, following InstructScene. Training this VQ-VAE module requires approximately 10 hours on a single NVIDIA RTX A5000 GPU using the 4,000+ objects from the filtered 3D-FRONT dataset.

**Object Retrieval**  After iterative sampling, we perform a nearest-neighbor retrieval to select the best-matching 3D asset for each object to construct a scene. First, we filter the asset database (3D-FUTURE (Fu et al., 2021b)) to select candidate objects that match the class label predicted by our Class Head. Then, the sequence of 4 discrete appearance tokens generated by the Appearance Head is mapped back to their corresponding codebook vectors and decoded through the pre-trained VQ-VAE decoder to reconstruct a continuous 1280-dimensional semantic feature. Finally, we compute the cosine similarity between this reconstructed feature and the pre-computed OpenShape features of all candidate assets, and the asset with the highest similarity is retrieved and placed into the scene.

**Masking Strategy**  Our masking strategy is inspired by the methodologies of MaskGIT (Chang et al., 2022) and BERT (Devlin et al., 2019). Diverging from BERT's fixed 15% masking ratio, we employ a dynamic masking ratio determined by a cosine schedule, which varies the proportion of masked tokens at each training step. The masking is applied at both an object-level and a token-level. Initially, a portion of the total masking budget is used to mask entire objects. The remaining budget is then met by randomly masking individual tokens from the objects that were not previously masked. For each token selected for masking, the following procedure is applied: 10% are substituted with a random token from the vocabulary. Of the remaining 90%, 88% are replaced with a special [MASK] token, and the rest are left unchanged with their original values.

### A.2  TRAINING

**Setup**  We implement our model using the PyTorch (Paszke et al., 2019) framework. All experiments were conducted on a single NVIDIA RTX A5000 GPU.

**Optimization** We use the AdamW (Loshchilov & Hutter, 2017) optimizer with a learning rate of 1e-4 and a weight decay of 0.02. We employ a cosine annealing scheduler with a linear warmup for the initial 50 epochs, after which the learning rate is decayed to a minimum of 1e-5. All models are trained for 2000 epochs with a batch size of 256. To ensure stability, we apply gradient clipping with a maximum norm of 1.0. The coefficients for the triplet loss components ($\lambda_s$, $\lambda_p$, and $\lambda_o$), as well as the reconstruction loss components ($\lambda_x$, $\lambda_v$, $\lambda_t$, $\lambda_l$, and $\lambda_\theta$), are all set to 1.0. We found that a triplet loss weight of $\lambda_{\text{triplet}} = 1.0$ yielded the best performance on our validation set after exploring values of 0.2, 0.5, 1.0, and 2.0. Also, we set the weight for null triplet to 0.1. After training, we select the model checkpoint with the best performance on our validation set for all evaluations.

**Classifier-Free Guidance (CFG)** To improve instruction fidelity without sacrificing diversity, we incorporate classifier-free guidance (Ho & Salimans, 2022) into training phase. During training, we randomly remove the conditioning input by nulling out CLIP (Radford et al., 2021) embedding of instruction $I$ with probability $p = 0.1$, enabling the model to learn both conditional and unconditional generation. At inference, guidance is applied by linearly combining conditional and unconditional logits $\omega_c, \omega_u$ before softmax:

$$\omega = (1 + \rho) \cdot \omega_c - \rho \cdot \omega_u, \tag{5}$$

where $\rho$ denotes the guidance scale.

### A.3 BASELINE IMPLEMENTATION

To ensure a fair comparison under our redefined problem setup, we reimplement and retrain all baselines from scratch using our refined dataset annotations (see Appendix D). For our work, we adapt three key baselines, ATISS (Paschalidou et al., 2021), DiffuScene (Tang et al., 2024), and InstructScene (Lin & Mu, 2024). We modify both ATISS and DiffuScene to predict discrete VQ-based appearance features and condition on CLIP (Radford et al., 2021) text embeddings. For ATISS, we replace the original global context token (a room mask feature) with a CLIP instruction embedding, while DiffuScene utilizes the text-based cross-attention from its original implementation. We also retrain InstructScene on our refined dataset using its original configuration, as its architecture does not require modifications. For training, we follow the optimization procedures, training steps, and hyperparameter configurations from the respective papers. We evaluate all baselines under the same experimental settings to ensure a fair comparison with our method.

### A.4 SPATIAL RELATION DEFINITIONS

We adopt the set of 11 geometric relationship definitions from InstructScene (Lin & Mu, 2024) to describe the spatial arrangement of objects. These definitions are based on the properties of object bounding boxes, assuming the XY-plane represents the ground and Z is the vertical axis. Let $s$ denote a "subject" and $o$ an "object" in a relationship. The definitions rely on three core metrics:

- Relative Orientation ($\theta_{so}$): The angle between the subject and object on the ground plane, computed as $\theta_{so} = \text{atan2}(Y_s - Y_o, X_s - X_o)$.

- Ground Distance ($d(s, o)$): The Euclidean distance between the centers of $s$ and $o$ projected onto the XY-plane.

- Overlap Predicate (inside$(s, o)$): A boolean function that is true if the subject's center lies within the 2D projection of the object.

Based on these metrics, the relationships are determined by the following rules:

**Horizontal Relationships** Defined by ground distance and relative orientation, categorized into two ranges. For objects at a medium distance ($1 < d(s, o) \leq 3$), the four primary directional relationships are defined as:

- Right of: $-\frac{\pi}{4} \leq \theta_{so} < \frac{\pi}{4}$.

- In front of: $\frac{\pi}{4} \leq \theta_{so} < \frac{3\pi}{4}$.

- Left of: $\theta_{so} \geq \frac{3\pi}{4}$ or $\theta_{so} < -\frac{3\pi}{4}$.

Table 7: Ablation studies on the architectural design of our model, comparing the number of transformer layers in our triplet predictor and the use of relational attention without bipartite matching. Performance is evaluated using Fréchet Inception Distance (FID) and instruction Recall (iRecall).

| # of layers | FID $\downarrow$ | iRecall $\uparrow$ |
|---|---|---|
| w/ relational attention | 110.05 | 65.04 |
| w/o triplet predictor | 110.91 | 62.77 |
| 1 | 110.06 | 65.87 |
| 2 | **109.55** | **70.45** |
| 3 | 110.48 | 68.50 |
| 4 | 109.80 | 67.54 |

- Behind: $-\frac{3\pi}{4} \leq \theta_{so} < -\frac{\pi}{4}$.

For objects in close proximity ($d(s,o) \leq 1$), four corresponding 'closely' variants (e.g., closely right of) apply. These use the exact same angular constraints as their primary counterparts.

**Vertical Relationships**

- Above: A subject $s$ is above an object $o$ if their vertical separation is greater than half their combined heights: $(Center_{Z_s} - Center_{Z_o}) > (Height_s + Height_o)/2$ and Inside$(s,o) \vee$ Inside$(o,s)$.

- Below: A subject $s$ is below an object $o$ if the inverse condition is met: $(Center_{Z_o} - Center_{Z_s}) > (Height_s + Height_o)/2$ and Inside$(s,o) \vee$ Inside$(o,s)$.

**Null Relationship**

- None: No direct spatial relationship exists if the ground distance is greater than 3 units: $d(s,o) > 3$.

A.5 DESIGN CHOICE FOR RELATION MODELING

To further validate our design choices for relation modeling, we conduct two analyses.

**Ablation on Triplet Predictor Layers**   We report an ablation study on the performance impact of the number of layers in our triplet predictor. As shown in Table 7, all configurations with the triplet predictor outperform the baseline without it, demonstrating its overall effectiveness. The best performance is achieved when using two transformer layers in terms of both FID and iRecall.

**Comparison with Relational Attention (RA)**   We present a comparative experiment to demonstrate that learning separate subject, predicate, and object embeddings via triplet-based bipartite matching is more effective than using a combined triplet embedding. To provide a fair comparison, we replace our triplet predictor, which is built on basic transformer blocks, with a specialized transformer architecture that uses relational attention (RA), as proposed in Altabaa et al. (2023); Altabaa & Lafferty (2024). The RA is a module specifically designed to learn relational features disentangled from sensory information.

The RA's operation is based on two key components, slot attention and relational attention. First, slot attention maps each object's feature vector $z$ obtained from the scene decoder to a set of learned slots that represent its semantic role. This is defined by the following equation:

$$SlotAttention(z) = \text{concat}\left(s^{(1)}, \ldots, s^{(n_h)}\right), \text{ where } \quad s^i = Softmax\left((zW_q^i)(F_{\text{lib}}^i)^\top\right) S_{\text{lib}}^i. \quad (6)$$

Here, $F_{\text{lib}}$ and $S_{\text{lib}}$ each denotes a set of learnable feature templates and slot vectors. The resulting slot features $s$ are used in a relational attention head to compute interactions between objects,

passing messages along fixed relational channels. Formally, this process is defined by the following equation:

$$RelationalAttention(z) = \sum_{i=1}^{n} Softmax\left((zW_q^i)(zW_k^i)^\top\right)(r_i W_r + s_i W_s) \qquad (7)$$

This dual-attention approach enables explicitly encode relational information between objects, effectively disentangling it from object-specific priors.

Unlike our triplet predictor, which is trained to decode explicit symbolic constraints and learn separate embeddings for the subject, predicate, and object via triplet-based bipartite matching, the transformer with RA is trained to predict a single, combined triplet embedding. The output of the trained transformer then serves as a conditioning signal for the layout decoder. As shown in Table 7, the triplet predictor achieves superior performance on iRecall, validating our design choice.

### A.6 ABLATION STUDY ON CLASSIFIER-FREE GUIDANCE SCALES

To analyze the impact of the classifier-free guidance (CFG) scale on the trade-off between instruction fidelity and generation quality, we give an ablation study by varying the scale $\rho$ from 0.0 to 10.0. The results are summarized in Table 8.

We observe a distinct trade-off across the guidance scales. While unconditional generation ($\rho = 0$) expectedly yields poor instruction adherence, increasing the scale up to 2.0 reveals a clear trade-off where iRecall improves at the cost of a slight degradation in visual quality. However, increasing the scale beyond 2.0 leads to performance drops in both metrics. Based on these results, we set $\rho = 1.0$ as our default setting to maintain the balance between high-quality generation and faithful instruction following.

Table 8: Ablation study on classifier-free guidance scales ($\rho$). The best results for each metric are highlighted in **bold**.

| Guidance Scale ($\rho$) | 0.0 | 0.5 | 1.0 | 1.5 | 2.0 | 3.0 | 4.0 | 5.0 | 10.0 |
|---|---|---|---|---|---|---|---|---|---|
| iRecall ($\uparrow$) | 11.98 | 66.26 | 70.45 | 71.39 | **72.62** | 68.22 | 63.81 | 66.99 | 43.52 |
| FID ($\downarrow$) | 128.17 | 111.61 | **109.55** | 110.10 | 111.49 | 113.25 | 116.99 | 120.15 | 133.45 |

### A.7 QUANTITATIVE ANALYSIS ON PHYSICAL PLAUSIBILITY

To evaluate the physical plausibility of the generated scenes, we conduct an additional quantitative evaluation focusing on object collisions. To ensure precise detection, intersections were computed using oriented bounding boxes. We employed three specific metrics to capture different aspects of physical plausibility.

**Total Intersection Volume ($\mathcal{V}_\cap^{\text{sum}}$)** This represents the aggregate sum of intersection volumes for all object pairs in the scene. It quantifies the absolute magnitude of physical interference, serving as a global measure of layout validity.

**Average Intersection Volume ($\mathcal{V}_\cap^{\text{avg}}$)** This metric calculates the mean intersection volume across all colliding object pairs. It serves as an indicator of the average collision density within a generated scene.

**Intersection-over-Minimum (IoMin)** Defined as the average ratio of the intersection volume to the volume of the smaller object in a pair ($\frac{V_\cap}{\min(V_A, V_B)}$), this metric captures the *degree* of collisions. Unlike standard IoU, IoMin effectively handles size imbalance, such as detecting when a small object is completely submerged within a larger one.

The quantitative comparisons presented in Table 9 indicate that while baselines such as DiffuScene and InstructScene achieve competitive performance in certain metrics, SceneNAT consistently demonstrates a strong capability in minimizing overall physical violations. Most notably, our model

Table 9: Quantitative comparison of physical plausibility. We report the average intersection volume ($\mathcal{V}_\cap^{\text{avg}}$), total intersection volume ($\mathcal{V}_\cap^{\text{sum}}$), and IoMin Score. **Bold** indicates the best result, and underline indicates the second best. Lower values are better ($\downarrow$) for all metrics.

| Room Type | Method | $\mathcal{V}_\cap^{\text{sum}}$ ($\downarrow$) | $\mathcal{V}_\cap^{\text{avg}}$ ($\downarrow$) | IoMin ($\downarrow$) |
|---|---|---|---|---|
| Bedroom | ATISS | 241.25 | 0.041 | 0.384 |
| | DiffuScene | 93.26 | **0.012** | **0.227** |
| | InstructScene | 79.62 | 0.018 | 0.242 |
| | **Ours** | **69.58** | 0.017 | 0.257 |
| Living Room | ATISS | 472.68 | 0.008 | 0.193 |
| | DiffuScene | 249.61 | 0.007 | 0.158 |
| | InstructScene | 204.91 | **0.006** | **0.152** |
| | **Ours** | **151.24** | 0.007 | 0.153 |
| Dining Room | ATISS | 367.71 | 0.011 | 0.259 |
| | DiffuScene | 240.46 | 0.010 | 0.218 |
| | InstructScene | 187.95 | 0.010 | 0.222 |
| | **Ours** | **169.31** | **0.008** | **0.207** |

achieves the lowest $\mathcal{V}_\cap^{\text{sum}}$ across all three room types. This suggests that the global context modeling of SceneNAT is particularly effective at reducing the aggregate magnitude of inter-object collisions within a scene. It is noteworthy that SceneNAT achieves these results without explicit collision supervision, whereas DiffuScene incorporates an IoU loss during training. This demonstrates that our masked generative approach effectively learns to adhere to physical constraints directly from the data distribution.

## A.8 ADDITIONAL COMPARISON WITH CONCURRENT STATE-OF-THE-ART

To benchmark against the most recent advancements, we provide an additional quantitative comparison with *FreeScene* (Bai et al., 2025), a concurrent state-of-the-art method in Table 10. Since the official code is unavailable, we directly reference the metrics reported in the original paper. To ensure a fair comparison, we exclude their VLM-enhanced "Graph Designer" variant since it utilizes a VLM to infer objects and relations that closely approximate the ground truth. Additionally, we evaluate SceneNAT's iRecall on the subset of instructions with 1 or 2 relational constraints ($N_{rel} \leq 2$) to match the complexity targeted by baselines, while visual quality metrics are averaged over the entire test set. As shown in the table, SceneNAT still demonstrates superior or highly competitive performance across most metrics. Furthermore, our discretization ablation study reveals that increasing the bin resolution to 128 further improves visual quality. This achieves an FID of 108.98 in the living room and 127.72 in the dining room, effectively matching or surpassing FreeScene's FID across all room types.

## B COMPUTATIONAL EFFICIENCY AND SCALABILITY

Figure 6 reports the inference latency of SceneNAT-S, SceneNAT-B, and the InstructScene (Lin & Mu, 2024) across varying batch sizes. Our models are significantly faster even at minimal throughput. For instance, SceneNAT-S (0.49s) is approximately $5\times$ faster than InstructScene (2.46s) at batch size 1. Furthermore, this advantage becomes more pronounced under high-throughput conditions, where models process multiple scenes in parallel. As the batch size scales to 128, InstructScene's latency surges to 6.73s. In contrast, SceneNAT-B maintains efficiency, reaching only 1.35s. These results clearly demonstrate the superior efficiency and scalability of our non-autoregressive parallel decoding mechanism, which is essential for deploying large-scale scene generation in real-world, high-demand applications.

To further evaluate the stability of our framework, we measured the peak memory consumption and runtime with respect to batch size and object count limits. Table 11 reports these values for bedroom ($N = 12$) and living & dining ($N = 21$) scenes, respectively. We emphasize that due to our fixed-size parallel decoding architecture, the computational cost is determined by the maximum object capacity $N$, making the inference cost invariant to the actual number of generated objects in a given scene.

Table 10: Quantitative comparison with concurrent state-of-the-art. The results for ATISS, DiffuScene, InstructScene, and FreeScene are sourced directly from (Bai et al., 2025). SceneNAT's iRecall is evaluated on instructions with up to 2 relational constraints ($N_{rel} \leq 2$) to match the baseline settings. Best results are in **bold**, and second best are underlined.

|  | Method | iRecall ($\uparrow$) | FID ($\downarrow$) | FID$_{\text{CLIP}}$ ($\downarrow$) | KID$_{\times 10^3}$ ($\downarrow$) |
|---|---|---|---|---|---|
| **Bedroom** | ATISS | 44.06 | 122.37 | 8.23 | 0.74 |
|  | DiffuScene | 53.32 | 129.34 | 9.66 | 0.81 |
|  | InstructScene | 72.71 | 114.86 | 6.52 | 0.68 |
|  | FreeScene | 73.69 | 111.21 | 6.43 | 0.35 |
|  | **Ours** | **77.96** | **109.55** | **6.19** | **-1.18** |
| **Living room** | ATISS | 36.31 | 120.10 | 7.43 | 16.44 |
|  | DiffuScene | 39.58 | 135.93 | 10.71 | 29.07 |
|  | InstructScene | 57.21 | 111.52 | 5.91 | 8.65 |
|  | FreeScene | 58.16 | 110.55 | 5.83 | 7.95 |
|  | **Ours** | **59.56** | **110.28** | **5.49** | **6.18** |
| **Dining room** | ATISS | 32.56 | 134.75 | 9.82 | 24.25 |
|  | DiffuScene | 37.17 | 142.37 | 11.76 | 28.36 |
|  | InstructScene | 61.47 | 129.13 | 8.24 | 15.27 |
|  | FreeScene | **63.39** | **127.28** | 8.01 | 14.83 |
|  | **Ours** | 61.89 | 129.65 | **7.51** | **12.26** |

Memory usage is dominated by fixed model parameters, resulting in a highly stable sub-linear scaling curve. Increasing the batch size by $128\times$ results in roughly a $2\times$ increase in peak memory. Regarding object count, the difference is negligible at lower batch sizes. Even at a batch size of 128, denser scenes ($N = 21$) show only an approximate 25% increase in memory usage compared to sparser scenes ($N = 12$). The peak memory reaches approximately 1.2 GB, ensuring decoding stability on standard hardware. At a batch size of 128, processing living and dining rooms ($N = 21$) costs approximately 35% more time compared to bedrooms ($N = 12$), primarily due to the increased attention complexity associated with a larger $N$. However, the absolute difference is marginal (less than 0.5s), confirming that the model maintains high throughput across different scene complexities.

Table 11: Peak memory consumption and mean runtime with respect to batch size and maximum object count ($N$). Values are reported as bedroom ($N = 12$) / living & dining ($N = 21$).

| Batch Size | Object Count ($N$) | Peak Memory (GB) | Mean Runtime (s) |
|---|---|---|---|
| 1 | 12 / 21 | 0.5945 / 0.5963 | 0.6223 / 0.6809 |
| 8 | 12 / 21 | 0.6144 / 0.6300 | 0.6436 / 0.7092 |
| 32 | 12 / 21 | 0.6821 / 0.7446 | 0.6543 / 0.7347 |
| 128 | 12 / 21 | 0.9530 / 1.1916 | 1.3459 / 1.8145 |

## C  SAMPLING PROCESS VISUALIZATION

To illustrate the iterative refinement procedure described in Sec. 3.4, we first show the mask state at step 0 (Figure 7), where all elements are masked and thus appear dark, while colored entries denote fixed tokens. Figure 9 then visualizes the 50-step sampling process, with intermediate results shown at every 10th step. Under the iterative remasking schedule, a subset of elements is masked again at each stage and subsequently recovered through further unmasking, leading to progressively improved scene quality and coherence.

## D  INSTRUCTION PROMPTING

We adapt the dataset introduced by InstructScene, which augments 3D-FRONT (Fu et al., 2021a) and 3D-FUTURE (Fu et al., 2021b) with relation annotations, object captions, and filtered natural descriptions. While we adopt the same preparation pipeline, we refine the instruction generation stage to support richer multi-relation prompts. Instruction generation converts annotated subject–predicate–object relations into natural-language prompts by randomly sampling a subset of relations and filling them into textual templates. InstructScene limits each prompt to at most

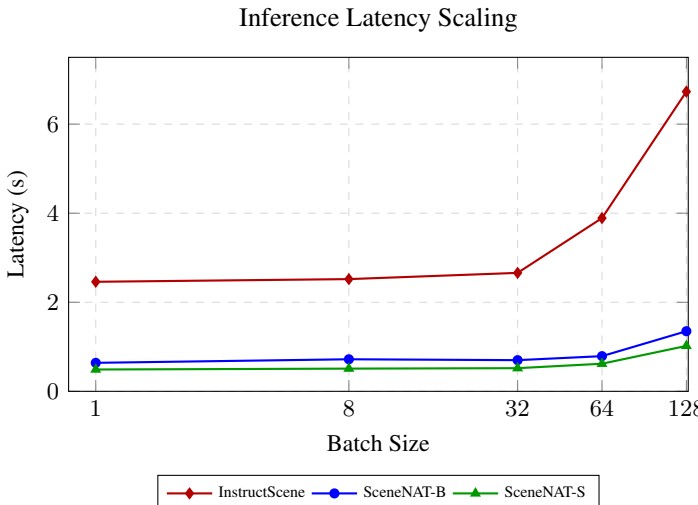

Figure 6: Inference Latency Scaling. Latency (in seconds) is plotted against batch size.

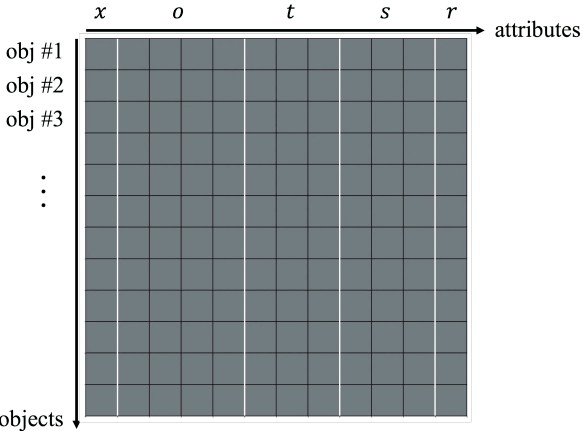

Figure 7: Mask state at step 0. The scene matrix structure is shown for clarity, with each row corresponding to an object and each column to an attribute. At this initial state, all elements are masked and appear dark.

two relations, whereas in our setup we extend this to a maximum of four to reflect more complex guidance. To further refine the generated instructions, we incorporate (i) filtering of redundant symmetric pairs, (ii) discourse-aware ordering so that successive relations preferentially reuse previously mentioned objects, and (iii) regulation of referring expressions (e.g., "the" vs. "another"). These changes yield instructions that are longer, more coherent, and better aligned with the multi-relation setting of our experiments, while preserving compatibility with the relational annotations provided by InstructScene.

## E  LIMITATIONS AND FUTURE WORK

While our framework demonstrates strong performance, we identify several limitations for future research. First, the scalability of our triplet predictor to a larger number of simultaneous relational constraints has yet to be explored. Second, our model relies on a uniform discretization for layout variables $(t, l, \theta)$, and the optimality of this fixed quantization for representing continuous 3D space is not strictly guaranteed. Third, the framework involves a number of hyperparameters governing both the training and inference process (e.g., remasking ratios, guidance scale), which require careful tuning. Finally, SceneNAT is trained and evaluated exclusively on synthetic assets and relies on a

closed retrieval library. We note that this reliance on synthetic data is a common constraint shared by most recent state-of-the-art baselines in this domain, limiting direct applicability to real-world environments.

These limitations also highlight promising directions. To address hyperparameter sensitivity, one could explore alternating optimization techniques, proposed in AutoNAT (Ni et al., 2024), for automatically tuning key hyperparameters such as the remasking ratio and guidance scale. Furthermore, a significant extension would be to move beyond the current retrieval-based approach. By integrating a generative mesh decoder, our framework could be enhanced to synthesize novel object geometries from scratch conditioned on text instruction, rather than relying solely on a predefined library of shapes. In addition, incorporating visual or structural cues from real environments such as partial scans, images, or coarse floorplans as additional conditioning signals could open up a path toward more grounded and real-world-aware scene generation.

## F    LLM USAGE

We used a large language model (LLM) to assist with minor editorial tasks. The use of the LLM was strictly confined to correcting grammatical errors, fixing typos, and rephrasing sentences for improved clarity. The LLM was not used for generating scientific ideas, experimental results, or any core content of this paper. The authors take full responsibility for all claims and the final content.

## G    USER STUDY

We conducted a user study involving 42 participants to assess the perceptual quality and instruction adherence of the generated scenes. We presented 18 sets of qualitative comparisons where each set displayed the text instruction alongside scenes generated by SceneNAT and three baselines including ATISS, DiffuScene, and InstructScene.

Participants were asked to rank the 1st and 2nd results based on a holistic evaluation of two criteria: (1) *Instruction Alignment* which measures how faithfully the scene adheres to the spatial and semantic constraints described in the text, and (2) *Overall Quality* which assesses the visual plausibility and aesthetic coherence of the layout. The results are summarized in Figure 8. SceneNAT was selected as the most preferred method in 54.8% of the cases which outperforms the strongest baseline, InstructScene (38.1%).

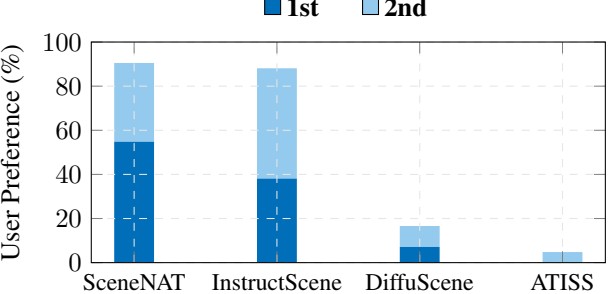

Figure 8: The User study across different methods. The stacked bars indicate the percentage of times each model was selected as the 1st and 2nd choice by 42 human evaluators with 18 comparison sets. SceneNAT shows a distinct advantage in capturing user intent and generation quality.

## H    ADDITIONAL RESULTS

In this section, we present various qualitative results. Figure 10 reports various visualizations of text-conditioned scene generation. From Figure 11 to Figure 14, we report visualization results for instructions with one to four relational constraints, respectively. Additionally, from Figure 15 to Figure 19, we report qualitative results for downstream tasks including stylization, re-arrangement,

completion, layout-to-object and unconditional generation. Figure 20 visualizes the system's behavior when encountering physically implausible or contradictory instructions, demonstrating the prioritization of learned physical priors. Figure 21 further demonstrates the model's generalization capacity with open vocabulary instructions, including unseen spatial predicates and descriptive object attributes.

Finally, Figure 22 presents an qualitative results of the model's performance on diverse "in-the-wild" instructions. We observe strong generalization in handling functional roles and aesthetic descriptions by mapping abstract concepts to concrete objects via the semantic embedding space. However, the model exhibits variable capability with negative constraints and high-level spatial reasoning. We attribute the inconsistent adherence to negative constraints to the dominance of positive samples in the training data which causes the model to overlook negation tokens, while the challenge with high-level spatial reasoning stems from the absence of explicit relational triggers required to ground abstract spatial logic into geometric constraints.

## I    FAILURE CASES

We present a qualitative analysis of typical failure modes observed in our framework. Figure 23 illustrates instances of relation violations, where the generated layout fails to strictly satisfy the spatial constraints specified in the instruction. Figure 24 demonstrates cases of incomplete adherence to visual descriptions, where the model overlooks specific object attributes such as color or style. Finally, Figure 25 showcases examples of compromised layout plausibility, exhibiting physical inconsistencies such as object collisions or geometric misalignments.

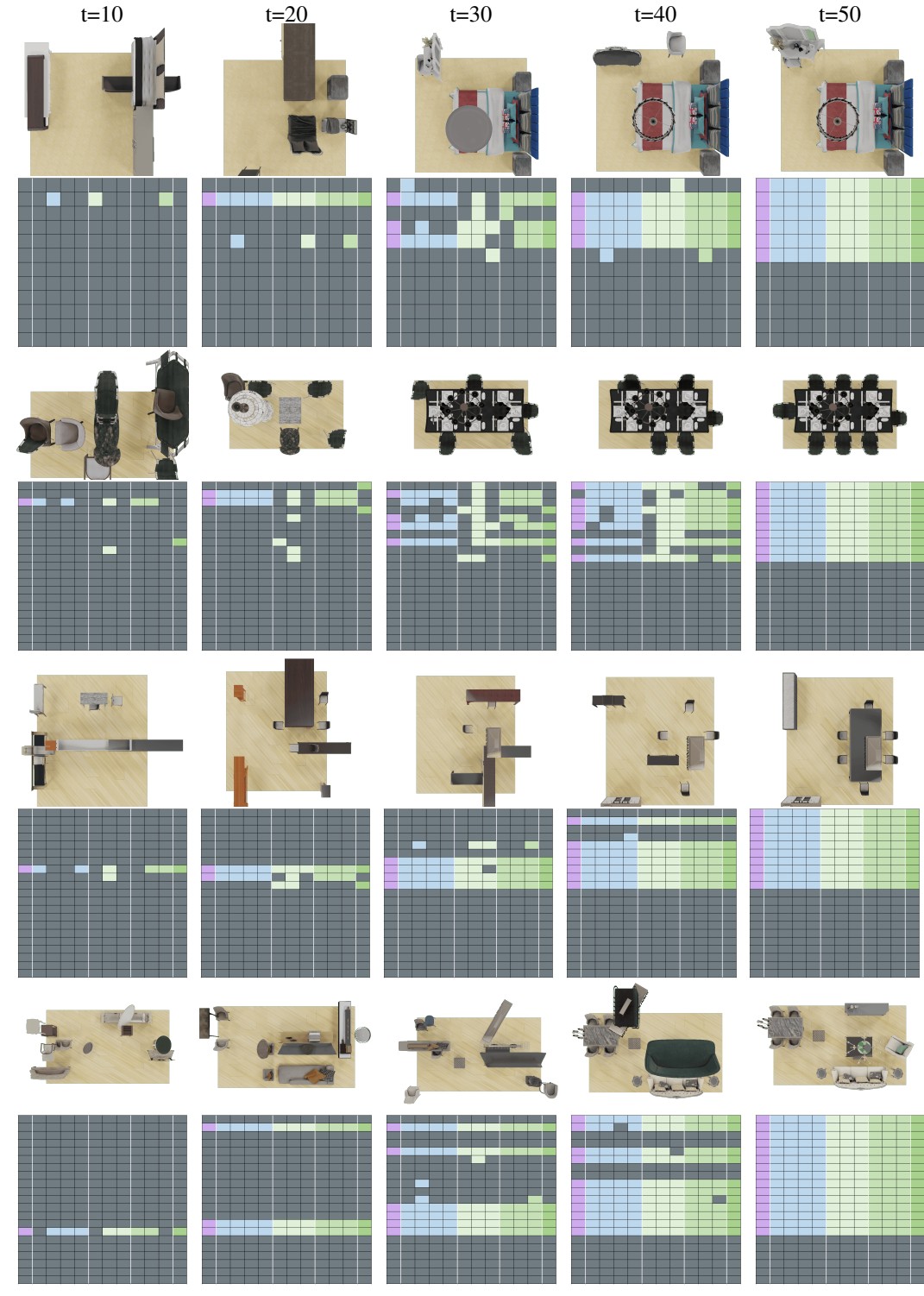

Figure 9: Visualization of a 50-step generation process, which progresses from left to right. Intermediate results are shown at every 10th step, with each synthesized scene displayed alongside a mask that highlights the fixed elements in color.

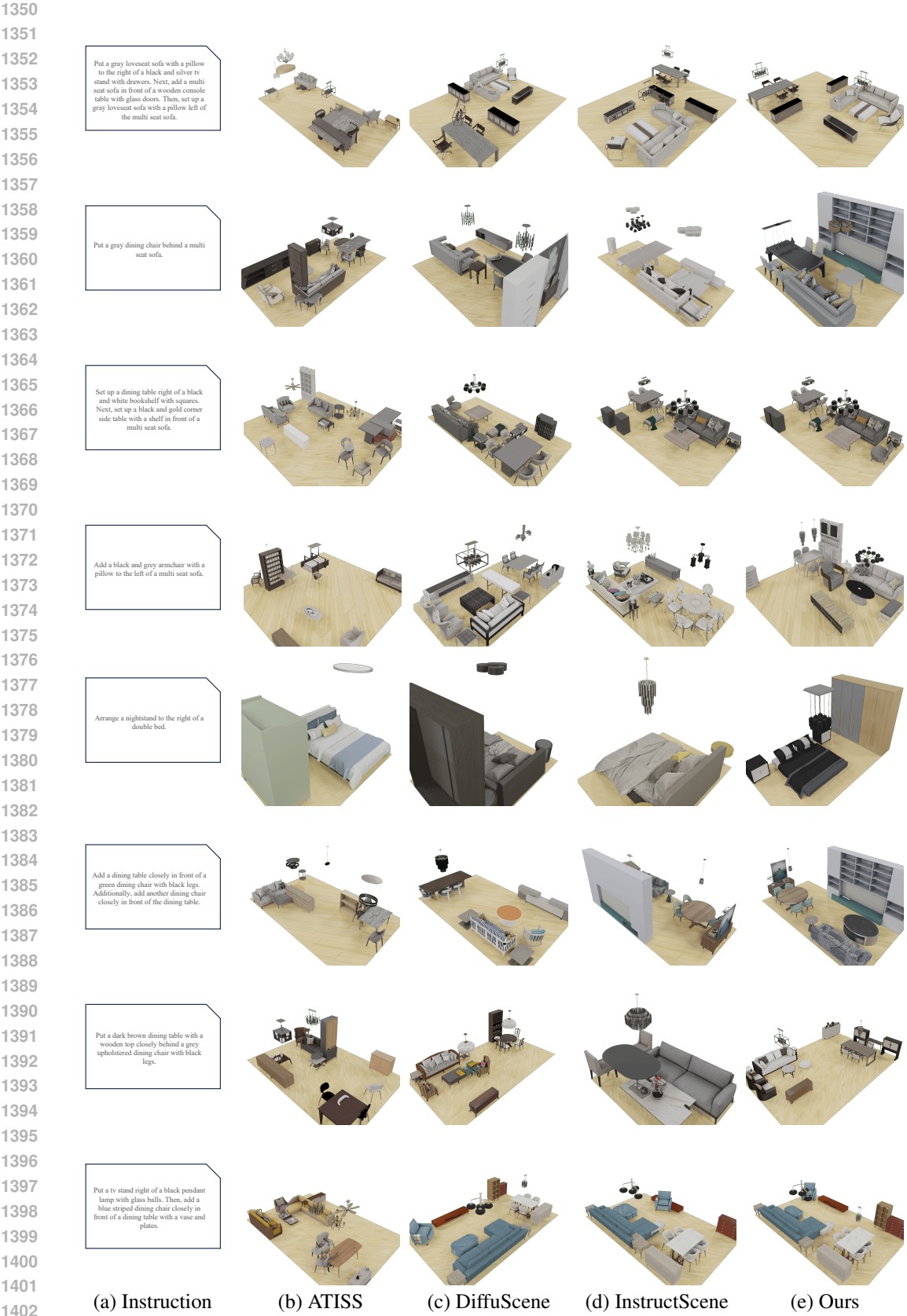

Figure 10: Comparison of qualitative results for text-to-scene generation.

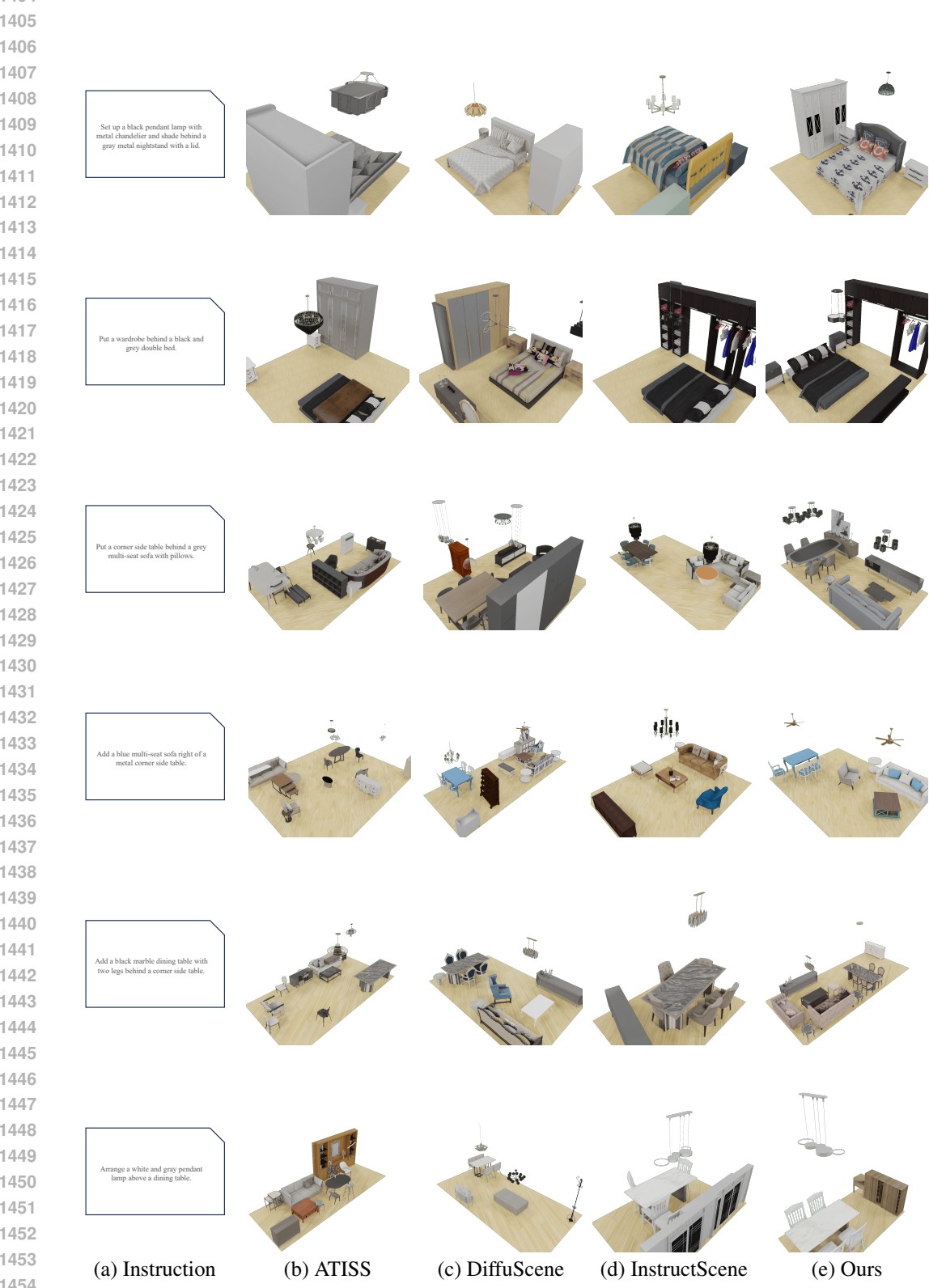

Figure 11: Comparison of qualitative results for 1 relation constraint in instruction.

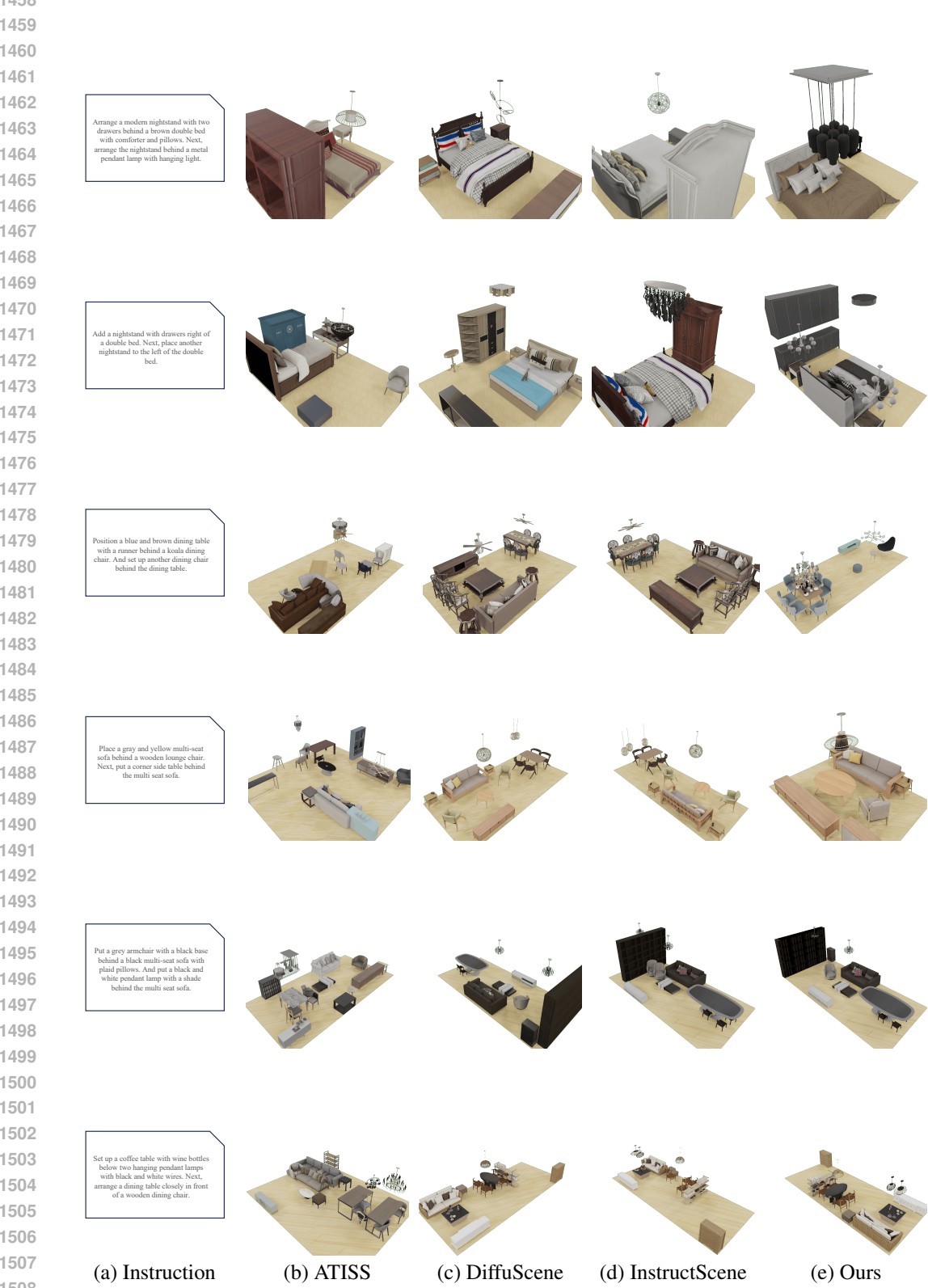

Figure 12: Comparison of qualitative results for 2 relation constraints in instruction.

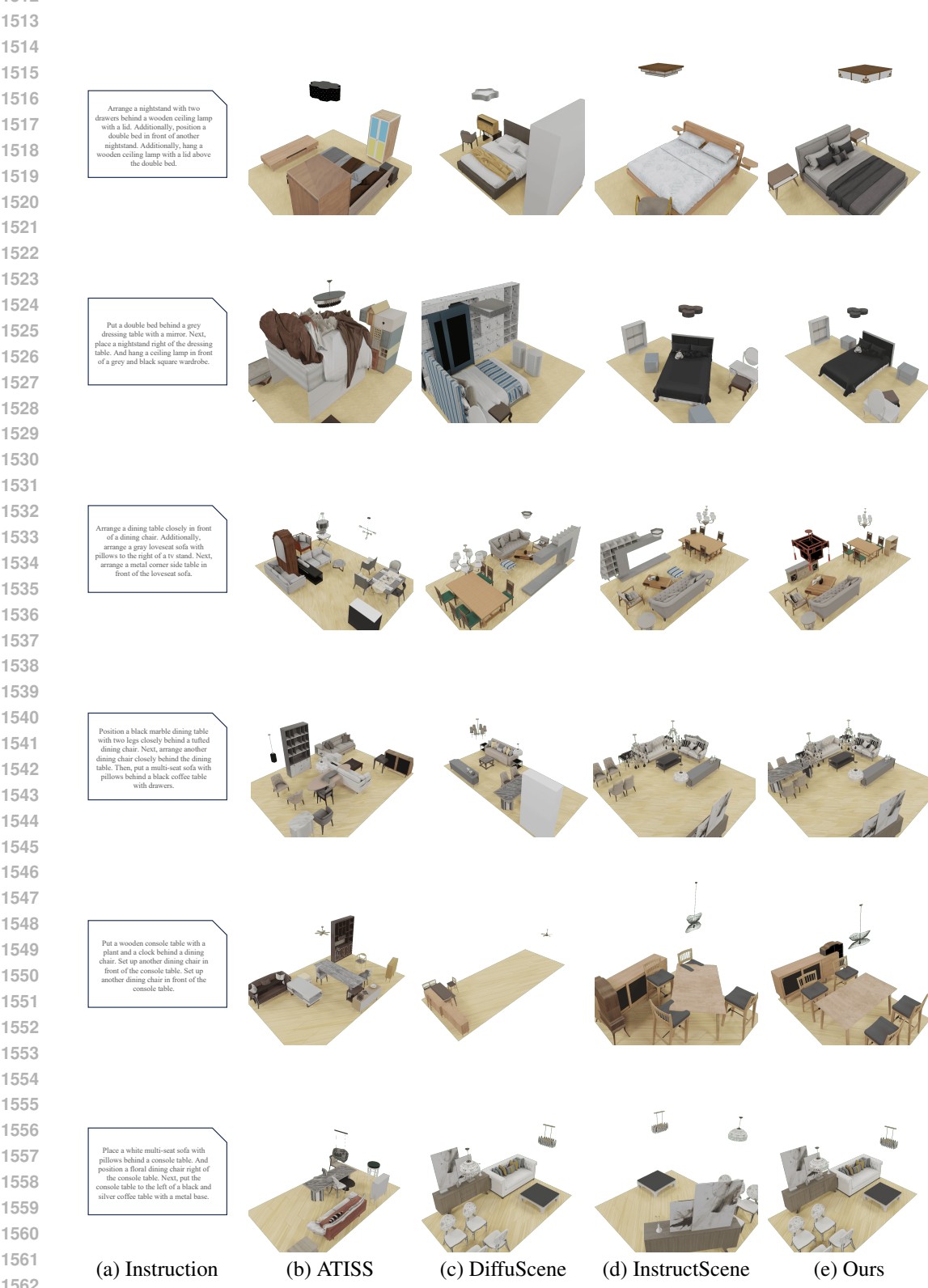

Figure 13: Comparison of qualitative results for 3 relation constraints in instruction.

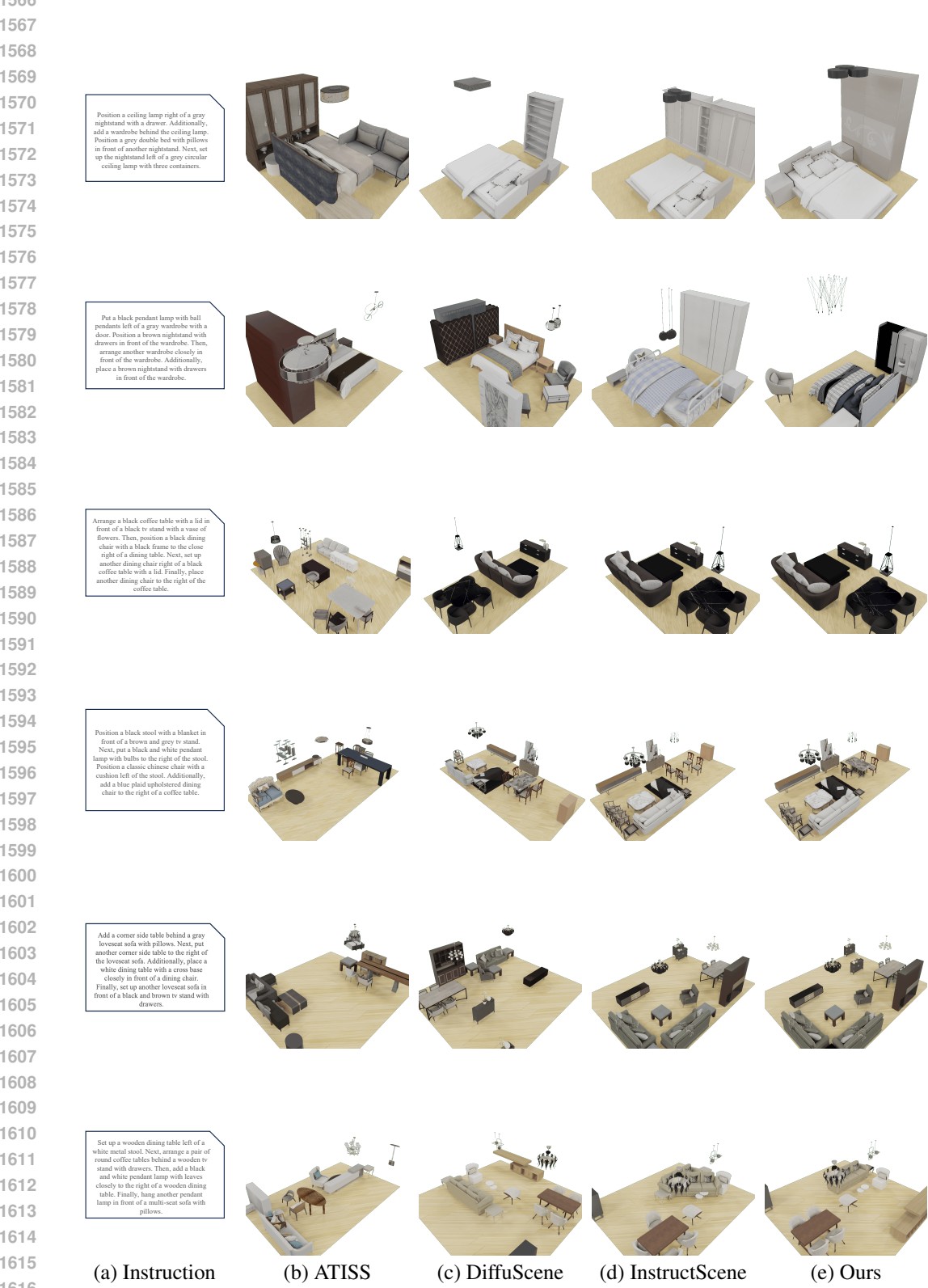

Position a ceiling lamp right of a gray nightstand with a drawer. Additionally, add a wardrobe behind the ceiling lamp. Position a grey double bed with pillows in front of another nightstand. Next, set up the nightstand left of a grey circular ceiling lamp with three containers.

Put a black pendant lamp with ball pendants left of a gray wardrobe with a door. Position a brown nightstand with drawers in front of the wardrobe. Then, arrange another wardrobe closely in front of the wardrobe. Additionally, place a brown nightstand with drawers in front of the wardrobe.

Arrange a black coffee table with a lid in front of a black tv stand with a vase of flowers. Then, position a black dining chair with a black frame to the close right of a dining table. Next, set up another dining chair right of a black coffee table with a lid. Finally, place another dining chair to the right of the coffee table.

Position a black stool with a blanket in front of a brown and grey tv stand. Next, put a black and white pendant lamp with bulbs to the right of the stool. Position a classic chinese chair with a cushion left of the stool. Additionally, add a blue plaid upholstered dining chair to the right of a coffee table.

Add a corner side table behind a gray loveseat sofa with pillows. Next, put another corner side table to the right of the loveseat sofa. Additionally, place a white dining table with a cross base closely in front of a dining chair. Finally, set up another loveseat sofa in front of a black and brown tv stand with drawers.

Set up a wooden dining table left of a white metal stool. Next, arrange a pair of round coffee tables behind a wooden tv stand with drawers. Then, add a black and white pendant lamp with leaves closely to the right of a wooden dining table. Finally, hang another pendant lamp in front of a multi-seat sofa with pillows.

(a) Instruction     (b) ATISS     (c) DiffuScene     (d) InstructScene     (e) Ours

Figure 14: Comparison of qualitative results for 4 relation constraints in instruction.

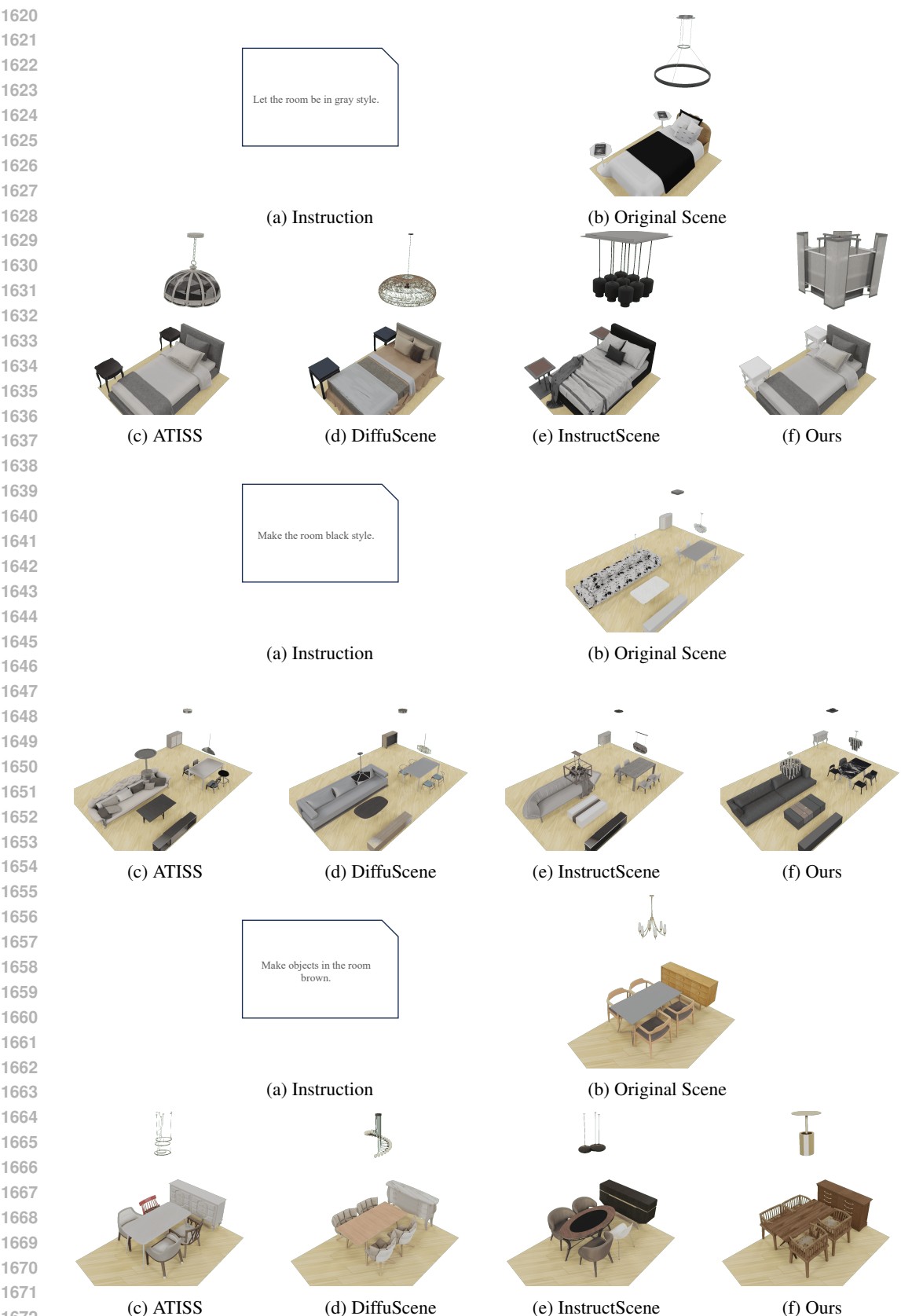

Figure 15: Comparison of qualitative results for stylization task.

Arrange a brown and beige dining chair below a metal and glass pendant lamp. Finally, place a black armchair with cushion in front of a black rectangular dining table with legs.

(a) Instruction

(b) Messy Scene

(c) ATISS

(d) DiffuScene

(e) InstructScene

(f) Ours

Figure 16: Comparison of qualitative results for rearrangement task.

Set up a coffee table with wine bottles below two hanging pendant lamps with black and white wires. Then, arrange a dining table closely in front of a wooden dining chair. Then, put another dining chair closely in front of the dining table. And put another dining chair in front of a multi seat sofa.

(a) Instruction

(b) Partial Scene

(c) ATISS

(d) DiffuScene

(e) InstructScene

(f) Ours

Figure 17: Comparison of qualitative results for completion task.

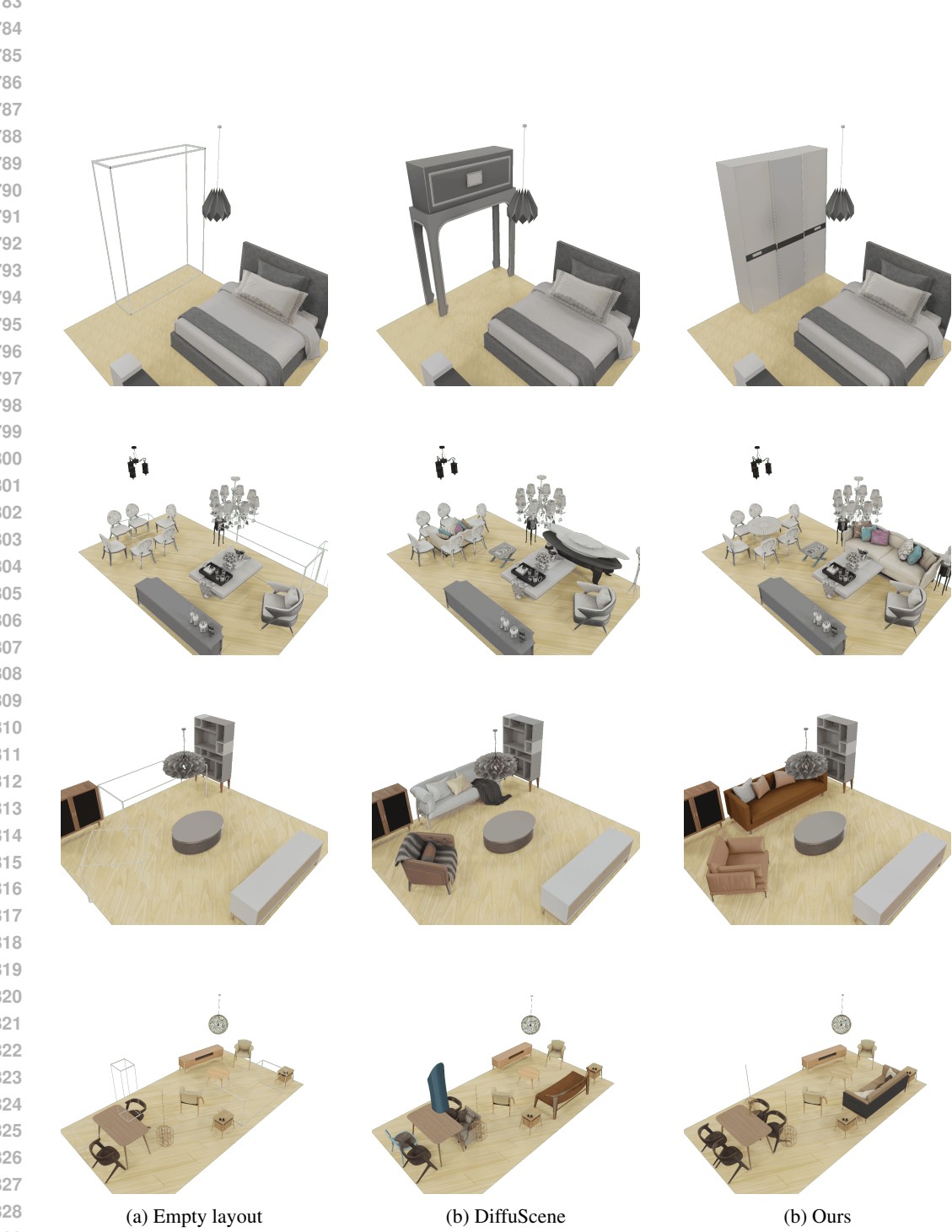

(a) Empty layout          (b) DiffuScene          (b) Ours

Figure 18: Comparison of qualitative results for layout-to-object task.

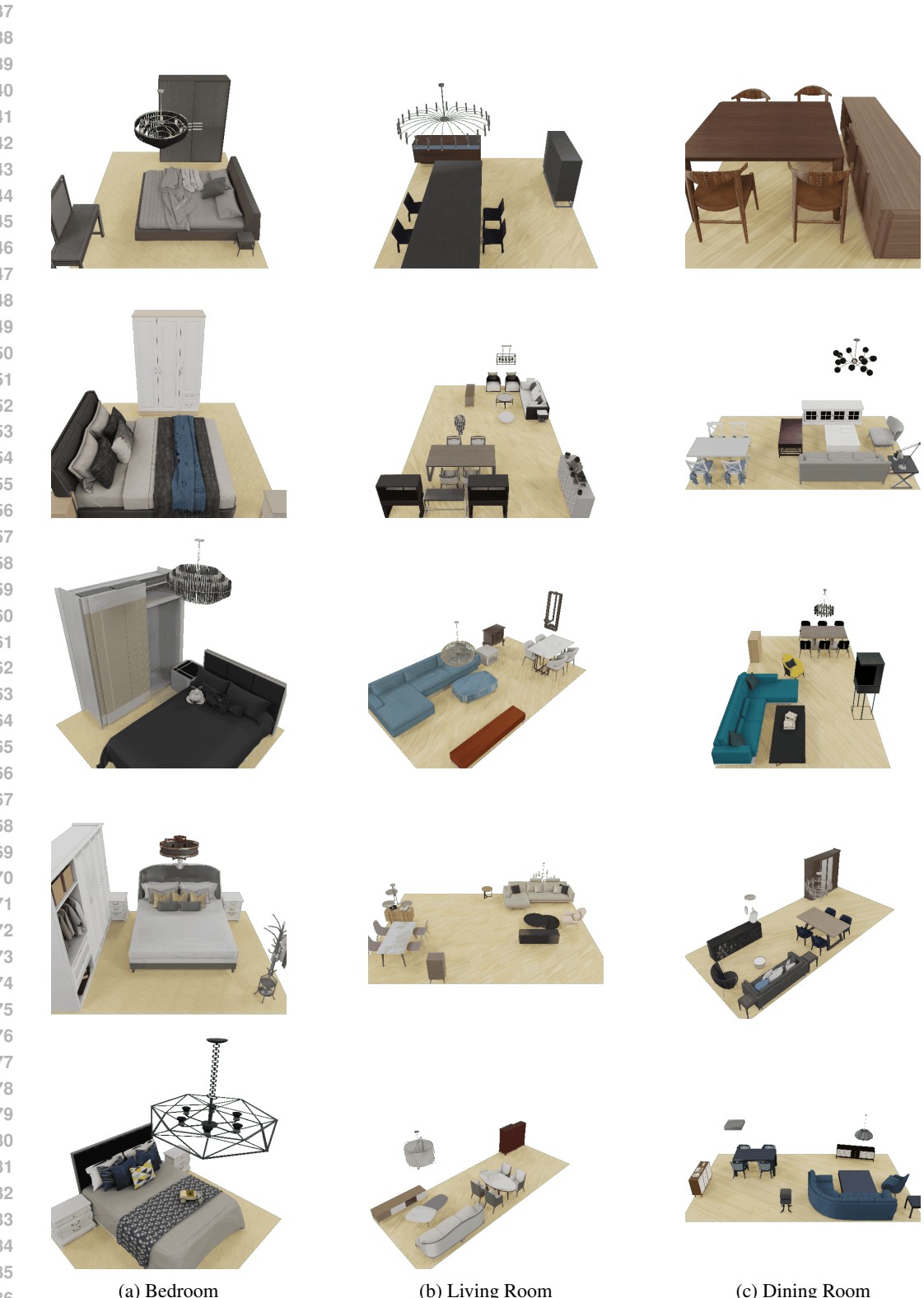

(a) Bedroom    (b) Living Room    (c) Dining Room

Figure 19: Unconditional generation results across three room types.

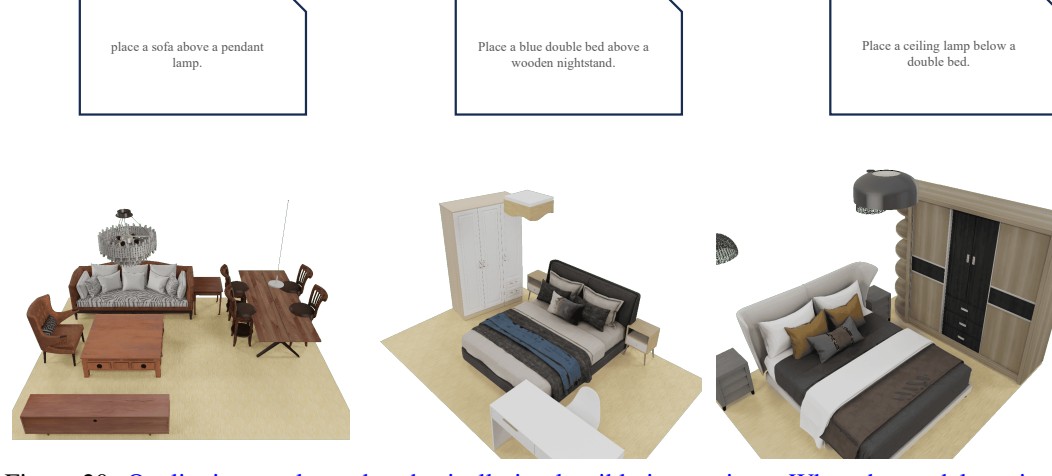

Figure 20: Qualitative results under physically implausible instructions. When the model receives constraints that violate physical common sense, SceneNAT prioritizes layout plausibility to produce valid arrangements rather than enforcing the erroneous relation.

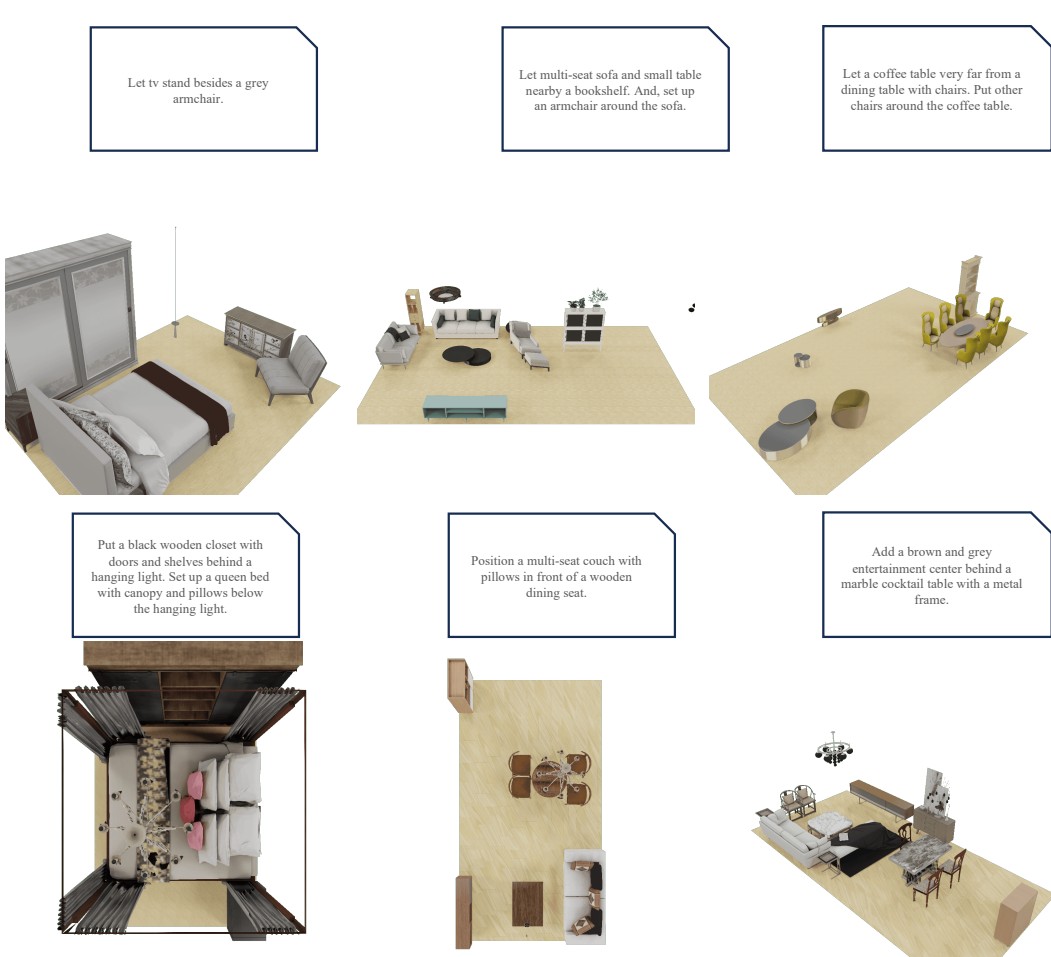

Figure 21: Qualitative results demonstrating generalization to open-vocabulary instructions that were not present during training. **Top:** The model accurately grounds unseen spatial predicates (e.g., 'besides', 'nearby', 'very far from', 'around'). **Bottom:** The model successfully interprets unseen object attributes (e.g., 'closet', 'couch', 'entertainment center', 'cocktail table'), effectively mapping free-form text to appropriate assets.

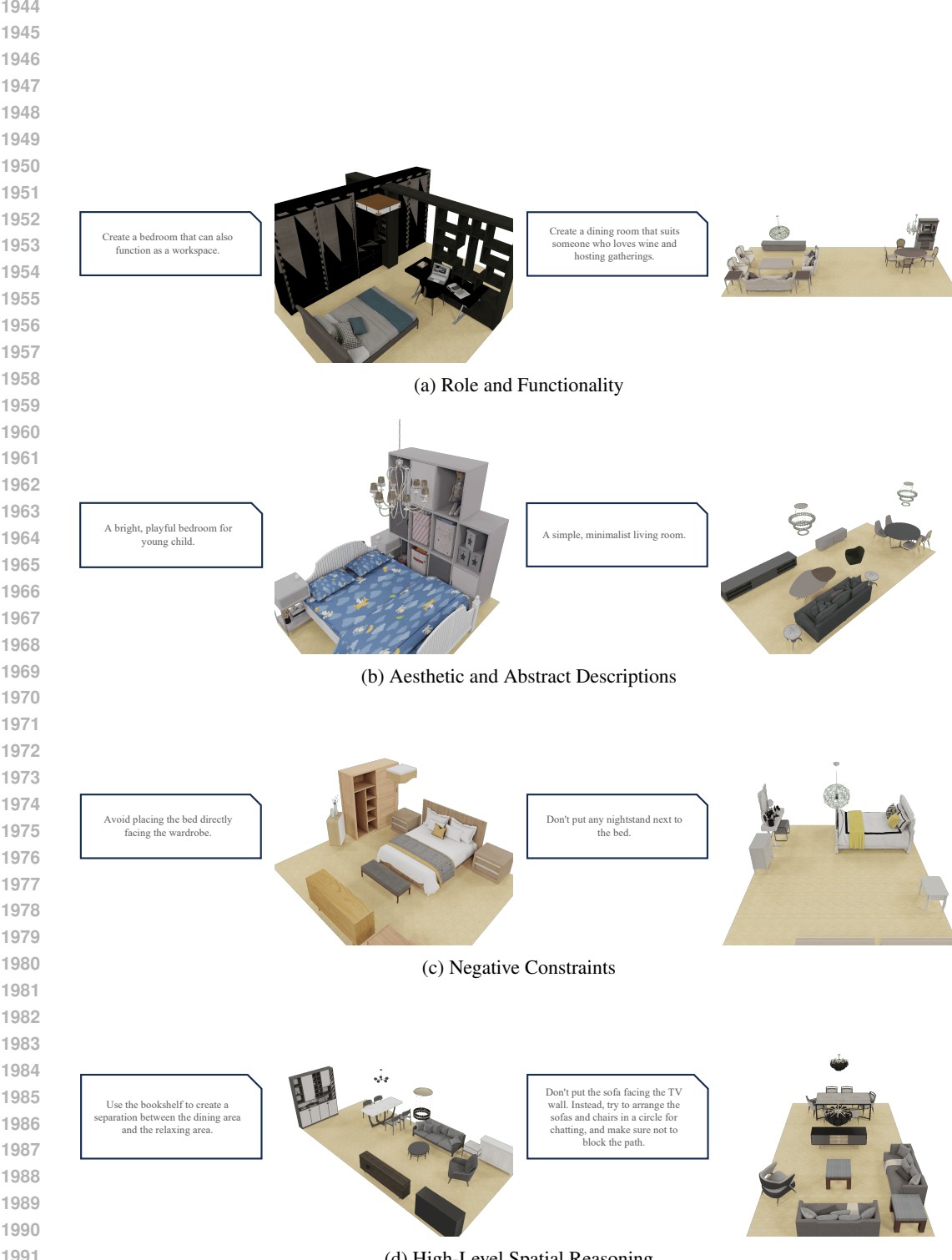

Figure 22: Qualitative results of various types of "in-the-wild" human instructions.

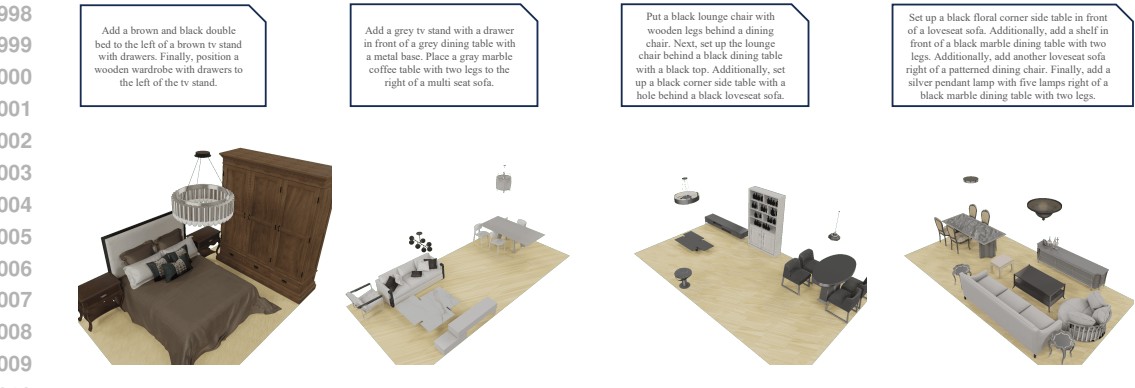

Add a brown and black double bed to the left of a brown tv stand with drawers. Finally, position a wooden wardrobe with drawers to the left of the tv stand.

Add a grey tv stand with a drawer in front of a grey dining table with a metal base. Place a gray marble coffee table with two legs to the right of a multi seat sofa.

Put a black lounge chair with wooden legs behind a dining chair. Next, set up the lounge chair behind a black dining table with a black top. Additionally, set up a black corner side table with a hole behind a black loveseat sofa.

Set up a black floral corner side table in front of a loveseat sofa. Additionally, add a shelf in front of a black marble dining table with two legs. Additionally, add another loveseat sofa right of a patterned dining chair. Finally, add a silver pendant lamp with five lamps right of a black marble dining table with two legs.

Figure 23: Failure case where the generated scene fails to satisfy the spatial relations specified in the instruction.

Set up a grey metal stool right of a nightstand with two drawers. Next, set up a blue double bed with wooden headboard left of a cabinet. Additionally, place a grey metal stool to the right of the double bed.

Arrange a black dining table with metal plates right of a black and blue stool with a cover. Then, position a black multi-seat sofa with a frame right of a black pendant lamp with brass finish. Position another stool behind a black dining table with metal plates. Arrange a tv stand with drawers left of a coffee table.

Put a tv stand right of a black pendant lamp with glass balls. Then, add a blue striped dining chair closely in front of a dining table with a vase and plates.

Put a black pendant lamp with a shade in front of a multi seat sofa. And install the pendant lamp above a silver flower coffee table runner. And add a corner side table left of a blue and white multi-seat sofa.

Figure 24: Failure case where the generated scene fails to satisfy the object attributes specified in the instruction.

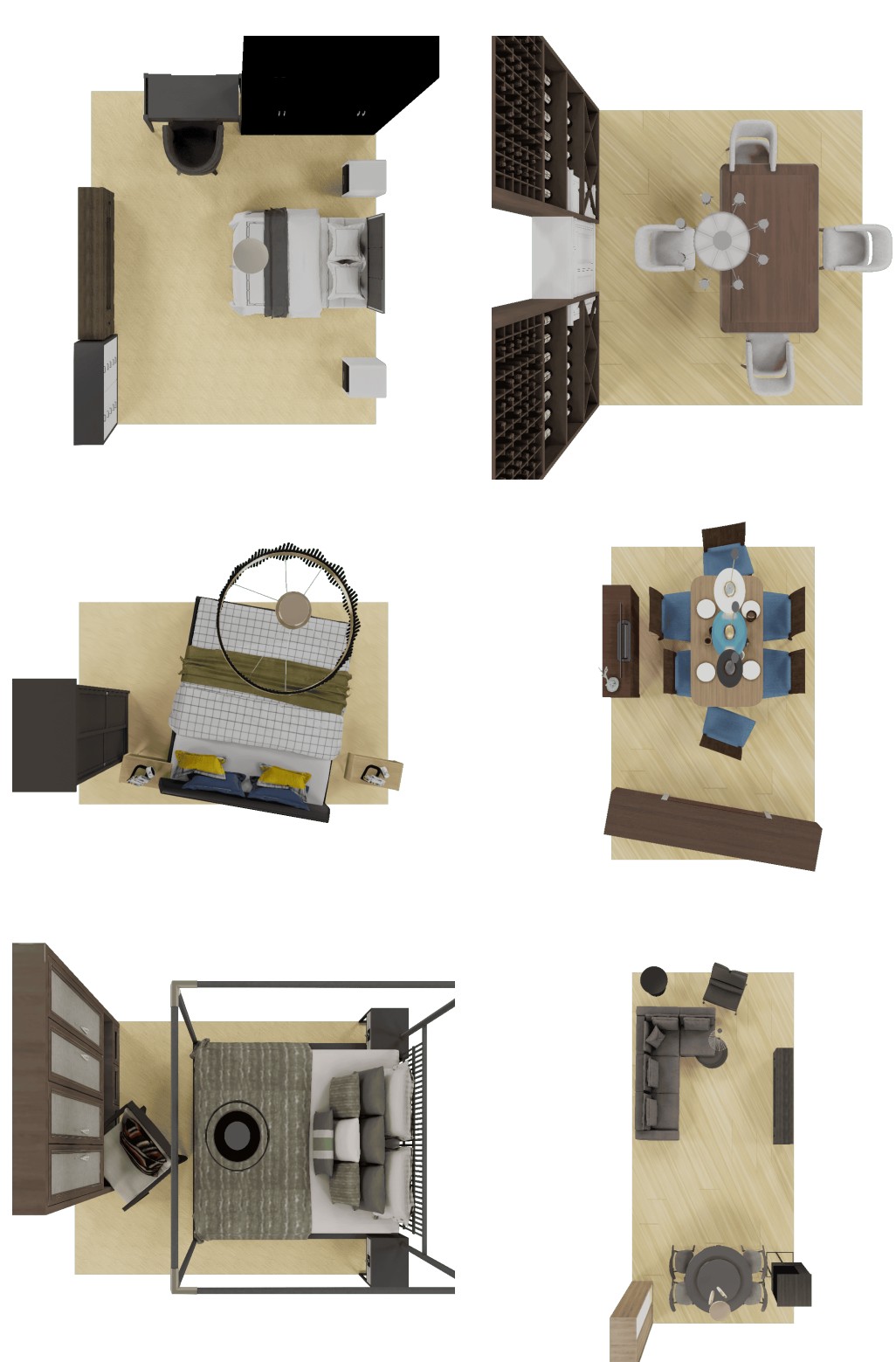

Figure 25: Failure case where the generated scene fails to satisfy physical plausibility, exhibiting object collisions or incorrect scale and rotation.

