# OpenReview forum: "SceneNAT: Masked Generative Modeling for Language-Guided Indoor Scene Synthesis"
_ICLR.cc/2026/Conference — Submitted to ICLR 2026_

### Official Review · Reviewer_kcWb · 2025-10-23

**Soundness:** 1
**Presentation:** 1
**Contribution:** 1
**Rating:** 2
**Confidence:** 5

**Summary:**

This paper proposes to use mask modeling for indoor scene representation. To improve the modeling of two-object relations, a bipartite matching is used to optimize the one-to-many mapping between each object pair. Experimental results have shown improved performance over three methods: ATISS, DiffuScene and InstructScene.

**Strengths:**

+ the introducing of a detection module for the detection of object-object relations

**Weaknesses:**

- The biggest issue for me is how model like BERT's masked modeling which is originally designed for representation learning can be used for generation. I cannot find the pipeline of how the inference is executed.
- I also doubt why a parallel decoder like DETR can outperfom a denoising deffusion based model and AR based model. There should be more analysis experimental to support this claim.
- There should be a discussion with COFS which is an encoder-decoder pipeline like BART structure for scene generation.
- Figure 2 is not clearly explained. Is triplet query an input? Is it learned or provided?
- The DETR module is not clearly explained either. In the original DETR, a bipartite matching is first optimized for matching each query with one of the preset query candidate (which is to be optimized) and then adjust all the matched query candidate to the GT ones. In the proposed method (along with Figure 2), I do not understand how the above CE loss is defined and even though it can somehow trained, how will the triplet feature update to the mask modeling is also unclear.
- Missing discussion and comparsions with some SOTA autoregressive models such as FOREST2SEQ [1] and CASAGPT [2]
[1] https://arxiv.org/abs/2407.05388
[2] https://arxiv.org/abs/2504.19478
- There are no failure case provided.
- It seems that the model cannot deal with layouts with various shapes. All the layouts shown are rectangles.

**Questions:**

See Weakness

---

> ### Author Response · Authors · 2025-11-17
> **Official Author Response to Reviewer kcWb (1/3)**
>
> Dear reviewer kcWb,
>
> We sincerely appreciate for your time and effort and hope our response satisfactorily addresses your concerns.
> # 1. Clarification on the Generative Paradigm and Inference Pipeline
>
> >  **W1**. *The biggest issue for me is how model like BERT's masked modeling... can be used for generation. I cannot find the pipeline of how the inference is executed.*
>
> We thank the reviewer for the opportunity to clarify our generative paradigm.
>
> **1-1. On Masked Modeling for Generation (vs. Representation)**
>
> We respectfully clarify that SceneNAT is not a representation model like BERT [1], but a **Masked Generative Model (MGM)**. This is a well-established class of generative models.
>
> * **Established Paradigm:** As discussed in our Related Work, MGMs are powerful alternatives to AR/diffusion models. Pioneered by Mask-Predict [2] (machine translation) and MaskGIT [3] (image synthesis), and successfully scaled in Muse [4] (text-to-image), MaskINT [5] (video editing), and MoMask [6] (motion generation), etc (sec Rel. Work). These works validate that masked modeling is a highly effective mechanism across diverse generative tasks.
> * **Bidirectional Parallel Decoding:** Unlike BART which relies on an autoregressive decoder to generate tokens sequentially, these works (and SceneNAT) utilize *bidirectional masked modeling* to generate all tokens in parallel. This allows the model to utilize global context from both directions simultaneously during generation, fundamentally differing from the sequential constraints of autoregressive models.
> * **Generative Masking Strategy:** Unlike BERT’s fixed 15% ratio, we use a dynamic cosine schedule to train the model to generate from any level of corruption, fully masked states to a nearly complete one, as detailed in Section 3.4 and Appendix A.1. Furthermore, our hierarchical masking (object- vs. token-level) is specifically designed to capture structured scene data.
>
> **1-2. On the Inference Pipeline**
> We respectfully point to Section 3.4 ("Sampling Strategy") and Appendix C (Fig. 7, 8), where the inference pipeline is detailed and visualized.
> * **Iterative Refinement:** Unlike BERT's single-pass prediction, our inference is an iterative process. Starting from a fully masked state, the model performs parallel decoding, then iteratively re-masks and regenerates low-confidence tokens over fixed steps. This progressively refines the scene from noise to completion.
>
> # 2. Clarification on Performance Claims and Experimental Support
>
> > **W2.** *I also doubt why a parallel decoder like DETR can outperfom .. analysis experimental to support this claim.*
>
> We respectfully point out that the reviewer’s premise (a parallel decoder like DETR) is a slight misunderstanding of our generative architecture.
>
> Our superior performance does not come from a DETR-like architecture, but rather from our Masked Generative Modeling (MGM) paradigm.
>
> **2-1. Architectural Clarification: MGM, not DETR:**
> Our model is not a set-prediction model like DETR. As detailed in Sec. 3.2, SceneNAT is a Masked Generative Transformer.
> * **The DETR-like module (Sec 3.3)**  is an auxiliary component (Triplet Predictor) trained to predict explicit symbolic constraints from the text instructions.
> * The Core Generator is the Masked Scene Decoder. It outperforms baselines because it utilizes Masked Indoor Scene Modeling (MISM) to capture holistic scene context bidirectionally and employs parallel iterative decoding in a discrete space (avoiding diffusion's slow continuous process).
>
> **2-2. Why SceneNAT Outperforms AR & Diffusion:**
> Our ablation studies and results support why this MGM approach is superior:
>
> * **vs. AR:** AR models (e.g., ATISS) suffer from error accumulation due to sequential generation. Our MGM generates the scene in parallel with bidirectional context, leading to significantly better structural coherence (FID 109.55 vs. 128.50 on bedroom).
> * **vs. Diffusion:** Diffusion models (e.g., InstructScene, DiffuScene) struggle to map symbolic text constraints to continuous noise. Our discrete token space combined with the auxiliary Triplet Predictor allows for much more precise control (iRecall 70.45% vs. 66.72% for InstructScene, 45.98% for DiffuScene) and faster convergence (>5x faster than InstructScene, >24x faster than DiffuScene).
>
> **2-3. Experimental Evidence of the Integration:**
> Table 3 explicitly analyzes this. When we remove the auxiliary "DETR-like" Triplet Predictor (`w/o triplet predictor`), the generative quality (FID) remains stable, but the controllability drops significantly (iRecall -7.7%).
>
> This confirms that:
> 1.  High Fidelity comes from the MGM architecture, not the DETR module.
> 2.  High Controllability comes from the integration of the MGM and the Triplet Predictor.
>
> In summary, our model outperforms baselines not because it is a DETR, but because it effectively integrates holistic scene synthesis (via MISM) with explicit constraint modeling (via the Triplet Predictor).

---

> ### Author Response · Authors · 2025-11-17
> **Official Author Response to Reviewer kcWb (2/3)**
>
> # Clarification on Comparisons with Autoregressive Models (COFS [8], Forest2Seq [9], CasaGPT [10])
>
> > **W3.** *There should be a discussion with COFS which is an encoder-decoder pipeline like BART structure for scene generation.* \
> > **W6.** *Missing discussion and comparisons with some SOTA autoregressive models such as FOREST2SEQ and CASAGPT.*
>
> We thank the reviewer for these helpful suggestions. We currently cite COFS in our Introduction as a key example of a strong autoregressive (AR) approach. We will also incorporate discussions on Forest2Seq and CasaGPT to contextualize our approach against these state-of-the-art AR models.
>
> We agree that discussing these works highlights the structural advantage of our SceneNAT (parallel Non-Autoregressive) over sequential AR pipelines. However, a direct quantitative experimental comparison is infeasible for two main reasons:
>
> 1.  **Fundamental Task Mismatch:** While these methods generate 3D scenes, their primary conditioning and objectives differ significantly from our **text-conditioned** instruction following task:
>     * **COFS:** Focuses on layout synthesis conditioned on room boundaries or partial object shapes, rather than natural language instructions.
>     * **Forest2Seq:** Formulates synthesis as an order-aware sequential learning problem, focusing on constructing hierarchical scene trees and forests to recover generation order, which is distinct from instruction-driven generation.
>     * **CasaGPT:** Focuses on arranging decomposed cuboid primitives for scene assembly. Its primary contribution lies in modeling objects as cuboids to minimize intersections during sequential arrangement, rather than interpreting complex text descriptions.
>
> 2.  **Reproducibility:** We investigated the official repositories for all three methods to attempt a comparison, but were unable to access working implementations:
>     * **COFS & Forest2Seq:** The official codes have not been publicly released.
>     * **CasaGPT:** While a GitHub repository exists, it is currently empty and marked as "preparing for release."
>
> Despite these limitations preventing direct comparison with these specific methods, **we remain committed to rigorous evaluation. We are actively surveying other recent SOTA works** to identify any feasible baselines that share our text-conditioned setting and have released code. Should we identify a reproducible method that allows for a fair quantitative comparison, we will eagerly conduct additional experiments and report the results to further demonstrate the capabilities of SceneNAT.
>
> # Clarification on Figure 2 and Triplet Queries
> > **W4.** *Figure 2 is not clearly explained. Is triplet query an input? Is it learned or provided?*
>
> We thank the reviewer for the question and the opportunity to clarify Figure 2.
>
> Yes, the triplet queries are inputs to the Triplet Predictor module. They are a fixed set of learnable queries (i.e., learned model parameters), not provided as data.
>
> Also, We respectfully point to our explanation in **Section 3.3**, which states:
>
> > "This module uses a transformer decoder to transform a fixed set of $N_q$ **learnable triplet queries** into relation-aware features..."
>
> We appreciate this feedback and are happy to revise the caption as below:
>
> **Revised Figure 2 Caption (excerpt):**
> "...Concurrently, a triplet predictor transforms a set of ***learnable triplet queries*** into specific relational constraints parsed from the text instruction..."
>
> We believe that this change will make the role of the triplet queries clearer in Figure 2.
>
>
> # Regarding Failure Cases
> > **W7.** *There are no failure case provided.*
>
> We agree that analyzing failure modes is essential for a balanced evaluation and for understanding the model's boundaries.
>
> We are currently preparing a dedicated **Failure Analysis** section and will update the Appendix shortly. To provide a comprehensive diagnostic, we are designing this section to address the concern raised by **R-kcWb** regarding the lack of such analysis, while specifically categorizing failure modes based on the feedback from **R-kcWb and other reviewers (vGcy, m5Vp, EG12)** as follows:
>
> 1.  **Complex & Ambiguous Instructions:** Addressing the concerns of **R-vGcy** and **R-EG12**, we will visualize cases where the model struggles with abstract, "in-the-wild" prompts that lack explicit relational structures or exceed the current relation capacity.
> 2.  **Physical Inconsistencies & Collisions:** Addressing **R-m5Vp's** suggestion regarding diagnostics, we will report cases where object collisions or physically implausible arrangements occur, particularly when user instructions conflict with learned spatial priors.
> 3.  **Discretization Artifacts:** We will also include examples showing minor spatial misalignments due to the quantization process, as noted by **R-m5Vp, EG12**.
>
> We will notify you as soon as the revised paper with these additional qualitative results is uploaded.

---

> ### Author Response · Authors · 2025-11-17
> **Official Author Response to Reviewer kcWb (3/3)**
>
> # Clarification on the DETR Module and Triplet Predictor Mechanism
> > **W5**. *The DETR module is not clearly explained either. In the original DETR, a bipartite matching is first optimized... I do not understand how the above CE loss is defined and... how will the triplet feature update to the mask modeling is also unclear.*
>
> We thank the reviewer for the opportunity to clarify the mechanisms of our Triplet Predictor and its integration with the scene generation process. It appears there may be a misunderstanding regarding the standard DETR formulation we adopted and the information flow within our architecture.
>
> **1. Triplet Predictor Mechanism and Loss (Sec. 3.3)**
>
> Regarding the matching mechanism, we follow the standard DETR formulation (Carion et al., 2020 [11]).
>
> * **Bipartite Matching:** In the DETR framework, the bipartite matching is performed directly between the **model's final predictions** (set of (s,p,o) generated from learnable queries) and the **ground truth set**, rather than utilizing intermediate preset candidates. Following this, our Triplet Predictor uses the **Hungarian algorithm** to find the optimal one-to-one matching between the $N_q$ *predicted* triplets and the $N_t$ *ground-truth* triplets parsed from the instruction. This ensures the model learns to generate the correct set of unordered relational constraints.
> * **CE Loss Definition:** Once the optimal matching is established (Eq. 1), the standard weighted cross-entropy loss is calculated for each matched pair (equation (2) in Section 3.3). We independently compute the classification loss for the subject, predicate, and object categories against their matched ground truth labels. If a query is matched to the "no-relation" ($\varnothing$) target, the loss is applied with a lower weight to handle class imbalance.
>
> **2. Integration of Triplet Features (Sec. 3.2)**
>
> The reviewer asked how the triplet feature "updates" the mask modeling. We would like to clarify that the triplet features do not update the masking process itself; rather, they serve as a conditioning signal for the **Layout Decoder**.
>
> * **Conditioning via Cross-Attention:** As illustrated in **Figure 2**, the explicit relational embeddings (triplet features) produced by the Triplet Predictor are fed into the **cross-attention** layers of the Layout Decoder. This allows the Layout Head to refine spatial attributes (translation, scale, rotation) by attending to the explicit symbolic constraints.
>
> **3. Distinction from Standard DETR Inference**
>
> We suspect the confusion regarding "updating the mask modeling" might stem from the standard usage of DETR.
> * **Standard DETR:** The Transformer Decoder output *directly becomes* the final object detection result (Class + Bounding Box).
> * **SceneNAT (Ours):** The Triplet Predictor (DETR-like module) output serves as an **intermediate representation** (Relational Embeddings). These embeddings are not the final scene objects themselves but are injected into the Layout Decoder via **cross-attention** to condition the spatial prediction.
>
> In the final version, we will **further highlight** the distinction from standard DETR inference in Section 3.3 to ensure clarity for a broader audience.
>
> # Clarification on Layout Shapes and Floor Visualization
>
> > **W8.** *It seems that the model cannot deal with layouts with various shapes. All the layouts shown are rectangles.*
>
> We thank the reviewer for this observation. We would like to clarify that the rectangular floors appearing in our figures are solely an artifact of the visualization process, not a constraint on the model's generation capabilities.
>
> **1. Diversity of Generated Layouts**
> Our model is designed to generate object arrangements in open space, conditioned on text instructions rather than a fixed floor plan. Consequently, the generated object layouts themselves are not constrained to rectangular boundaries. As illustrated in our qualitative results (e.g., **Figure 3** and **Figure 9**), the model produces diverse object distributions that reflect the spatial relations described in the text, independent of a specific room shape.
>
> **2. Floor Visualization Method (Post-processing)**
> The rectangular floors seen in the figures are added **post-generation** to provide a visual ground for the floating objects.
> * **Method:** For visualization, we calculate the bounding box (minimum and maximum x, y coordinates) of the generated object set and simply fit a rectangular floor tile to this area.
> * **Standard Practice:** This is a standard visualization technique widely used in works that focus on layout synthesis without explicit floor plan conditioning, such as **DiffuScene** and **InstructScene**.
>
> Therefore, the rectangular shape is merely a visualization aid and does not reflect a limitation in the model's ability to handle diverse spatial arrangements.
>
> ---
> We hope these clarifications satisfactorily address your concerns, and we welcome any further discussion.

---

> > ### Author Response · Authors · 2025-11-17
> > **References**
> >
> > [1] Devlin, J., et al. Bert: Pre-training of deep bidirectional transformers for language understanding. NAACL, 2019. \
> > [2] Ghazvininejad, M., et al. Mask-Predict: Parallel decoding of conditional masked language models. EMNLP, 2019. \
> > [3] Chang, H., et al. MaskGIT: Masked generative image transformer. CVPR, 2022 \
> > [4] Chang, H., et al. Muse: Text-to-image generation via masked generative transformers. PMLR, 2023. \
> > [5] Ma, H., et al. Maskint: Video editing via interpolative non-autoregressive
> > masked transformers. CVPR, 2024 \
> > [6] Guo, C., et al. Momask: Generative masked modeling of 3d human motions. CVPR, 2024. \
> > [7] Lewis, M., et al. BART: Denoising sequence-to-sequence pre-training for natural language generation, translation, and comprehension. ACL, 2020 \
> > [8] Para, W., et al. Cofs: Controllable furniture layout synthesis. ACM SIGGRAPH, 2023. \
> > [9] Sun, Q., et al. Forest2seq: Revitalizing order prior for sequential indoor scene synthesis. ECCV, 2024. \
> > [10] Feng, W., et al. CasaGPT: cuboid arrangement and scene assembly for interior design. CVPR, 2025. \
> > [11] Carion, N., et al. End-to-end object detection with transformers. ECCV, 2020.

---

> ### Author Response · Authors · 2025-11-27
> **Follow-up Author Response to Reviewer kcWb (1)**
>
> ### **Comparison with recent state-of-the-art baseline**
>
> To rigorously address the feedback regarding comparisons with recent state-of-the-art systems, we evaluated the feasibility of benchmarking against various concurrent works. We identified *FreeScene (CVPR 2025)* [12] as the most appropriate and rigorous baseline. As a Mixed Graph Diffusion Transformer, it represents a strong state-of-the-art approach with a task formulation comparable to ours. Although the official code is unavailable, we selected it for comparison as its reported metrics allow for a meaningful quantitative assessment, unlike other concurrent works with differing evaluation protocols.
>
> For a fair comparison,**we source the quantitative results for both the standard FreeScene model and all baseline methods directly from Table 3 of the FreeScene paper.** We excluded their "Graph Designer" variant because it utilizes a VLM to infer objects and relations that closely approximate the ground truth, which would provide an unfair advantage over standard text-conditioned baselines. Crucially, while SceneNAT is trained on a harder dataset with up to 4 relational constraints, baselines typically target only 1 or 2. Therefore, we evaluate SceneNAT's iRecall on the subset of instructions with 1 or 2 relational constraints ($N \le 2$) to match the complexity, while visual quality metrics are averaged over the entire test set.
>
> | Room Type | Method | iRecall ($\uparrow$) | FID ($\downarrow$) | FID$_\text{CLIP}$ ($\downarrow$) | KID$_{\times 10^3}$ ($\downarrow$) |
> | :--- | :--- | :---: | :---: | :---: | :---: |
> | **Bedroom** | ATISS | 44.06 | 122.37 | 8.23 | 0.74 |
> | | DiffuScene | 53.32 | 129.34 | 9.66 | 0.81 |
> | | InstructScene | 72.71 | 114.86 | 6.52 | 0.68 |
> | | FreeScene | *73.69* | *111.21* | *6.43* | *0.35* |
> | | **Ours** | **77.96** | **109.55** | **6.19** | **-1.18** |
> | **Living room** | ATISS | 36.31 | 120.10 | 7.43 | 16.44 |
> | | DiffuScene | 39.58 | 135.93 | 10.71 | 29.07 |
> | | InstructScene | 57.21 | 111.52 | 5.91 | 8.65 |
> | | FreeScene | *58.16* | *110.55* | *5.83* | *7.95* |
> | | **Ours** | **59.56** | **110.28** | **5.49** | **6.18** |
> | **Dining room** | ATISS | 32.56 | 134.75 | 9.82 | 24.25 |
> | | DiffuScene | 37.17 | 142.37 | 11.76 | 28.36 |
> | | InstructScene | 61.47 | *129.13* | 8.24 | 15.27 |
> | | FreeScene | **63.39** | **127.28** | *8.01* | *14.83* |
> | | **Ours** | *61.89* | 129.65 | **7.51** | **12.26** |
>
> SceneNAT demonstrates superior or highly competitive performance against FreeScene in the standard setting. In bedroom and living room, SceneNAT outperforms FreeScene across all metrics, achieving significant gains in iRecall. In dining room, while FreeScene holds a slight advantage in iRecall and FID, SceneNAT achieves superior perceptual quality (FID$_\text{CLIP}$ and KID).
>
> Furthermore, our discretization ablation study reveals that increasing the bin resolution to 128 further improves visual quality, achieving an FID of **108.98** in living room and **127.72** in dining room. This effectively matches or surpasses FreeScene's FID across all room types.
>
> **Limitations on Further Comparison**
>
> Please note that since the official code for FreeScene has not been released as of November 27, 2025 (https://github.com/cangmushui/FreeScene) , it is currently infeasible to conduct further comparative experiments such as retraining the model on our exact split or generating qualitative visualizations for a side-by-side comparison. We believe that re-implementing such a complex system without the original source code risks missing critical implementation details, which could lead to an unfair comparison for either approach. Therefore, we limit our evaluation to the reported quantitative metrics which sufficiently demonstrate the competitiveness of SceneNAT.
>
> ---
> [12] Bai, T., et al. FreeScene: Mixed Graph Diffusion for 3D Scene Synthesis from Free Prompts. CVPR, 2025.
>
> ---
> # **Closing Remark**
>
> Dear Reviewer kcWb,
>
> As we approach the final 5 days of the discussion period, we respectfully invite you to review our responses along with the revised manuscript.
>
> If you find that our responses and revisions have satisfactorily resolved your concerns, we would be grateful if you could consider updating your score. We remain available for any further discussions should you have additional questions.
>
> Sincerely,
> The Authors

---

> > ### Comment · Reviewer_kcWb · 2025-11-27
> >
> > Hi, thanks for your response. I believe my major concerns are still there. The model design looks not novel for me; the discussion of the difference between COFS and the propsed method is not clear, where the addition seems to be the language prompt, which is simply a condition that has already been applied in indoor scene synthesis tasks. Additionally, for the failure case shown in Appendix I, it seems that the model still cannot OVERCOME the mismatch between the language instruction and scenes. I'd like to know the detailed remedies or further ablations.

---

> ### Author Response · Authors · 2025-11-28
> **Discussion on the difference between COFS and SceneNAT**
>
> Thank you for your follow-up comments. We hope a detailed discussion below fully addresses your concerns.
>
> SceneNAT fundamentally differs from COFS [8] not merely in input modality, but in its **generative paradigm**, **scene representation**, **inference mechanism**, and **performance capabilities**.
>
>
> 1. While COFS employs a **masking** objective during training (similar to BART [7]), this is primarily for **representation learning**. Crucially, its **inference remains autoregressive**, generating tokens sequentially one by one ($P(s_i | S_{<i})$). This relies on a **causal mask** in the decoder, forcing the model to operate with unidirectional context.
>
> 2. This distinction is grounded in the fundamental definitions of these architectures.
> We cite Section 2 of BART which explicitly define themselves as:
>     > "It is implemented as a sequence-to-sequence model with a bidirectional encoder over corrupted text and a **left-to-right autoregressive** decoder." [Lewis et al., 2019]
>
>    The COFS paper aligns itself with BART while citing MaskGIT as a distinct parallel sampling approach, stating:
>    > "MaskGIT shows that... high quality image samples can be generated... with parallel sampling... Our model most closely resembles BART." [Para et al., 2023]
>
>    Furthermore, in Appendix C.4 in COFS, it states:
>    > "MaskGIT find that using a robust masking strategy is important, ... We see in Fig. 8e that masking with a uniform ratio of 15% leads to better NLL ...  we found out that the **we could not sample from such a trained network**, as it would output a stop token after only generating a few objects, ..." [Para et al., 2023]
>
>    This confirms that the authors of COFS view their work as *distinct* from the parallel generative paradigm that SceneNAT belongs to.
>
>     In contrast, MaskGIT establishes the Masked Generative Modeling paradigm specifically to enable parallel decoding via bidirectional attention:
>     > "In autoregressive decoding, tokens are generated sequentially... This process is not parallelizable... We introduce a **novel decoding method** where all tokens in the image are generated simultaneously in parallel. This is feasible due to the bi-directional self-attention..." [Chang et al., 2022]
>
>    It also cites **ImageBART [13]** as distinct previous work, which share the same architectural and generation philosophy with BART, COFS (transformer encoder & decoder).
>
> 3. As a result of this sequential nature, COFS shares the intrinsic limitations of AR models (e.g., error accumulation). This is evident in **Table 3 of the COFS paper**, where its generation quality (FID) is often comparable to or even slightly worse than ATISS. Even though direct comparison is infeasible due to limited reproducibility, at least SceneNAT significantly outperform ATISS.
>
> 4. SceneNAT utilizes **Masked Generative Modeling (MGM)** as its decoding mechanism. We do not use causal masking; instead, we employ **bidirectional attention** during **both training and inference**. Our model generates the entire scene **in parallel (similar to diffusion models)** by iteratively predicting masked tokens based on the full global context ($P(S | S_{\text{masked}})$).
>
>    To the best of our knowledge, SceneNAT is the first framework to successfully apply such a masked non-autoregressive transformer to the domain of 3D indoor scene synthesis. This structural advantage allows SceneNAT to significantly outperform AR baseline in both scene quality and instruction adherence.
>
> 5. As detailed in Section 3.1 of the COFS paper, it models spatial attributes (position, size, orientation) as continuous variables using a *mixture of logistic* distributions.
> We unify all scene attributes into a fully discretized token space by quantizing continuous spatial variables into discrete bins.
>
> 6. Last but not least, we respectfully point that it is more than simply adding a language condition. Our experiments with ATISS conditioned on CLIP text embeddings demonstrate that the **naive** approach is inadequate for capturing fine-grained spatial instructions (iRecall 31.30% vs. SceneNAT 70.45% in bedroom). It struggles to ground complex relations from unstructured text. Also, **naively conditioning** text instruction yields suboptimal results, demonstrated in our ablation study: the full SceneNAT model (70.45%) clearly outperforms the variant without the triplet predictor (62.77%).
> We believe that this architectural design is beneficial for grounding unstructured text into geometric constraints and **is absent** in other methods including COFS, which is mainly boundary-conditioned.
>
> ---
> **References**
>
> [13] Esser, P., et al. Imagebart: Bidirectional context with multinomial diffusion for autoregressive image synthesis. NeurIPS, 2021.

---

> ### Author Response · Authors · 2025-11-28
> **Follow-up Author Response**
>
> # Regarding Failure Cases
>
> 1. We agree that "mismatches" are an inevitable part of probabilistic generation, as no model currently achieves perfect instruction adherence.
>
>    We clarify that our claim of "**overcoming** a fundamental limitation" refers to addressing the structural bottlenecks of prior methods rather than guaranteeing zero error generation. Prior autoregressive approaches suffered from unidirectional context and diffusion models often lacked explicit relation modeling. SceneNAT resolves these issues via bidirectional masked modeling and a dedicated triplet predictor. We demonstrate the effectiveness of our approach through the **notable decrease of such mismatches relative to prior methods.** The results in Table 1 and Figure 4 confirm that SceneNAT outperforms baselines in adhering to complex spatial constraints.
>
>    However, if the term "overcome" might sound too absolute given the remaining failure cases, we are willing to adopt a more precise term such as "ameliorates" or "mitigates" in the final manuscript to reflect the probabilistic nature of our model.
>
> 2. We respectfully argue that the Triplet Predictor proposed in our framework is precisely the **structural remedy** designed to address the mismatch between instructions and scenes. Instruction adherence (iRecall) requires accurate alignment of not just spatial relations but also the semantic correctness of the objects themselves.
>
>    To achieve this our Triplet Predictor is trained via a direct set prediction formulation with bipartite matching which explicitly supervises the model to capture subject-predicate-object dependencies. Then the learned relational features are utilized during inference to guide the scene generation.
>
>    This approach fundamentally differs from InstructScene and the concurrent work FreeScene (**discussed in previous rebuttal**) which rely on graph transformer backbones within a diffusion framework (also **computationally heavy**). The proposed methodology allows for more explicit grounding of relational information compared to recent baselines and we believe our contribution is clearly presented regarding this. Our ablation study in Table 7 quantitatively proves its effectiveness as removing the **triplet predictor** results in a significant drop in adherence performance from 70.45% to 62.77%.
>
>    Furthermore, Table 8 demonstrates that increasing the **guidance scale** can further improve adherence which allows for a controllable trade off between visual diversity and instruction fidelity. As evidenced in Table 1 and our supplementary comparison our model significantly outperforms baselines like DiffuScene and InstructScene and achieves competitive or superior performance against FreeScene particularly in bedroom and living room scenarios.
>
>    While we acknowledge that mismatches cannot be entirely eliminated due to the probabilistic nature of the task, we **will explicitly discuss this remaining issues to limitation section.**

---

### Official Review · Reviewer_EG12 · 2025-10-27

**Soundness:** 3
**Presentation:** 3
**Contribution:** 3
**Rating:** 6
**Confidence:** 4

**Summary:**

This work introduces SceneNAT, a single-stage, masked non-autoregressive Transformer for language-guided 3D indoor scene synthesis. SceneNAT enhances performance and efficiency by employing masked modeling on discretized semantic and spatial attributes, alongside a novel triplet predictor for relational reasoning.

**Strengths:**

- **Novelty**. The method introduces a tailored non-autoregressive Transformer (NAT) for 3D scene synthesis, using dual-granularity masked modeling (attribute- and instance-level) to capture intra- and inter-object dependencies. The decoupled triplet predictor improves spatial relation modeling, overcoming limitations of traditional text representations.

- **Comprehensive Experiments**. The experiments are thorough, including quantitative metrics (iRecall, FID/CLIP-FID/KID) and qualitative comparisons (Figures 3, 9-13). Ablation studies (Tables 3, 5) validate key components, and zero-shot tasks (e.g., stylization, layout-to-object) demonstrate generalization.

- **Writing**. The manuscript is well-structured and clearly written, with precise technical details that make it accessible to both specialists and a broader audience.

**Weaknesses:**

1. **Limited Generalization to Complex Relational Scenarios**. This work limits the maximum number of relational constraints per instruction to 4, citing the token length limit of the CLIP text encoder. However, it does not address how the model would scale to more complex and realistic design scenarios. Additionally, the paper only validates 11 predefined spatial relations (e.g., "right of", "above") from InstructScene, missing common fine-grained or ambiguous relations, and still faces the issue of limited instruction diversity as noted in InstructScene.

2. **Unaddressed Discretization Bias and Error**. This work discretizes continuous 3D attributes into fixed bins but does not analyze how the granularity of discretization impacts generation quality. This is a significant oversight: coarser bins may introduce spatial errors (e.g., object overlaps due to imprecise position prediction), while finer bins could lead to increased model uncertainty.

**Questions:**

Please refer to the "Weaknesses" section.

---

> ### Author Response · Authors · 2025-11-20
> **Official Author Response to Reviewer EG12 (1/2)**
>
> Dear reviewer EG12,
>
> We sincerely thank the reviewer for the positive assessment and constructive feedback. We are encouraged that the reviewer recognized the novelty of our masked non-autoregressive framework and the thoroughness of our experimental validation. We address the specific concerns below.
> # Generalization capacity
>
> > **W1-1.** *This work limits the maximum number of relational constraints per instruction to 4, citing the token length limit of the CLIP text encoder. However, it does not address how the model would scale to more complex and realistic design scenarios.*
>
> We appreciate the reviewer's inquiry regarding the scalability of our model to more complex scenarios.
>
> **Robustness against Baselines on Complex Instructions**
> We conducted additional experiments on instructions containing 5 and 6 relational constraints using the model trained with $N_q=4$. As templated instructions with 5 or 6 relational constraints often exceed the CLIP [1] token limit, we additionally pre-processed these instructions by pruning non-essential descriptive words while preserving the core subject-predicate-object $(s, p, o)$ triplets.
>
> As shown in the results below, while InstructScene shows a degradation of iRecall as the number of relations increases, SceneNAT remarkably maintains consistent performance across all room types. We highlight that sustaining such high iRecall is a non-trivial achievement, especially considering that these complex scenarios were **unseen during training**. We interpret this results that training the scene decoder along with auxiliary triplet predictor promotes the inference of relational information (i.e., layout).
>
> | # Relations | Room Type | **SceneNAT (Ours)** | InstructScene | DiffuScene | ATISS |
> | :--- | :--- | :---: | :---: | :---: | :---: |
> | **n=4** | Bedroom | **70.45** | 66.72 | 45.98 | 31.30 |
> | | Living | **50.01** | 47.97 | 27.39 | 20.46 |
> | | Dining | **56.29** | 46.54 | 36.68 | 30.52 |
> | **n=5** | Bedroom | **68.08** | 64.69 | 43.13 | 31.71 |
> | | Living | **47.56** | 43.21 | 27.53 | 20.73 |
> | | Dining | **53.08** | 46.17 | 37.01 | 27.66 |
> | **n=6** | Bedroom | **69.16** | 62.43 | 48.22 | 31.78 |
> | | Living | **50.15** | 41.69 | 28.72 | 20.12 |
> | | Dining | **53.64** | 44.94 | 33.70 | 31.33 |
>
> **Ablation Study on Triplet Predictor**
>
> We also conducted an additional ablation study to verify the effectiveness of the triplet predictor in bedroom. The table below validates our architectural design.
>
> | # Relations | **SceneNAT (Ours)** | w/o Triplet Predictor | $\Delta$ (Gain) |     |
> | :---------- | :-----------------: | :-------------------: | :-------------: | --- |
> | **n=4**     |      **70.45**      |         62.77         |      +7.68      |     |
> | **n=5**     |      **68.08**      |         61.95         |      +6.13      |     |
> | **n=6**     |      **69.16**      |         64.79         |      +4.37      |     |
> |             |                     |                       |                 |     |
>
>
> > **W1-2.** *Additionally, the paper only validates 11 predefined spatial relations (e.g., "right of", "above") from InstructScene, missing common fine-grained or ambiguous relations, and still faces the issue of limited instruction diversity as noted in InstructScene.*
>
> We appreciate the reviewer's feedback regarding the limitation with respect to predefined set of spatial relations adopted from InstructScene.
>
> **Generalization via Semantic Embedding Space**
> SceneNAT processes instruction features via the pre-trained CLIP text encoder. A key property of the CLIP embedding space is that semantically similar phrases map to proximal vectors.
> Consequently, the model can interpret diverse natural language expressions (e.g., "next to", "beside", "sitting by") by mapping them to the most geometrically appropriate constraint from our canonical set (e.g., "closely right of").
> Also this can be extended to handle "open-vocabulary" style instructions with novel objects (e.g., cradle).
> We note that this capability fundamentally relies on the alignment quality of the embedding space.
> While CLIP provides a robust baseline, we anticipate that integrations with more advanced VLMs will further enhance this open-vocabulary understanding as part of the future development.
> Furthermore, one can explore architectures that leverage Large Language Models (LLMs) to directly parse in-the-wild instructions into symbolic triplet vectors, thereby explicitly bridging diverse natural language to our geometric constraints.
>
> We recognize that demonstrating this generalization capability is crucial, a point also highlighted by **R-vGcy, m5Vp** regarding instruction diversity. We are currently preparing qualitative results using instructions with diverse, unseen predicates or open-voca objects to empirically validate this capability. We **will include these qualitative results in the Appendix** and notify all reviewers as soon as the revision is uploaded.

---

> ### Author Response · Authors · 2025-11-20
> **Official Author Response to Reviewer EG12 (2/2)**
>
> # Discretization analysis
>
> > **W2.** *Unaddressed Discretization Bias and Error. This work discretizes ... but does not analyze how the granularity of discretization impacts generation quality. This is a significant oversight: ...*
>
> We acknowledge the importance of analyzing discretization granularity. To address this, we conducted an ablation study varying the spatial bin size from 32 to 256 across all layout attribtues.
>
> | Room Type   | Metrics     | **32 bins** | **64 bins (reported)** | **128 bins** | **256 bins** |
> | :---------- | :---------- | :---------: | :----------------: | :----------: | :----------: |
> | **Bedroom** | iRecall (%) |    70.17    |     **70.45**      |    69.93     |    68.02     |
> |             | FID         |   110.28    |       109.55       |    109.69    |  **108.56**  |
> | **Living**  | iRecall (%) |    49.19    |       50.01        |    50.80     |  **52.60**   |
> |             | FID         |   109.30    |       110.28       |  **108.98**  |    110.41    |
> | **Dining**  | iRecall (%) |    53.36    |       56.29        |    56.02     |  **59.52**   |
> |             | FID         |   130.70    |       129.65       |  **127.72**  |    129.92    |
>
> The experimental results indicate that the proposed model maintains consistent performance across different granularities.
> * **Coarser Bins**: While coarser bins result in a marginal reduction in iRecall, our model still surpasses state-of-the-art baselines and the overall generation quality remains consistent.
> * **Finer Bins**: Increasing the bin density sometimes provides marginal gains both in iRecall and FID. Importantly, we do not observe the performance degradation that might comes from increased search space complexity (related to uncertainty) in finer discretizations.
> * **Design Choice**: We set 64 bins as the default configuration as this setting offers an reasonable balance between generation quality and model size.
>
> Also, we will include both the table and qualitative results about the discretization artifacts (minor misalignments) in Appendix with the table above in the revised version as soon as possible .
>
> We hope these additional experiments and clarifications satisfactorily address your concerns, and we welcome any further discussion.
>
> ---
> **References**
>
> [1] Radford, A., et al. Learning transferable visual models from natural language supervision. ICML, 2021.

---

> > ### Author Response · Authors · 2025-11-25
> > **Follow-up Author Response to Reviewer EG12 (1)**
> >
> > # Additional experiments regarding generalization capacity
> >
> > Here are the additional experimental results for more diverse instruction including larger object vocabularies that are **unseen during training**.
> >
> > While our training strategy leverages the subject-predicate-object structure as a strong inductive bias for precise spatial control, we investigated the model's behavior on inputs that deviate from this rigid format. We categorized "in-the-wild" instructions into four types to analyze the boundaries of our current framework:
> >
> > **1. Role and Functionality**
> > * *Example:* `Create a bedroom that can also function as a workspace` or `A dining room suitable for hosting wine gatherings.`
> > * *Results:* The model successfully handles these instructions. Leveraging the generalized semantic embedding space of the CLIP text encoder, the model maps abstract functional concepts to concrete object categories (e.g., "workspace" $\to$ desk/chair, "wine gathering" $\to$ wine cabinet). The Scene Decoder then places these objects based on learned co-occurrence priors.
> >
> > **2. Aesthetic and Abstract Descriptions**
> > * *Example:* `A bright, playful bedroom for a young child` or `A simple, minimalist living room.`
> > * *Results:* Similar to our zero-shot stylization task, the model effectively synthesizes scenes that match these descriptors. It combines scene priors with attribute tokens to select appropriate textures and shapes, demonstrating that aesthetic conditioning transfers well even without explicit structural templates.
> >
> > **3. Negative Constraints**
> > * *Example:* `Avoid placing the bed directly facing the wardrobe` or `Don't put any nightstand next to the bed.`
> > * *Results:* The model struggles with negations (e.g., "Don't", "Avoid"). Since the training data consists primarily of positive constructive instructions, the attention mechanism tends to focus on the object entities (e.g., "nightstand") while ignoring the negation token, often resulting in the generation of the prohibited object or relation.
> >
> > **4. High-Level Spatial Reasoning**
> > * *Example:* `Use the bookshelf to create a separation between the dining area and the relaxing area.`
> > * *Results:* While the model correctly infers the necessary objects (bookshelf, sofa, table), it fails to deduce the complex spatial arrangement required for "separation." The triplet predictor is designed to trigger on explicit relational cues; without them, it cannot map abstract spatial goals into geometric constraints.
> >
> > The model exhibits strong semantic generalization (cases 1 & 2), it faces challenges with abstract spatial logic (cases 3 & 4) due to the absence of explicit relational information.
> >
> > To address this challenge, an LLM can be seamlessly integrated into our pipeline to paraphrase abstract directives into explicit natural language instructions containing the required relational information. We verified that off-the-shelf LLMs can effectively convert implicit constraints into concrete spatial descriptions. For example, regarding the instruction to create a separation, an LLM can translate this into a specific description that implicitly contains `(bookshelf, left of, dining table)` and `(sofa, left of, bookshelf)`. This integration combines the high-level reasoning capabilities of LLMs with the precise spatial generation of SceneNAT and presents an exciting direction for future research to fully bridge the gap between natural language and 3D synthesis.
> >
> > ## **Open-Vocabulary Generalization**
> >
> > We conducted qualitative tests using predicates (e.g., *“nearby”, “next to”, “beside”, “around”*) and objects (e.g., *sofa → couch*, *nightstand → bedside table*, *wardrobe → armoire*, *tv stand → entertainment center*) which were **unseen during training**. SceneNAT is capable to map these unseen expressions to the appropriate spatial or semantic concept via the CLIP embedding space, producing plausible and spatially consistent scenes.

---

> > > ### Author Response · Authors · 2025-11-27
> > > **Follow-up Author Response to Reviewer EG12 (2)**
> > >
> > > Dear Reviewer EG12,
> > >
> > > As we approach the final 5 days of the discussion period, we respectfully invite you to review our responses along with the revised manuscript.
> > >
> > > We have made significant efforts to address your concerns through additional experiments and textual revisions. We firmly believe these updates have strengthened the quality and rigor of our work.
> > >
> > > If you find that our responses and revisions have satisfactorily resolved your concerns, we would be grateful if you could consider updating your score. We remain available for any further discussions should you have additional questions.
> > >
> > > Sincerely,
> > > The Authors

---

### Official Review · Reviewer_m5Vp · 2025-10-30

**Soundness:** 3
**Presentation:** 3
**Contribution:** 2
**Rating:** 4
**Confidence:** 3

**Summary:**

SceneNAT proposes a masked non-autoregressive transformer for language-guided indoor scene synthesis. It discretizes semantic and spatial attributes, applies attribute- and instance-level masking, and adds a relation triplet predictor (subject, predicate, object). Compared to autoregressive/diffusion baselines, it targets fewer decoding passes with better efficiency while improving semantic compliance and spatial arrangement on 3D-FRONT.

**Strengths:**

1. The paper introduces a masked non-autoregressive Transformer for language-guided 3D scene synthesis, augmented with an explicit triplet-based relational module. This design is well-motivated and yields favorable decoding efficiency relative to conventional autoregressive or diffusion approaches.
2. On synthetic benchmarks, the method attains strong results on instruction adherence (iRecall), perceptual realism (FID), and object-level recall, while maintaining fast inference with the non-autoregressive bench.

**Weaknesses:**

1. The comparisons largely stop at pre-2025 methods and omit recent state-of-the-art systems (e.g., ReSpace [1])
2. The paper lacks qualitative or quantitative diagnostics (e.g., relation violations, object collisions, discretization artifacts and so on).
3. All results are on synthetic data; there is no real-world study (assets/layouts) or human evaluation, so external validity and deployment readiness remain unclear.
4. The manuscript does not examine multi-room layouts, longer relational prompts, larger object vocabularies, the senarios tested are pretty narrow.

[1] ReSpace: Text-Driven 3D Indoor Scene Synthesis and Editing with Preference Alignment

**Questions:**

1. Can you also specific the VQVAE design and training time on this?
2. Can the approach handle multi-room scenes and longer prompts (beyond the current relation cap)? What are the runtime/memory curves versus object count and room count, and how stable is decoding under these settings?
3. How does the model perform with real-world data?
4. More recent baseline will be helpful to understand the recent works and trends.

---

> ### Author Response · Authors · 2025-11-22
> **Official Author Response to Reviewer m5Vp (1/4)**
>
> Dear reviewer m5Vp,
>
> We sincerely thank the reviewer for the thoughtful assessment and constructive feedback. We are encouraged that the reviewer recognized the well-motivated design framework and its favorable decoding efficiency.
>
> # Clarification on VQ-VAE Design and Training Cost
>
> > **Q1:** *Can you also specify the VQVAE design and training time on this?*
>
> The VQ-VAE utilizes a frozen OpenShape encoder (PointBERT-ViT-G14) [1] to extract 1280-dimensional semantic features from 3D point clouds. These features are processed by a Transformer-based encoder-decoder architecture with a vector-quantization bottleneck. Specifically, the model employs a codebook of size 64 with a dimension of 512 to compress each object into a sequence of 4 ordered discrete tokens, following InstructScene.
>
> Training this VQ-VAE module requires approximately 10 hours on a single NVIDIA RTX A5000 GPU using the 4,000+ objects from the filtered 3D-FRONT dataset. We will add these specifications in the Appendix of the revised paper.
>
> # Multi-room synthesis
> > **Q2-1, W4**. *Can the approach handle multi-room scenes?*
>
> We agree that this is an important direction for future research, but we must clarify the distinction between the two problem spaces.
>
> Multi-room layout generation, as seen in studies like Holodeck [2], is effectively tackled using hierarchical approaches. These methods prioritize determining the macro level architectural structure, such as ***room count, connectivity, and door placement***, with the optimization focusing on **architectural validity and structural planning**.
>
> In contrast, we focus on the lower level challenge of indoor scene synthesis, which involves achieving high accuracy and efficiency in modeling complex relational constraints maintaining high-quality results.
> We chose to focus on single-room arrangement, based on the understanding that each problem domain is most effectively addressed by a distinct, specialized modeling strategy.
>
>
> # Analysis beyond the current relation cap
> > **Q2-2, W4**. *Can the approach handle longer relational prompts?*
>
> ### **Robustness against Baselines on Complex Instructions**
> We conducted additional experiments on instructions containing 5 and 6 relational constraints using the model trained with $N_q=4$. As templated instructions with 5 or 6 relational constraints often exceed the CLIP [3] token limit, we additionally pre-processed these instructions by pruning non-essential descriptive words while preserving the core subject-predicate-object $(s, p, o)$ triplets.
>
> As shown in the results below, while InstructScene shows a degradation of iRecall as the number of relations increases, SceneNAT remarkably maintains consistent performance across all room types. We highlight that sustaining such high iRecall is a non-trivial achievement, especially considering that these complex scenarios were unseen during training. We interpret this results that training the scene decoder along with auxiliary triplet predictor promotes the inference of relational information (i.e., layout).
>
> | # Relations | Room Type | **SceneNAT (Ours)** | InstructScene | DiffuScene | ATISS |
> | :---------- | :-------- | :-----------------: | :-----------: | :--------: | :---: |
> | **n=4**     | Bedroom   |      **70.45**      |     66.72     |   45.98    | 31.30 |
> |             | Living    |      **50.01**      |     47.97     |   27.39    | 20.46 |
> |             | Dining    |      **56.29**      |     46.54     |   36.68    | 30.52 |
> | **n=5**     | Bedroom   |      **68.08**      |     64.69     |   43.13    | 31.71 |
> |             | Living    |      **47.56**      |     43.21     |   27.53    | 20.73 |
> |             | Dining    |      **53.08**      |     46.17     |   37.01    | 27.66 |
> | **n=6**     | Bedroom   |      **69.16**      |     62.43     |   48.22    | 31.78 |
> |             | Living    |      **50.15**      |     41.69     |   28.72    | 20.12 |
> |             | Dining    |      **53.64**      |     44.94     |   33.70    | 31.33 |
>
> ### **Ablation Study on Triplet Predictor**
>
> We also conducted an additional ablation study to verify the effectiveness of the triplet predictor in bedroom. The table below validates our architectural design.
>
> | # Relations | **SceneNAT (Ours)** | w/o Triplet Predictor | $\Delta$ (Gain) |
> | :---------- | :-----------------: | :-------------------: | :-------------: |
> | **n=4**     |      **70.45**      |         62.77         |      +7.68      |
> | **n=5**     |      **68.08**      |         61.95         |      +6.13      |
> | **n=6**     |      **69.16**      |         64.79         |      +4.37      |

---

> ### Author Response · Authors · 2025-11-22
> **Official Author Response to Reviewer m5Vp (2/4)**
>
> ### **Runtime / Memory analysis**
> > **Q2-3**. *What are the runtime/memory curves versus object count and room count, and how stable is decoding under these settings?*
>
> We measured the peak memory consumption and runtime w.r.t. batch size in the table below. It reports values for **Bedroom ($N=12$) / Living & Dining ($N=21$)** scenes respectively. We emphasize that due to our fixed-size parallel decoding architecture, the computational cost is determined by the maximum object count $N$, making inference cost invariant to the actual number of generated objects in a given scene.
>
> | Batch Size | Object Count ($N$) | Peak Memory (GB) | Mean Runtime (s) |     |
> | :--------- | :----------------: | :--------------: | :--------------: | --- |
> | **1**      |      12 / 21       | 0.5945 / 0.5963  | 0.6223 / 0.6809  |     |
> | **8**      |      12 / 21       | 0.6144 / 0.6300  | 0.6436 / 0.7092  |     |
> | **32**     |      12 / 21       | 0.6821 / 0.7446  | 0.6543 / 0.7347  |     |
> | **128**    |      12 / 21       | 0.9530 / 1.1916  | 1.3459 / 1.8145  |     |
>
> **Memory Analysis:** Memory usage is dominated by fixed model parameters, resulting in a highly stable sub-linear scaling curve. Increasing the batch size by 128x results in roughly a 2x increase in peak memory. Regarding object count, the difference is negligible at lower batch sizes. At batch size of 128, denser scenes ($N=21$) show an approximate 25% increase in memory usage compared to sparser scenes ($N=12$). The peak memory reaches approximately 1.2 GB, ensuring **decoding stability** on standard hardware.
>
> **Runtime Analysis:** At a batch size 128, it costs 35% more processing time for living and dining rooms ($N=21$) compared to bedrooms ($N=12$) mainly due to increased attention complexity. However, the absolute difference is marginal (less than 0.5s), confirming that the model maintains **high throughput**.
>
> # Regarding Real-World Data and Deployment
>
> > **W3.** *All results are on synthetic data; there is no real-world study (assets/layouts) or human evaluation, so external validity and deployment readiness remain unclear.* \
> >**Q3.** *How does the model perform with real-world data?*
>
> We thank the reviewer for the insightful question regarding the external validity and deployment readiness of our model. We would like to clarify the intended deployment scenario of SceneNAT and the scope of our contribution.
>
> ### **Simulation-Ready Scene Synthesis**
>
> SceneNAT targets controllable and semantically coherent 3D indoor scene generation for virtual staging and interactive layout design. In these settings, which are increasingly relevant to embodied and physical AI research, curated synthetic assets are generally preferred over raw real-world scans. Synthetic assets offer clean geometry and well-defined semantics. These properties are crucial for reliable physics simulation and agent training. SceneNAT aligns with this objective to ensure the generated environments are directly usable in simulation.
>
> ### **Architectural Scope and Distinction from Real-World Generation**
>
> We acknowledge that applying our model to generate or reconstruct real-world geometry would require a fundamental architectural shift. SceneNAT currently operates as a layout solver that retrieves assets from a closed synthetic library. To generate arbitrary real-world scenes, the architecture would need to incorporate a generative mesh decoder or a scan-to-asset alignment pipeline. For instance:
>
> * **DiffInDScene** [4]: This method employs a cascaded diffusion framework to generate high-fidelity geometry (TSDF) from scratch. Its primary focus is on detailed shape generation and geometry refinement rather than the structural arrangement of layouts.
> * **MetaScenes [5], ACDC [6]**: These approaches target Real2Sim reconstruction. MetaScenes, for example, constructs digital replicas from real-world scans using a complex pipeline that involves point cloud segmentation and multi-modal asset retrieval to match specific visual observations.
>
> In contrast to these approaches, SceneNAT focuses on the text-driven synthesis of diverse indoor layouts rather than mesh generation or visual reconstruction. While all these directions are valuable, they serve different deployment contexts and require distinct modeling strategies.
>
> ---
> To the best of our knowledge, effectively all recent state-of-the-art methods in text-conditioned indoor scene generation rely on synthetic datasets like 3D-FRONT. This is due to the scarcity of large-scale and text-annotated real-world 3D datasets. Consequently, the reliance on synthetic data is a **shared challenge across the field** rather than a specific limitation of our work alone. We recognize bridging this gap as a critical direction for the community and will discuss these potential extensions in the revised manuscript.

---

> ### Author Response · Authors · 2025-11-22
> **Official Author Response to Reviewer m5Vp (3/4)**
>
> ### **Human Evaluation**
>
> To complement our quantitative metrics, we conducted a user study involving 42 participants. In this study, we compared the results of SceneNAT against three baseline methods. Participants were presented with 18 sets of qualitative comparisons, each displaying the generated scenes alongside their corresponding text instructions.
>
> For each set, participants were instructed to select the **Top-1 and Top-2** results based on a holistic evaluation of two criteria:
> 1.  **Instruction Alignment:** How faithfully the scene adheres to the spatial and semantic constraints described in the text.
> 2.  **Overall Quality:** Which scene is the most visually plausible and aesthetically pleasing.
>
> We will include a histogram summarizing the user preferences in the revised Appendix shortly.
>
> # Comparisons with recent SOTA baselines
> > **W1**. *The comparisons largely stop at pre-2025 methods and omit recent state-of-the-art systems (e.g., ReSpace)* \
> > **Q4**. *More recent baseline will be helpful to understand the recent works and trends.*
>
> We appreciate the suggestion to benchmark against more recent systems. We fully agree that discussing recent state-of-the-art baselines is crucial for understanding current trends and positioning our contributions effectively.
>
> Regarding to **ReSpace [7]**, we identified this as a concurrent ICLR submission. More critically, its source code is not publicly available, which makes conducting a fair comparison on text-conditioned generation is infeasible.
>
> Similarly, regarding the other recent models (e.g., **COFS [8]**, **Forest2Seq [9]**, **CasaGPT [10]**) suggested by **R-kcWb**, we verified that their official implementations are either unreleased or empty. Furthermore, these methods also inherently target different objectives, such as boundary or shape conditioning, rather than our specific text-conditioned task.
>
> While these limitations prevent retraining them for our benchmark, we are actively identifying the relevant recent SOTA baseline and **remain committed to rigorously reimplementing** a suitable candidate from scratch if necessary, in order to fully address the concerns raised by Reviewers R-m5Vp and R-kcWb. We will notify all reviewers if any updates available as soon as possible.
>
> # Additional diagnostics
> > **W2.** *The paper lacks qualitative or quantitative diagnostics (e.g., relation violations, object collisions, discretization artifacts and so on).*
>
> We appreciate the reviewer's suggestion to include more comprehensive diagnostics. We address each point below:
>
> ### **1. Relation Violations**
>
> We quantitatively evaluate relation violations using the **iRecall** metric (Table 1 in the main paper), which measures the percentage of generated spatial relations that adhere to the instruction. To provide further insight into failure modes and success cases, we **will include detailed qualitative visualizations of relation violations** in the revised Appendix.
>
> ### **2. Object Collisions**
>
> We conducted an additional quantitative evaluation on object collisions. To ensure precise detection, we computed intersections using oriented bounding boxes. We employed three specific metrics to capture different aspects of physical plausibility:
>
> 1.  **Average Intersection Volume (`inter_vol_mean`)**: This represents the average intersection volume per object pair in a scene, indicating the average density of collisions.
> 2.  **Total Intersection Volume (`inter_vol_sum`)**: This measures the sum of intersection volumes across all object pairs in the scene. It quantifies the absolute total amount of physical interference, serving as a global indicator of layout validity.
> 3.  **IoMin Score (`IoMin`)**: This is the average ratio of the intersection volume to the volume of the smaller object in the pair ($\frac{Inter}{\min(Vol_A, Vol_B)}$). This metric captures the **degree** of collisions, effectively handling size imbalance (e.g., a small cup buried inside a table) which standard IoU might underrepresent.
>
> | Room Type | Method | `inter_vol_mean` ($\downarrow$) | `inter_vol_sum` ($\downarrow$) | `IoMin` ($\downarrow$) |
> | :--- | :--- | :---: | :---: | :---: |
> | **Bedroom** | ATISS | 0.041 | 253.75 | 0.384 |
> | | DiffuScene | **0.012** | *74.20* | **0.227** |
> | | InstructScene | 0.018 | 79.31 | *0.242* |
> | | **SceneNAT (Ours)** | *0.017* | **69.57** | 0.257 |
> | **Living** | ATISS | 0.008 | 466.67 | 0.193 |
> | | DiffuScene | *0.007* | 260.16 | 0.158 |
> | | InstructScene | **0.006** | *194.68* | **0.152** |
> | | **SceneNAT (Ours)** | *0.007* | **151.17** | *0.153* |
> | **Dining** | ATISS | 0.011 | 363.56 | 0.259 |
> | | DiffuScene | *0.010* | 257.88 | *0.218* |
> | | InstructScene | *0.010* | *200.89*| 0.222 |
> | | **SceneNAT (Ours)** | **0.008** | **157.18** | **0.207** |

---

> ### Author Response · Authors · 2025-11-22
> **Official Author Response to Reviewer m5Vp (4/4)**
>
> #### **Analysis of Physical Plausibility:**
> * While DiffuScene utilizes an explicit IoU loss, its advantage is largely confined to bedrooms. In living and dining rooms, SceneNAT consistently outperforms DiffuScene across all metrics. For instance, in dining rooms, SceneNAT achieves the best scores in every category (`inter_vol_mean`: 0.008, `inter_vol_sum`: 157.18, `IoMin`: 0.207).
> * Notably, SceneNAT achieves the lowest total intersection volume (`inter_vol_sum`) across all room types. This indicates global context modeling of SceneNAT is superior at minimizing the total amount of physical violation in the entire scene. This is achieved without explicit collision supervision, validating that our masked generative approach effectively learns physical constraints from the data distribution.
>
> ### **3. Discretization Artifacts**
>
> To analyze the impact of discretization, we conducted an ablation study varying the spatial bin size from 32 to 256. As shown below, our model maintains consistent performance in both structural accuracy (iRecall) and visual quality (FID) across all granularities.
>
> | Room Type | Metrics | **32 bins** | **64 bins (Ours)** | **128 bins** | **256 bins** |
> | :--- | :--- | :---: | :---: | :---: | :---: |
> | **Bedroom** | iRecall (%) | 70.17 | **70.45** | 69.93 | 68.02 |
> | | FID | 110.28 | 109.55 | 109.69 | **108.56** |
> | **Living** | iRecall (%) | 49.19 | 50.01 | 50.80 | **52.60** |
> | | FID | 109.30 | 110.28 | **108.98** | 110.41 |
> | **Dining** | iRecall (%) | 53.36 | 56.29 | 56.02 | **59.52** |
> | | FID | 130.70 | 129.65 | **127.72** | 129.92 |
>
> This robustness confirms that the chosen discretization does not introduce significant performance degradation. We **will also include qualitative examples in the revised Appendix** to visually demonstrate the minor misalignments associated with discretization.
>
> ---
> We hope these additional experiments and clarifications satisfactorily address your concerns, and we welcome any further discussion.
>
> **References**
>
> [1] Liu, M., et al. Openshape: Scaling up 3d shape representation towards open-world understanding. NeurIPS, 2023.\
> [2] Yang, Y., et al. Holodeck: Language guided generation of 3d embodied ai environments. CVPR, 2024.\
> [3] Radford, A., et al. Learning transferable visual models from natural language supervision. ICML, 2021.\
> [4] Ju, X., et al. Diffindscene: Diffusion-based high-quality 3d indoor scene generation. CVPR, 2024.\
> [5] Yu, H., et al. METASCENES: Towards Automated Replica Creation for Real-world 3D Scans. CVPR, 2025.\
> [6] Sakaridis, C., et al. ACDC: The adverse conditions dataset with correspondences for semantic driving scene understanding. ICCV, 2021.\
> [7] Bucher, M. J., et al. ReSpace: Text-Driven 3D Scene Synthesis and Editing with Preference Alignment. arXiv, 2025.\
> [8] Para, W., et al. Cofs: Controllable furniture layout synthesis. ACM SIGGRAPH, 2023.  \
> [9] Sun, Q., et al. Forest2seq: Revitalizing order prior for sequential indoor scene synthesis. ECCV, 2024.  \
> [10] Feng, W., et al. CasaGPT: cuboid arrangement and scene assembly for interior design. CVPR, 2025.

---

> ### Author Response · Authors · 2025-11-25
> **Follow-up Author Response to Reviewer m5Vp (1)**
>
> # Generalization capacity
> > **W4**. *Larger object vocabularies.*
>
> Here are the additional experimental results for more diverse instruction including larger object vocabularies that are **unseen during training**.
>
> While our training strategy leverages the subject-predicate-object structure as a strong inductive bias for precise spatial control, we investigated the model's behavior on inputs that deviate from this rigid format. We categorized "in-the-wild" instructions into four types to analyze the boundaries of our current framework:
>
> **1. Role and Functionality**
> * *Example:* `Create a bedroom that can also function as a workspace` or `A dining room suitable for hosting wine gatherings.`
> * *Results:* The model successfully handles these instructions. Leveraging the generalized semantic embedding space of the CLIP text encoder, the model maps abstract functional concepts to concrete object categories (e.g., "workspace" $\to$ desk/chair, "wine gathering" $\to$ wine cabinet). The Scene Decoder then places these objects based on learned co-occurrence priors.
>
> **2. Aesthetic and Abstract Descriptions**
> * *Example:* `A bright, playful bedroom for a young child` or `A simple, minimalist living room.`
> * *Results:* Similar to our zero-shot stylization task, the model effectively synthesizes scenes that match these descriptors. It combines scene priors with attribute tokens to select appropriate textures and shapes, demonstrating that aesthetic conditioning transfers well even without explicit structural templates.
>
> **3. Negative Constraints**
> * *Example:* `Avoid placing the bed directly facing the wardrobe` or `Don't put any nightstand next to the bed.`
> * *Results:* The model struggles with negations (e.g., "Don't", "Avoid"). Since the training data consists primarily of positive constructive instructions, the attention mechanism tends to focus on the object entities (e.g., "nightstand") while ignoring the negation token, often resulting in the generation of the prohibited object or relation.
>
> **4. High-Level Spatial Reasoning**
> * *Example:* `Use the bookshelf to create a separation between the dining area and the relaxing area.`
> * *Results:* While the model correctly infers the necessary objects (bookshelf, sofa, table), it fails to deduce the complex spatial arrangement required for "separation." The triplet predictor is designed to trigger on explicit relational cues; without them, it cannot map abstract spatial goals into geometric constraints.
>
> The model exhibits strong semantic generalization (cases 1 & 2), it faces challenges with abstract spatial logic (cases 3 & 4) due to the absence of explicit relational information.
>
> To address this challenge, an LLM can be seamlessly integrated into our pipeline to paraphrase abstract directives into explicit natural language instructions containing the required relational information. We verified that off-the-shelf LLMs can effectively convert implicit constraints into concrete spatial descriptions. For example, regarding the instruction to create a separation, an LLM can translate this into a specific description that implicitly contains `(bookshelf, left of, dining table)` and `(sofa, left of, bookshelf)`. This integration combines the high-level reasoning capabilities of LLMs with the precise spatial generation of SceneNAT and presents an exciting direction for future research to fully bridge the gap between natural language and 3D synthesis.
>
> ## **Open-Vocabulary Generalization**
>
> We conducted qualitative tests using predicates (e.g., *“nearby”, “next to”, “beside”, “around”*) and objects (e.g., *sofa → couch*, *nightstand → bedside table*, *wardrobe → armoire*, *tv stand → entertainment center*) which were **unseen during training**. SceneNAT is capable to map these unseen expressions to the appropriate spatial or semantic concept via the CLIP embedding space, producing plausible and spatially consistent scenes.

---

> ### Author Response · Authors · 2025-11-27
> **Follow-up Author Response to Reviewer m5Vp (2)**
>
> ### **Comparison with recent state-of-the-art baseline**
>
> To rigorously address the feedback regarding comparisons with recent state-of-the-art systems, we evaluated the feasibility of benchmarking against various concurrent works. We identified *FreeScene (CVPR 2025)* [11] as the most appropriate and rigorous baseline. As a Mixed Graph Diffusion Transformer, it represents a strong state-of-the-art approach with a task formulation comparable to ours. Although the official code is unavailable, we selected it for comparison as its reported metrics allow for a meaningful quantitative assessment, unlike other concurrent works with differing evaluation protocols.
>
> For a fair comparison,**we source the quantitative results for both the standard FreeScene model and all baseline methods directly from Table 3 of the FreeScene paper.** We excluded their "Graph Designer" variant because it utilizes a VLM to infer objects and relations that closely approximate the ground truth, which would provide an unfair advantage over standard text-conditioned baselines. Crucially, while SceneNAT is trained on a harder dataset with up to 4 relational constraints, baselines typically target only 1 or 2. Therefore, we evaluate SceneNAT's iRecall on the subset of instructions with 1 or 2 relational constraints ($N \le 2$) to match the complexity, while visual quality metrics are averaged over the entire test set.
>
> | Room Type | Method | iRecall ($\uparrow$) | FID ($\downarrow$) | FID$_\text{CLIP}$ ($\downarrow$) | KID$_{\times 10^3}$ ($\downarrow$) |
> | :--- | :--- | :---: | :---: | :---: | :---: |
> | **Bedroom** | ATISS | 44.06 | 122.37 | 8.23 | 0.74 |
> | | DiffuScene | 53.32 | 129.34 | 9.66 | 0.81 |
> | | InstructScene | 72.71 | 114.86 | 6.52 | 0.68 |
> | | FreeScene | *73.69* | *111.21* | *6.43* | *0.35* |
> | | **Ours** | **77.96** | **109.55** | **6.19** | **-1.18** |
> | **Living room** | ATISS | 36.31 | 120.10 | 7.43 | 16.44 |
> | | DiffuScene | 39.58 | 135.93 | 10.71 | 29.07 |
> | | InstructScene | 57.21 | 111.52 | 5.91 | 8.65 |
> | | FreeScene | *58.16* | *110.55* | *5.83* | *7.95* |
> | | **Ours** | **59.56** | **110.28** | **5.49** | **6.18** |
> | **Dining room** | ATISS | 32.56 | 134.75 | 9.82 | 24.25 |
> | | DiffuScene | 37.17 | 142.37 | 11.76 | 28.36 |
> | | InstructScene | 61.47 | *129.13* | 8.24 | 15.27 |
> | | FreeScene | **63.39** | **127.28** | *8.01* | *14.83* |
> | | **Ours** | *61.89* | 129.65 | **7.51** | **12.26** |
>
> SceneNAT demonstrates superior or highly competitive performance against FreeScene in the standard setting. In bedroom and living room, SceneNAT outperforms FreeScene across all metrics, achieving significant gains in iRecall. In dining room, while FreeScene holds a slight advantage in iRecall and FID, SceneNAT achieves superior perceptual quality (FID$_\text{CLIP}$ and KID).
>
> Furthermore, our discretization ablation study reveals that increasing the bin resolution to 128 further improves visual quality, achieving an FID of **108.98** in living room and **127.72** in dining room. This effectively matches or surpasses FreeScene's FID across all room types.
>
> **Limitations on Further Comparison**
>
> Please note that since the official code for FreeScene has not been released as of November 27, 2025 (https://github.com/cangmushui/FreeScene) , it is currently infeasible to conduct further comparative experiments such as retraining the model on our exact split or generating qualitative visualizations for a side-by-side comparison. We believe that re-implementing such a complex system without the original source code risks missing critical implementation details, which could lead to an unfair comparison for either approach. Therefore, we limit our evaluation to the reported quantitative metrics which sufficiently demonstrate the competitiveness of SceneNAT.
>
> ---
> [11] Bai, T., et al. FreeScene: Mixed Graph Diffusion for 3D Scene Synthesis from Free Prompts. CVPR, 2025.
>
> ---
> # **Closing Remark**
>
> Dear Reviewer m5Vp,
>
> As we approach the final 5 days of the discussion period, we respectfully invite you to review our responses along with the revised manuscript.
>
> We have made significant efforts to address your concerns through additional experiments and textual revisions. We firmly believe these updates have strengthened the quality and rigor of our work.
>
> If you find that our responses and revisions have satisfactorily resolved your concerns, we would be grateful if you could consider updating your score. We remain available for any further discussions should you have additional questions.
>
> Sincerely,
> The Authors

---

### Official Review · Reviewer_vGcy · 2025-11-03

**Soundness:** 3
**Presentation:** 3
**Contribution:** 3
**Rating:** 6
**Confidence:** 3

**Summary:**

This paper introduces SceneNAT, a Non-Autoregressive Transformer (NAT) for 3D indoor scene generation that addresses the efficiency limitations of AR and diffusion models. The core contribution is a Masked Generative Modeling approach, akin to a BERT-style objective, which reconstructs discretized scene attributes in a few iterative steps. To ensure high-fidelity language control, the model employs a DETR-inspired Triplet Predictor to explicitly parse spatial relations. The final generated attributes guide an Object Retrieval step to build the scene. Experiments demonstrate that SceneNAT achieves SOTA performance, especially in instruction-following and efficiency.

**Strengths:**

+The paper's core insight—that a Non-Autoregressive Transformer is a powerful alternative to AR or Diffusion models for this task—is a significant strength. This approach opens a promising third direction for high-speed, high-quality structured 3D generation.

+The concept of the Triplet Predictor is also a key strength. Decoupling symbolic relation-understanding from the geometric generation task is an intelligent design choice.

+The paper shows impressive performance and efficiency via comprehensive experiments, outperforming existing SoTA methods.

**Weaknesses:**

-A key concern is the reliance on templated data. The training instructions are synthetically generated from structured relations. This implies the model is primarily learning to "invert" this synthetic generation process rather than parsing free-form human language directly.  It remains unclear how the model would generalize to ambiguous, "in-the-wild" human instructions that do not follow the rigid structure. Therefore, the claim of handling "complex instructions" may be somewhat overstated.

-The model appears to use a fixed-size scene matrix with a maximum of N object slots. This is an inflexible architectural choice. It's unclear how the model handles scenes that require a number of objects significantly different from N (either much sparser or much denser than the training data). This contrasts with AR models that naturally support variable-length generation.

-The model has two parallel reasoning systems: the Scene Decoder learns implicit rules of scene plausibility, while the Triplet Predictor enforces explicit user instructions. The paper does not discuss how the model arbitrates conflicts between these two systems. What happens if a user instruction is physically implausible or violates the model's learned common sense? The system's behavior under such contradictory signals is a key aspect of controllability and remains unaddressed

**Questions:**

1. I am curious about Triplet Predictor’s potential to generalize to more ambiguous, real-world instructions that may not follow a strict "subject-predicate-object" template. For example, a more complex instruction with abstract/implicit entities and relations might be: "Don't put the sofa facing the TV wall. Instead, try to arrange the sofas and chairs in a circle for chatting, and make sure not to block the path." Have the authors considered using an LLM to paraphrase the original ground-truth template sentences into more free-form structures?

2. Could the authors specify the number of learnable queries ($N_q$) used in the Triplet Predictor? And what happens to performance if the actual number of relations in an instruction exceeds $N_q$?

3. Please provide more details about the Object Retrieval step. Does the retrieval step find the nearest neighbor in the asset database that matches the predicted appearance tokens?

4. Please specify what guidance scale was used for the main experiments. It would also be interesting to know how this scale impacts the trade-off between instruction fidelity and generation quality.

---

> ### Author Response · Authors · 2025-11-24
> **Official Author Response to Reviewer vGcy (1/3)**
>
> Dear reviewer vGcy,
>
> We sincerely thank the reviewer for the positive assessment and encouraging feedback. We are delighted that the reviewer recognized the potential of our non-autoregressive framework as a promising direction for high-speed generation and the effectiveness of the decoupled triplet predictor design. We address the specific concerns below.
>
> # Generalization Capacity
>
> > **Q1.** *A key concern is the reliance on templated data. ... It remains unclear how the model would generalize to ambiguous, "in-the-wild" human instructions that do not follow the rigid structure.* \
> > **W1**. *... potential to generalize to more ambiguous, real-world instructions ... Have the authors considered using an LLM to paraphrase the original ground-truth template sentences into more free-form structures?*
>
>
> We appreciate the reviewer’s concern regarding the reliance on templated instructions. We agree that demonstrating robustness to free-form, ambiguous, and “in-the-wild” human instructions is an essential part of validating a language-conditioned generation model.
>
> While our training strategy leverages the subject-predicate-object structure as a strong inductive bias for precise spatial control, we investigated the model's behavior on inputs that deviate from this rigid format. We categorized "in-the-wild" instructions into four types to analyze the boundaries of our current framework:
>
> **1. Role and Functionality**
> * *Example:* `Create a bedroom that can also function as a workspace` or `A dining room suitable for hosting wine gatherings.`
> * *Results:* The model successfully handles these instructions. Leveraging the generalized semantic embedding space of the CLIP text encoder, the model maps abstract functional concepts to concrete object categories (e.g., "workspace" $\to$ desk/chair, "wine gathering" $\to$ wine cabinet). The Scene Decoder then places these objects based on learned co-occurrence priors.
>
> **2. Aesthetic and Abstract Descriptions**
> * *Example:* `A bright, playful bedroom for a young child` or `A simple, minimalist living room.`
> * *Results:* Similar to our zero-shot stylization task, the model effectively synthesizes scenes that match these descriptors. It combines scene priors with attribute tokens to select appropriate textures and shapes, demonstrating that aesthetic conditioning transfers well even without explicit structural templates.
>
> **3. Negative Constraints**
> * *Example:* `Avoid placing the bed directly facing the wardrobe` or `Don't put any nightstand next to the bed.`
> * *Results:* The model demonstrates variable capability with negations (e.g., "don't", "avoid"). Although successful in some cases, it is prone to errors because the training data consists primarily of positive constructive instructions. In failure cases, the attention mechanism tends to focus on the object entities ignoring the negation token, resulting in the generation of the prohibited object or relation.
>
> **4. High-Level Spatial Reasoning**
> * *Example:* `Use the bookshelf to create a separation between the dining area and the relaxing area.`
> * *Results:* While the model correctly infers the necessary objects (bookshelf, sofa, table), it fails to deduce the complex spatial arrangement required for "separation." The triplet predictor is designed to trigger on explicit relational cues; without them, it cannot map abstract spatial goals into geometric constraints.
>
> The model exhibits strong semantic generalization (cases 1, 2), it sometimes struggles with abstract spatial logic (cases 3, 4) due to the absence of explicit relational information.
>
> To address this challenge, an LLM can be seamlessly integrated into our pipeline to paraphrase abstract directives into explicit natural language instructions containing the required relational information. We verified that off-the-shelf LLMs can effectively convert implicit constraints into concrete spatial descriptions. For example, regarding the instruction to create a separation, an LLM can translate this into a specific description that implicitly contains `(bookshelf, left of, dining table)` and `(sofa, left of, bookshelf)`. This integration combines the high-level reasoning capabilities of LLMs with the precise spatial generation of SceneNAT and presents an exciting direction for future research to fully bridge the gap between natural language and 3D synthesis.
>
> ## **Open-Vocabulary Generalization**
>
> As **R-EG12** also pointed out, the model’s ability to generalize to unseen vocabulary would be another concern.
>
> To further examine this, we conducted qualitative tests using predicates (e.g., *“nearby”, “next to”, “beside”, “around”*) and objects (e.g., *sofa → couch*, *nightstand → bedside table*, *wardrobe → armoire*, *tv stand → entertainment center*) which were **unseen during training**. SceneNAT is capable to map these unseen expressions to the appropriate spatial or semantic concept via the CLIP embedding space, producing plausible and spatially consistent scenes.

---

> ### Author Response · Authors · 2025-11-24
> **Official Author Response to Reviewer vGcy (2/3)**
>
> # Response to Question regarding Learnable Queries and Relation Limit
>
> > **Q2.** *Could the authors specify the number of learnable queries ($N_q$) used in the Triplet Predictor? And what happens to performance if the actual number of relations in an instruction exceeds $N_q$?*
>
> We appreciate the reviewer's request for clarification regarding the Triplet Predictor. The number of learnable queries $N_q$ is set to 4 for all experiments. This aligns with the maximum number of relational constraints used during the training phase. We will explicitly specify this hyperparameter in the Implementation Details section of the revised manuscript.
>
> ### **Robustness against Baselines on Complex Instructions**
> We conducted additional experiments on instructions containing 5 and 6 relational constraints using the model trained with $N_q=4$. As templated instructions with 5 or 6 relational constraints often exceed the CLIP token limit, we additionally pre-processed these instructions by pruning non-essential descriptive words while preserving the core subject-predicate-object $(s, p, o)$ triplets.
>
> As shown in the results below, while InstructScene shows a degradation of iRecall as the number of relations increases, SceneNAT remarkably maintains consistent performance across all room types. We highlight that sustaining such high iRecall is a non-trivial achievement, especially considering that these complex scenarios were unseen during training. We interpret this results that training the scene decoder along with auxiliary triplet predictor promotes the inference of relational information (i.e., layout).
>
> | # Relations | Room Type | **SceneNAT (Ours)** | InstructScene | DiffuScene | ATISS |
> | :---------- | :-------- | :-----------------: | :-----------: | :--------: | :---: |
> | **n=4**     | Bedroom   |      **70.45**      |     66.72     |   45.98    | 31.30 |
> |             | Living    |      **50.01**      |     47.97     |   27.39    | 20.46 |
> |             | Dining    |      **56.29**      |     46.54     |   36.68    | 30.52 |
> | **n=5**     | Bedroom   |      **68.08**      |     64.69     |   43.13    | 31.71 |
> |             | Living    |      **47.56**      |     43.21     |   27.53    | 20.73 |
> |             | Dining    |      **53.08**      |     46.17     |   37.01    | 27.66 |
> | **n=6**     | Bedroom   |      **69.16**      |     62.43     |   48.22    | 31.78 |
> |             | Living    |      **50.15**      |     41.69     |   28.72    | 20.12 |
> |             | Dining    |      **53.64**      |     44.94     |   33.70    | 31.33 |
>
> ### **Ablation Study on Triplet Predictor**
>
> We also conducted an additional ablation study to verify the effectiveness of the triplet predictor in bedroom. The table below validates our architectural design.
>
> | # Relations | **SceneNAT (Ours)** | w/o Triplet Predictor | $\Delta$ (Gain) |
> | :---------- | :-----------------: | :-------------------: | :-------------: |
> | **n=4**     |      **70.45**      |         62.77         |      +7.68      |
> | **n=5**     |      **68.08**      |         61.95         |      +6.13      |
> | **n=6**     |      **69.16**      |         64.79         |      +4.37      |
>
> # Clarification on Object Retrieval Mechanism
>
> > **Q3.** *Please provide more details about the Object Retrieval step. Does the retrieval step find the nearest neighbor in the asset database that matches the predicted appearance tokens?*
>
> We thank the reviewer for the question. Yes, the retrieval step performs a nearest neighbor search to find the best-matching 3D asset.
>
> The process operates as follows:
>
> 1.  **Category Filtering:** First, we filter the asset database (3D-FUTURE) to select candidate objects that match the class label predicted by our Class Head.
>
> 2.  **Feature Reconstruction:** We map the sequence of 4 discrete appearance tokens predicted by our Appearance Head back to their corresponding codebook vectors. These are then passed through the pre-trained VQ-VAE decoder (from InstructScene) to reconstruct a continuous 1280-dimensional semantic feature vector.
>
> 3.  **Nearest Neighbor Search:** We compute the cosine similarity between this reconstructed feature vector and the pre-computed OpenShape features of all candidate assets. The asset with the highest similarity (nearest neighbor) is retrieved and placed into the scene.
>
> We will add these specific details to the Appendix in the revised version.

---

> ### Author Response · Authors · 2025-11-25
> **Official Author Response to Reviewer vGcy (3/3)**
>
> # Analysis on CFG
>
> > **Q4.** *Please specify what guidance scale was used for the main experiments. It would also be interesting to know how this scale impacts the trade-off between instruction fidelity and generation quality.*
>
> We thank the reviewer for the question regarding the Classifier-Free Guidance (CFG) scale.
>
> To analyze the impact of the guidance scale on the trade-off between instruction fidelity and generation quality, we conducted an additional ablation study by varying the scale $w$ from 0 to 10. The results are summarized in the table below.
>
> | Guidance Scale ($w$)     |  0.0   |  0.5   |    1.0     |  1.5   |    2.0    |  3.0   |  4.0   |  5.0   |  10.0  |
> | :----------------------- | :----: | :----: | :--------: | :----: | :-------: | :----: | :----: | :----: | :----: |
> | **iRecall** ($\uparrow$) | 11.98  | 66.26  |   70.45    | 71.39  | **72.62** | 68.22  | 63.81  | 66.99  | 43.52  |
> | **FID** ($\downarrow$)   | 128.17 | 111.61 | **109.55** | 110.10 |  111.49   | 113.25 | 116.99 | 120.15 | 133.45 |
>
> We observe a distinct trade-off across the guidance scales. While unconditional generation ($w=0$) expectedly yields poor instruction adherence, increasing the scale up to $2.0$ reveals a clear trade-off where iRecall improves (peaking at 72.62%) at the cost of a slight degradation in visual quality. However, increasing the scale beyond $2.0$ leads to performance drops in both metrics.
>
> Based on these results, we set $w=1.0$ as our default setting to maintain the optimal balance between high-quality generation and faithful instruction following. We **will include** the experimental results in the revised Appendix.
>
> # Discussion on fixed-size scene matrix
>
> > **W2.** *The model appears to use a fixed-size scene matrix with a maximum of N object slots. ... much sparser or much denser ...AR models that naturally support variable-length generation.*
>
> We acknowledge that the fixed-size representation is an architectural constraint compared to the variable-length nature of autoregressive models. However, this design is a deliberate trade-off essential for our **Masked Indoor Scene Modeling (MISM)** framework to enable parallel decoding and bidirectional context modeling.
>
> Despite the fixed size, the model effectively handles variable complexities.
> * **Sparse Scenes**: The model utilizes learnable "empty" tokens for unused slots. As our training data already includes scenes with as few as 3 objects, the model is well-adapted to handling high sparsity within the fixed matrix.
> * **Dense Scenes**: The capacity $N$ is set to safely exceed the maximum object count observed in the 3D-FRONT dataset, which is approximately 21 per room. Thus, the model covers effectively 100% of the realistic layouts in the target distribution.
>
> We also note that this fixed-size representation aligns with state-of-the-art baselines like DiffuScene and InstructScene, confirming it as a standard practice for parallel generative models. However, we agree that applying SceneNAT to significantly denser environments would necessitate increasing $N$, identifying the validation of our framework in such regimes as an important direction for future research.
>
> # **Behavior Under Contradictory or Physically Implausible Instructions**
>
> > **W3.** *The model has two parallel reasoning systems: ... how the model arbitrates conflicts between these two systems. What happens if a user instruction is physically implausible or violates the model's learned common sense? ...*
>
> We think this is very insightful question.
>
> In our architecture, the scene decoder and triplet predictor do not operate as competing systems. Instead, the output of the triplet predictor is incorporated into the decoder through cross-attention, serving as soft conditional guidance rather than hard constraints.
>
> To examine this behavior, we tested explicitly contradictory or physically implausible instructions, such as `placing a sofa above a pendant lamp` or `positioning a nightstand below a bed`. In such cases, SceneNAT does not attempt to enforce the geometrically impossible relation. The learned priors of the scene decoder act as a validity filter, overriding the implausible guidance. Consequently, the model produces physically reasonable alternatives that remain semantically aligned with the user's intent (e.g., placing the sofa *near* the lamp rather than literally *above* it). These observations indicate that SceneNAT naturally resolves conflicts by prioritizing physical plausibility when instructions violate common sense, ensuring robust generation. We will include qualitative examples of these behaviors in the revised Appendix.
>
> ---
> We hope these additional experiments and clarifications satisfactorily address your concerns, and we welcome any further discussion.

---

> > ### Author Response · Authors · 2025-11-27
> > **Follow-up Author Response to Reviewer vGcy (1)**
> >
> > Dear Reviewer vGcy,
> >
> > As we approach the final 5 days of the discussion period, we respectfully invite you to review our responses along with the revised manuscript.
> >
> > We have made significant efforts to address your concerns through additional experiments and textual revisions. We firmly believe these updates have strengthened the quality and rigor of our work.
> >
> > If you find that our responses and revisions have satisfactorily resolved your concerns, we would be grateful if you could consider updating your score. We remain available for any further discussions should you have additional questions.
> >
> > Sincerely, The Authors

---

### Author Response · Authors · 2025-11-25
**Notification of Revised Manuscript**

Dear Reviewers,

We sincerely thank for the time and effort dedicated to reviewing our work. We are truly grateful for the constructive feedback, which has significantly helped us strengthen the quality and rigor of our manuscript.

We have uploaded a revised version of the paper reflecting your suggestions. All major changes and additions are highlighted in blue.

**Summary of Key Updates:**

**Main Manuscript:**
* **Robustness Analysis:** Experiments on robustness to unseen relational complexity (Section 4.3).
* **Ablation Study:** An ablation study on discretization granularity (Section 4.6).
* **Clarification:** Revised the caption of Figure 2 for better clarity.

**Appendix:**
* **Implementation Details:**  Specifications for VQ-VAE and detailed explanations of the object retrieval step (Appendix A.1).
* **Additional Ablations:** An ablation study on Classifier-Free Guidance (CFG) scales (Appendix A.6).
* **Physical Plausibility:** Quantitative experiments on object collision and layout validity (Table 1, 9).
* **Comparison with more recent baseline:** Quantitative comparison with FreeScene (Appendix A.8).
* **Efficiency Analysis:** The runtime and memory analysis (Appendix B).
* **Discussion:** Discussion on real-world experiments and future directions (Appendix E).
* **Human Evaluation:** Results of the user study (Appendix G).
* **Qualitative Results:** Qualitative results for open-vocabulary, contradictory, and in-the-wild instructions to clarify the generalization capacity. (Appendix H).
* **Failure Analysis:** Qualitative results of failure cases (relation violation, object attribute violation, implausible layouts)  (Appendix I).

We hope these revisions satisfactorily address your main concerns.
It would be much appreciated for a re-evaluation of the revised manuscript to confirm that the updates adequately reflect reviewers' valuable suggestions.
We firmly believe that these updates have improved our manuscript and welcome any additional feedback or questions.

Sincerely,
The Authors

---

### Meta-Review · Area_Chair_qdz9 · 2026-01-07

**Summary:**

The paper presents SceneNAT, a single-stage, non-autoregressive transformer (NAT) designed for generating 3D indoor scenes from natural language instructions. Reviewers' concerns center on four core areas:
1. Generalization & Instruction Adaptability: Reviewers question the model's reliance on templated data and its ability to handle "in-the-wild" human language or instructions exceeding the training capacity of 4 relations.
2. Experimental Completeness: Reviewer m5Vp and kcWb request comparisons with SOTA baselines (especally relevant works in 2025), a more detailed study about the granularity of discretization, and the human evaluations.
3. Technical Clarity & Architecture Novelty: Reviewer kcWb expresses skepticism regarding whether masked modeling (like BERT) could be used for generation rather than just representation, and also questions the architectual distinctions from previous work COFS.
4. Practical Validity: Concerns included fixed-size scene matrices, multi-room support, and the missing of failure cases.
While the majority of concerns have been addressed during the rebuttal phase, Reviewer kcWb remains skeptical regarding the architectural novelty of the proposed method, as well as the alignment between natural language instructions and scene layouts.

**Reviewer Concerns:**

Addressed Concerns
1. Generalization & Instruction Adaptability: Authors validate robustness to 5–6 relational constraints (exceeding training limits) and "in-the-wild" instructions (functional/aesthetic prompts), clarify fixed-size matrix design (handles sparse/dense scenes via empty tokens), and explain conflict arbitration (scene decoder prioritizes physical plausibility over implausible instructions).
2. Experimental Completeness: Authors add comparisons with 2025 SOTA method FreeScene, the ablation study of discretization (32~256 bins), and a 42-participant user study.
3. Practical Validity: Authors explain the matrix design’s compatibility with parallel decoding, treat multi-room synthesis as a separate task, and add failure case analysis (relation violations, collisions, discretization artifacts).

Outstanding Concerns

Novelty doubt: Reviewer kcWb remained unconvinced of novelty since the distinctions between this work and COFS have not been sufficiently clarified.

**Reviewer Scores:**

1. Reviewer vGcy (Original Score: 6/10, marginally above acceptance)

Would maintain or slightly increase the score to 7 since most concerns are addressed during the rebuttal phase.

2. Reviewer m5Vp (Original Score: 4/10, marginally below acceptance)

Would maintain or slightly increase the score to 5.

3. Reviewer EG12 (Original Score: 6/10, marginally above acceptance)

Would maintain or slightly increase the score to 7 since most concerns are addressed during the rebuttal phase.

4. Reviewer kcWb (Original Score: 2/10, reject)

Would maintain the original score since this reviewer persists in questioning the novelty and the instruction-scene mismatches.

---

### Decision · Program_Chairs · 2026-01-26

Reject